# Globally asynchronous sulphur isotope signals require re-definition of the Great Oxidation Event

Pascal Philippot[1,2], Janaína N. Ávila [3], Bryan A. Killingsworth [4], Svetlana Tessalina [5], Franck Baton[1], Tom Caquineau[1], Elodie Muller[1], Ernesto Pecoits[1,8], Pierre Cartigny[1], Stefan V. Lalonde [4], Trevor R. Ireland [3], Christophe Thomazo [6], Martin J. van Kranendonk[7] & Vincent Busigny [1]

The Great Oxidation Event (GOE) has been defined as the time interval when sufficient atmospheric oxygen accumulated to prevent the generation and preservation of mass-independent fractionation of sulphur isotopes (MIF-S) in sedimentary rocks. Existing correlations suggest that the GOE was rapid and globally synchronous. Here we apply sulphur isotope analysis of diagenetic sulphides combined with U-Pb and Re-Os geochronology to document the sulphur cycle evolution in Western Australia spanning the GOE. Our data indicate that, from ~2.45 Gyr to beyond 2.31 Gyr, MIF-S was preserved in sulphides punctuated by several episodes of MIF-S disappearance. These results establish the MIF-S record as asynchronous between South Africa, North America and Australia, argue for regional-scale modulation of MIF-S memory effects due to oxidative weathering after the onset of the GOE, and suggest that the current paradigm of placing the GOE at 2.33–2.32 Ga based on the last occurrence of MIF-S in South Africa should be re-evaluated.

[1] Institut de Physique du Globe de Paris, Sorbonne-Paris Cité, Université Paris Diderot, CNRS, 1 rue Jussieu, 75238 Paris, France. [2] Géosciences Montpellier, CNRS-UMR 5243, Université de Montpellier, 34095 Montpellier, France. [3] Research School of Earth Sciences, The Australian National University, 142 Mills Road, Canberra ACT 2601, Australia. [4] European Institute for Marine Studies, UMR6538 Laboratoire Géosciences Océan, Place Nicolas Copernic, 29280 Plouzané, France. [5] John de Laeter Centre for Isotope Research, Bld 301, Faculty of Science and Engineering, Curtin University, GPO Box U1987, Perth, WA 6845, Australia. [6] UMR CNRS/uB6282 Biogéosciences, Université de Bourgogne Franche-Comté, 6 Bd Gabriel, 21000 Dijon, France. [7] School of Biological, Earth and Environmental Sciences, University of New South Wales, Kensington NSW 2052, Australia. [8] Present address: Department of Environmental Sciences, Uruguay Technological University (UTEC), Francisco Antonio Maciel, Durazno, CP 97000, Uruguay. Correspondence and requests for materials should be addressed to P.P. (email: pascal.philippot@umontpellier.fr)

Before ~2.45 Ga ago, atmospheric oxygen was at $pO_2 < 10^{-5}$ present atmospheric level (PAL), as shown by the occurrence of large MIF-S anomalies[1,2] ($\Delta^{33}S$ and $\Delta^{36}S$, see Methods for details) in sedimentary sulphide and sulphate of Archaean age and by photochemical models of the production and preservation of these anomalies[3]. The rise of atmospheric oxygen to above $10^{-5}$ to $10^{-2}$ PAL has been loosely constrained between ~2.50 Ga, the age of the youngest sediments with no, or only minor $\Delta^{33}S$ anomalies (top of the McRae Shale Formation and Whaleback Shale Member of the Brockman Formation, Hamersley Group[4,5]), and ~2.32 Ga, the age of the youngest rocks with strong MIF-S (Rooihoogte and Timeball Hill formations, South Africa[6–8]). However, the persistence and disappearance of the MIF-S record may not directly reflect changes in global atmospheric gas concentrations. It was previously suggested that MIF-S may be recorded for a hundred million years or more after atmospheric oxygenation due to weathering and recycling of an older MIF-S reservoir, as based on evidence from empirical observations[9], and modelling[10]. Crucially, such a memory process would obscure the true timing and tempo of atmospheric oxygenation as inferred from the record of S-MIF alone. Furthermore, as early as 2.50 Ga ago, mass-dependent fractionation of sulphur isotopes (MDF-S, noted $\delta^{34}S$) in sedimentary sulphides

indicates that microbial sulphate reducers were thriving in an ocean experiencing an increased delivery of sulphate derived from oxidative weathering of continental sulphides even before the final disappearance of MIF-S[4,5]. The transition from MIF-S to MDF-S is linked in time with a series of glacial events[11] culminating in a possible snowball Earth[12]. The main constraints used to mark the timing of the GOE worldwide are from sedimentary rocks recording the transition from MIF-S to MDF-S, and associated glacial deposits, from the Transvaal Supergroup in South Africa[6–8] and the Huronian Supergroup in North America[13–16]. In situ U-Pb zircon ages for volcanic tuffs and a Re-Os age for sedimentary sulphide in these successions constrain the timing of four glaciations to between 2.45 and 2.22 Ga, with the three oldest predating 2.31 Ga, and the final glaciation, which is only recognized in South Africa, occurring between ~2.26 and 2.22 Ga[6,17,18]. However, large depositional gaps are reported in the sedimentary successions at both localities, making it difficult to constrain the exact timing of events surrounding the GOE (Fig. 1).

In order to bring new perspective on the chronology of the GOE, we focused our efforts on the Turee Creek Group in Western Australia, which, in contrast to the Transvaal and Huronian supergroups, is characterised by a continuous marine sedimentary succession spanning the GOE and two[19] or possibly

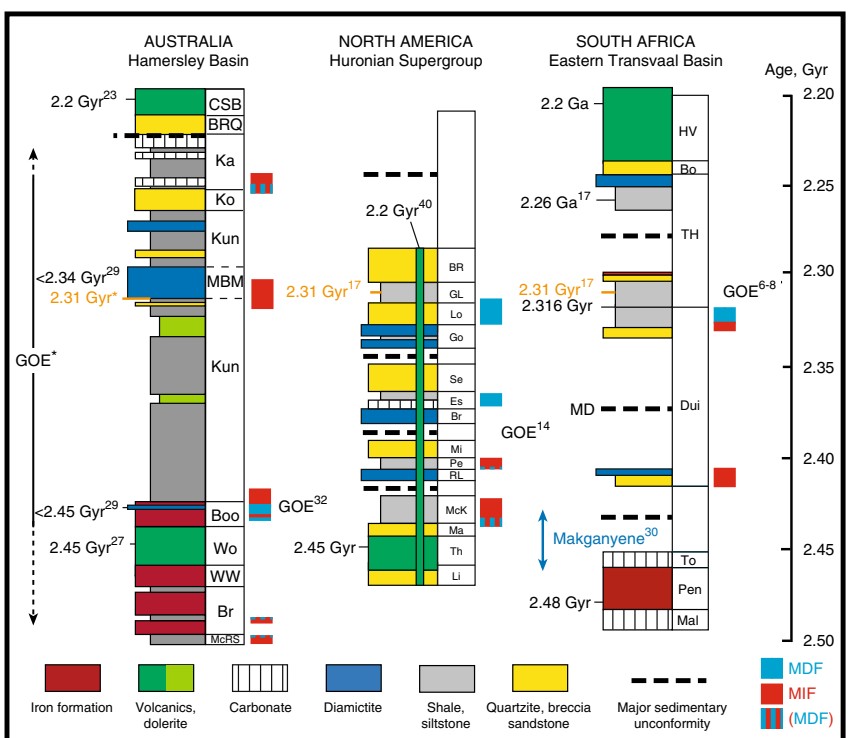

**Fig. 1** New age-calibrated correlation of Late Archaean to Paleoproterozoic sedimentary successions. The Re-Os sulphide age from this study is shown as 2.31 Ga*. Other age constraints are labelled accordingly. Ages with no reference can be found in Rasmussen et al.[17]. The inferred Great Oxidation Event (GOE*) from this study is shown as a ~300 Myr interval based on combined age, sulphur isotope and stratigraphic data. Other GOEs inferred from previous studies are labelled accordingly. The vertical arrow labelled Makganyene[30] refers to the range of age inferred for the Makganyene glacial deposit based on a new U-Pb age of the Ongeluk Volcanics, Transvaal Supergroup, South Africa[30]. MIF, mass-independent fractionation of sulphur isotopes; MDF, mass-dependent fractionation of sulphur isotopes. The mixed light blue-red domain labelled (MDF) shown at the top of the Turee Creek Group (this study), in the Whaleback Member of the Brockman Formation[39] and McRae Shale Formation[10] of the Hamersley Group, and at the base of the Huronian sedimentary column[14–16], corresponds to sedimentary horizons in which strongly attenuated $\Delta^{33}S$ and $\Delta^{33}S/\Delta^{36}S$ systematics are attributed to increases in $pO_2$ above $10^{-5}$ to $10^{-2}$ PAL (see text). Hamersley Basin: McRS, McRae Shale Formation, Br, Brockman Formation, WW, Weeli Wolli Formation, Wo, Woongarra Rhyolite, Boo, Boolgeeda Iron Fm., Kun, Kungarra Fm., MBM, Meteorite Bore Member, Ko, Koolbye Fm., Ka, Kazput Fm., BRG, Beasley River Quartzite, CSB, Cheela Springs Basalt. Huronian Supergroup: Li, Livingstone Creek Fm., Th, Thessalon Fm., Ma, Matinenda Fm., McK, McKim Fm., RL, Ramsay Lake Fm., Pe, Pecors Fm., Mi, Mississagi, Fm., Br, Bruce Fm., Es, Espanola Fm., Se, Serpent Fm., Go, Gowganda Fm., Lo, Lorrain Fm., GL, Gordon Lake Fm., BR, Bar River Fm. Eastern Transvaal Basin: Mal, Malmani Fm., Pen, Penge Iron Fm., To, Tongwane Fm., Dui, Duitschland Fm., TH, Timeball Hill Fm., Bo, Boshoek Fm., HV, Hekpoort Volcanics

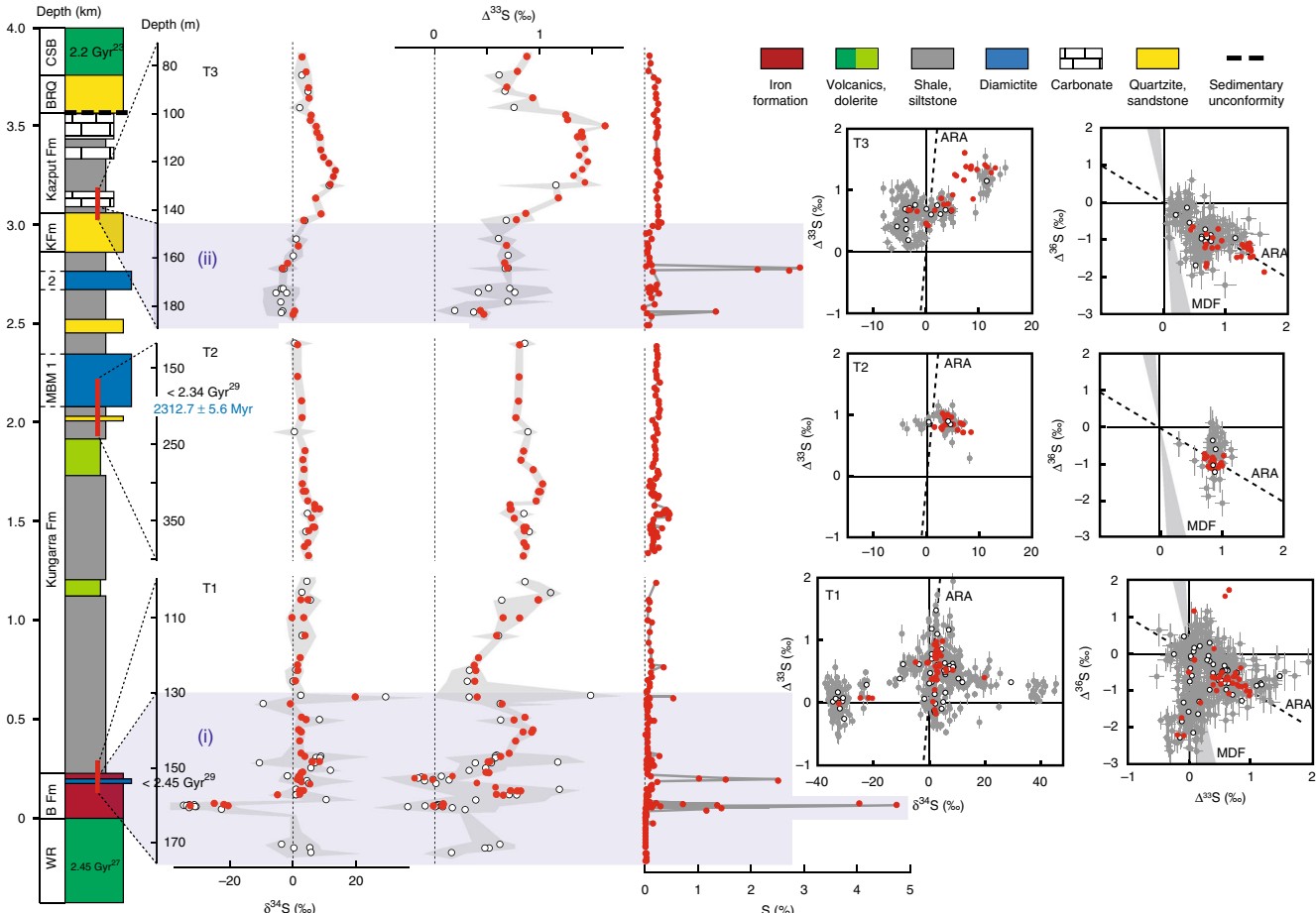

**Fig. 2** Sulphur content and δ³⁴S and Δ³³S profiles of sulphides in the Turee Creek drill cores and associated δ³⁴S-Δ³³S and Δ³³S-Δ³⁶S compositional diagrams. Red dots, bulk analysis; white dots, mean value of in situ analysis for each sample; grey dots, individual spot analysis using the SHRIMP-SI (error bars are standard deviation of 1σ, see Methods). The grey shaded areas in the δ³⁴S and Δ³³S profiles represent the range of individual spot analysis for each sample. The two blue-shaded areas labelled (i) below 130 m depth in T1 and (ii) below 145 m depth in T3 are discussed in the text. ARA, Archaean Reference Array. MDF, mass-dependent fractionation of sulphur isotopes. The stratigraphic section of the Turee Creek Group on the left is modified after Van Kranendonk and Mazumder[19]. Re-Os age of 2312.7 ± 5.6 Ma is from this study. Other age constraints are labelled accordingly. Abbreviations associated with the stratigraphic section are as follows: WR, Woongarra Rhyolite, B Fm, Boolgeeda Iron Fm., Kun, Kungarra Fm., MBM1 and 2, two diamictite horizons of the Meteorite Bore Member, KFm, Koolbye Fm., Ka, Kazput Fm., BRQ, Beasley River Quartzite, CSB, Cheela Springs Basalt

three[20] glacial events (Fig. 1; Supplementary Note 1). The Turee Creek Group consists of a 4-km-thick siliciclastic sedimentary succession conformably overlying the Boolgeeda Iron Formation (Boolgeeda IF) of the Hamersley Group and unconformably overlain by the Beasley River Quartzite and Cheela Springs Basalt of the lower Wyloo Group[21,22]. From bottom to top, the succession consists of clastic sedimentary rocks, minor stromatolitic carbonates and the glaciogenic Meteorite Bore Member (MBM) of the Kungarra Formation, quartzites of the Koolbye Formation, and shales and stromatolitic carbonates of the Kazput Formation[23,24] (Figs. 1 and 2; Supplementary Fig. 1). Metamorphism of the group did not exceed 300 °C[25,26]. The time of deposition is constrained by a U-Pb zircon age of 2450 ± 3 Ma on the Woongarra Rhyolite underlying the Boolgeeda IF[27] and a Pb-Pb baddeleyite age of 2208 ± 10 Ma from a sill intruding the Meteorite Bore Member in the Hardey Syncline[28]. Detrital zircon U-Pb ages from quartzite horizons of the Turee Creek Group have led Krapez et al.[20] to suggest that the Turee Creek basin was closed by about 2430 Ma. However, new U-Pb age constraints from detrital zircons collected at the base of the MBM indicate a maximum age of deposition of ca. 2340 Ma for the diamictites[29], which are therefore not correlative with the >2426 Ma oldest Paleoproterozoic glaciation[30].

Here we study three drill cores that intercept representative stratigraphic sections at the base (Boolgeeda–Kungarra transition, drill core T1), middle (MBM diamictite and underlying Kungarra shales and carbonates, drill core T2), and top (Kazput–Koolbye transition, drill core T3) of the Hamersley–Turee Creek groups (Supplementary Fig. 1; Supplementary Notes 1, 2). We document a new glaciogenic horizon at the top of the Boolgeeda IF, a precise calibration point within the MBM glacial diamictites using Re-Os dating, and multiple sulphur isotope analyses throughout the three drill cores using bulk extraction and in situ ion microprobe techniques (see Methods). We show the persistence of a monotonous small-magnitude MIF-S signal in sedimentary sulphides from 2.45 to beyond 2.31 Gyr, which is punctuated by short episodes of sulphur isotope perturbations attributed to oxidative weathering of the Archaean continental surface. This is inconsistent with an abrupt, globally synchronous, atmospheric transition at about 2.32–2.33 Ga[6–8]. Rather, it is consistent with the view of the GOE as a transitional period[10,31], where an early (younger than 2.45 Ga) rise of atmospheric oxygen above the threshold value of 10⁻⁵ PAL was followed by a long period of continued delivery of MIF-S anomalies to the oceans until their exhaustion from weathering catchments. In the case of the Turee Creek sedimentary basin, the source MIF-S rocks were not

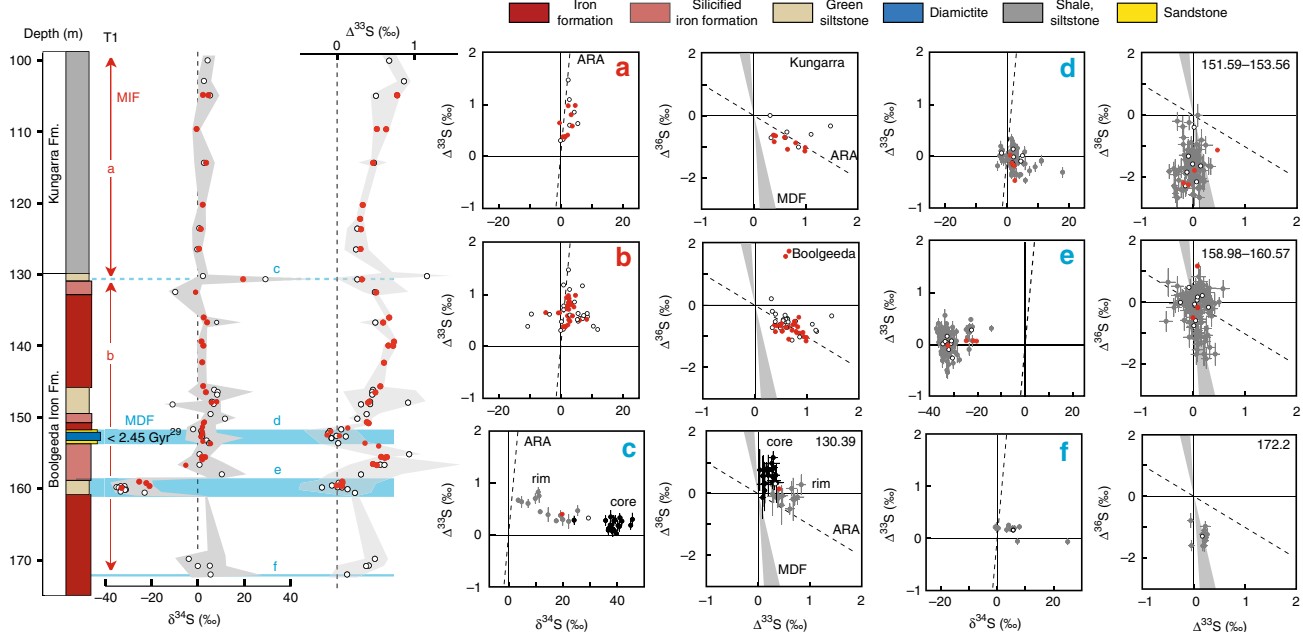

**Fig. 3** $\delta^{34}S$ and $\Delta^{33}S$ profiles of sulphides in drill core T1 and associated $\delta^{34}S$-$\Delta^{33}S$ and $\Delta^{33}S$-$\Delta^{36}S$ compositional diagrams. **a**, **b** Samples of the Boolgeeda and Kungarra Formations characterized by a limited range of $\delta^{34}S$ values, small $^{33}S$-anomalies and plotting on the Archaean reference array (ARA) with a slope of approximately −1 in a $\Delta^{33}S$-$\Delta^{36}S$ diagram. **c**–**f** Sedimentary intervals (151.59–153.56 and 158.98–160.57) or individual samples (130.39 and 172.2) showing variable $\delta^{34}S$ values, near-zero $\Delta^{33}S$ values and plotting on the MDF $\Delta^{33}S$-$\Delta^{36}S$ array of slope approximately −7 (light blue areas labelled **c**–**f** on the $\delta^{34}S$ and $\Delta^{33}S$ profiles). Red dots, bulk analysis; white dots, mean value of in situ analysis for each sample; grey dots, individual spot analysis using the SHRIMP-SI (error bars are standard deviation of $1\sigma$, see Methods). Black and grey dots in **c** correspond to core and rim S-isotope composition, respectively (see text). The grey shaded areas in the $\delta^{34}S$ and $\Delta^{33}S$ profiles represent the range of individual spot analysis for each sample. The stratigraphic column below 130 m depth (Boolgeeda IF) corresponds to the domain (i) discussed in the text and shown in Fig. 2

weathered out by 2.31 Ga as indicated by a persistent MIF-S signal in Kazput carbonates, which are stratigraphically located ~1000 m above the 2.31 Ga old MBM diamictite.

## Results and Discussion

**Age of the Meteorite Bore Member diamictite.** Re-Os dating was performed on glacial diamictites (bulk mudstone matrix and early diagenetic pyrite separates; Supplementary Figs. 2, 3, 4, 5, 6, 7, 8; Supplementary Notes 2, 3; Supplementary Data 1) sampled at the base of the MBM diamictite (the rationale for choosing the MBM diamictite for Re-Os dating is discussed in the Supplementary Note 2). These samples yield an isochron age of 2312.7 ± 5.6 Ma (Supplementary Table 1 and Supplementary Fig. 9a), the first directly obtained from Paleoproterozoic glaciogenic sediments. This date implies that the MBM diamictites were deposited ~140 Myr after the Boolgeeda IF and that the newly identified diamictite within the upper part of the Boolgeeda IF cannot be correlated with the MBM (see below; Figs. 1 and 2)[32,33]. Rather, this lowermost of the three Turee Creek diamictites must represent a much older event, as advocated by Swanner et al. (2013)[34] and recently confirmed by a zircon U-Pb age of 2.454 ± 23 Ga[29]. Considering the 300 m maximum thickness of the Boolgeeda IF and its inferred deposition rate of 225 m/Myr[27], the underlying Woongarra Rhyolite U-Pb age constrains the Boolgeeda glacial event to ca. 2.45 Ga ago.

**Multiple sulphur isotope systematics.** The pyrite analysed by bulk extraction techniques and those analysed in situ using an ion microprobe (SHRIMP-SI) show similar sulphur isotope compositions, with significant shifts occurring in all three drill cores (Figs. 2–5; Supplementary Data 2–4). On the basis of temporal trends in $\delta^{34}S$ and $\Delta^{33}S$, and the relationships between $\Delta^{33}S$ and

$\Delta^{36}S$, we distinguish three domains of distinct sulphide S-isotope systematics. The first domain consists of strongly variable $\delta^{34}S$ sulphides showing either no ($\Delta^{33}S = 0$‰), or slightly positive or negative $\Delta^{33}S$ values in the interval below 130 m depth in T1 (i.e., Boolgeeda IF) (Figs. 2 and 3). The second domain comprises $^{34}S$-depleted sulphides with small positive $^{33}S$-anomalies that define a $\Delta^{36}S/\Delta^{33}S$ trend of slope approximately −7 characteristic of MDF processes[35] and with variable sulphur contents occurring in the interval below 145 m depth in T3 (Figs. 2 and 4). The third domain concerns sulphides with monotonous isotopic signals of small amplitude ($\delta^{34}S_{bulk} = 4.46 ± 2.31$‰, $\delta^{34}S_{SIMS} = 3.80 ± 3.10$‰; $\Delta^{33}S_{bulk} = 0.88 ± 0.27$‰, $\Delta^{33}S_{SIMS} = 0.82 ± 0.15$‰; $\Delta^{36}S_{bulk} = −1.01 ± 0.44$‰, $\Delta^{36}S_{SIMS} = −0.82 ± 0.31$‰) and a $\Delta^{36}S$-$\Delta^{33}S$ trend overlapping the Archaean Reference Array (ARA) of slope approximately −1[1,36] in the remaining ~370 m of cores comprising the Kungarra Formation and MBM diamictites (T2 and top of T1) and the Kazput Formation carbonates above 145 m depth in T3 (Figs. 2, 3 and 5).

The interval below 130 m depth in T1 appears to record an important oxidative event near the Archaean/Proterozoic boundary (Fig. 3). Specifically, a large–magnitude negative shift in $\delta^{34}S$ occurs between 159.98 and 160.57 m depth (Fig. 3e) that is accompanied by the highest sulphur content observed in our core samples (Fig. 2). Pyrite in this 1.5-m-thick green siltstone interval (Supplementary Figs. 5 and 6) has $\delta^{34}S$ values as low as −35‰ ($\delta^{34}S_{bulk} = −24.98 ± 5.38$‰, $\delta^{34}S_{SIMS} = −31.48 ± 3.33$‰), has $\Delta^{33}S$ ~0‰ ($\Delta^{33}S_{bulk} = 0.06 ± 0.05$‰, $\Delta^{33}S_{SIMS} = 0.04 ± 0.14$‰), and overlaps the mass-dependent $\Delta^{36}S/\Delta^{33}S$ signature of slope approximately −7. This interval is the oldest that exhibits such low $\delta^{34}S$ values in bulk rock analyses. The absence of MIF-S and the occurrence of strongly $^{34}S$-depleted values indicate that microbial sulphate reduction occurred under non-limiting sulphate conditions, which is consistent with higher inputs of

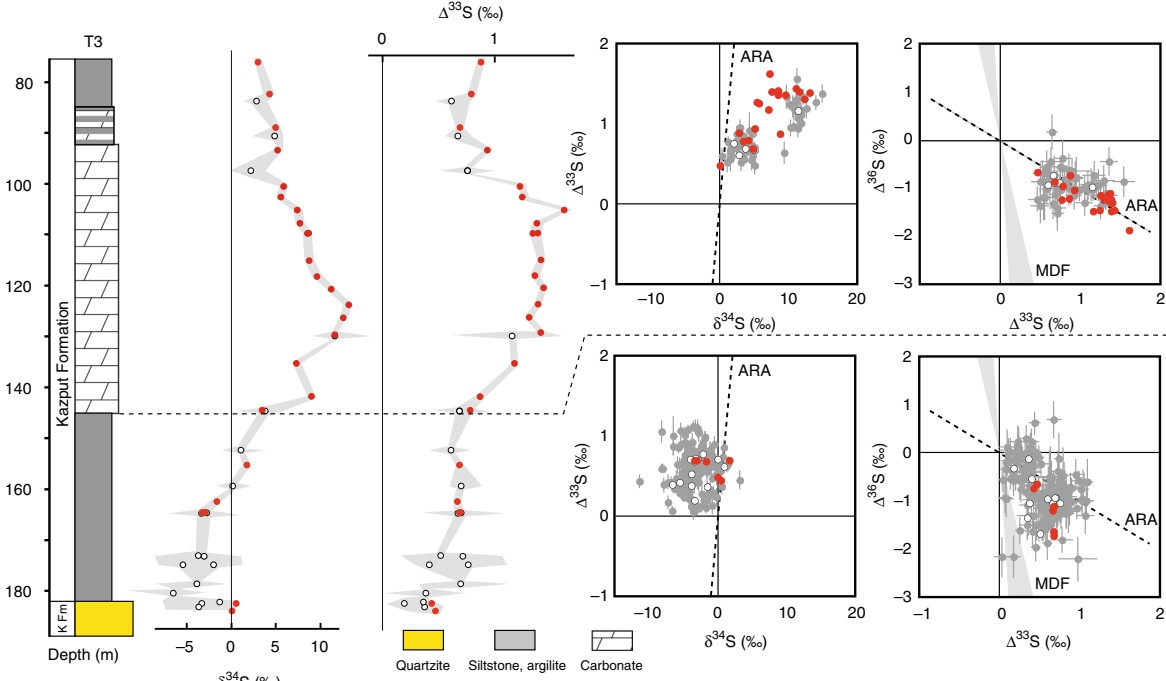

**Fig. 4** $\delta^{34}S$ and $\Delta^{33}S$ profiles of sulphides in drill core T3 and associated $\delta^{34}S$-$\Delta^{33}S$ and $\Delta^{33}S$-$\Delta^{36}S$ compositional diagrams. Red dots, bulk analysis; white dots, mean value of in situ analysis for each sample; grey dots, individual spot analysis using the SHRIMP-SI (error bars are standard deviation of $1\sigma$, see Methods). The grey shaded areas in the $\delta^{34}S$ and $\Delta^{33}S$ profiles represent the range of individual spot analysis for each sample. ARA, Archaean Reference Array, MDF, mass-dependent fractionation of sulphur isotopes, KFm, quartzite of the Koolbye Formation. The stratigraphic column below 145 m depth corresponds to the domain (ii) discussed in the text and shown in Fig. 2

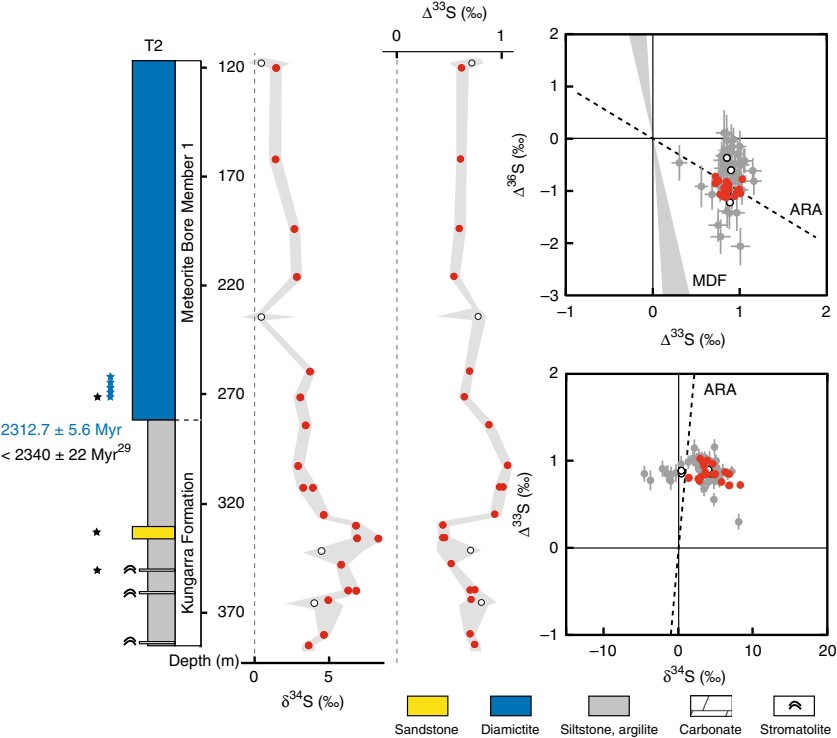

**Fig. 5** $\delta^{34}S$ and $\Delta^{33}S$ profiles of sulphides in drill core T2 and associated $\delta^{34}S$-$\Delta^{33}S$ and $\Delta^{33}S$-$\Delta^{36}S$ compositional diagrams. Red dots, bulk analysis; white dots, mean value of in situ analysis for each sample; grey dots, individual point analysis using the SHRIMP-SI (error bars are standard deviation of $1\sigma$, see Methods). The grey shaded areas in the $\delta^{34}S$ and $\Delta^{33}S$ profiles represent the range of individual point analysis for each sample. ARA, Archaean Reference Array, MDF, mass-dependent fractionation of sulphur isotopes. Sulphide Re-Os date of 2312.7 ± 5.6 Ma is from this study. Zircon U-Pb date of 2340 ± 22 Ma is from Caquineau et al.[29]. The black and blue stars adjacent to the lithologic column correspond to the samples used for Re-Os (blue) and U-Pb (black) dating

sulphate due to oxidative weathering. A 2-m-thick layer of clastic and glaciogenic sediments between 151.6 and 153.4 m depth (i.e., Boolgeeda diamictite, Figs. 2 and 3d; Supplementary Fig. 2) also has high sulphur content and shows the same minimal deviation from mass dependent fractionation sulphur isotope signal ($\Delta^{33}S_{bulk} = 0.05 \pm 0.28‰$, $\Delta^{33}S_{SIMS} = -0.02 \pm 0.11‰$; MDF $\Delta^{36}S/\Delta^{33}S$) but near-zero $\delta^{34}S$ values ($\delta^{34}S_{bulk} = 1.88 \pm 0.48‰$, $\delta^{34}S_{SIMS} = 2.19 \pm 2.20‰$). However, several in situ analyses show positive $\delta^{34}S$ values up to 18‰. This, together with the mass-dependent $\Delta^{36}S/\Delta^{33}S$ slope, argue for quantitative consumption of a limited sulphate reservoir, limited availability of organic matter, or both[35,37,38]. Alternatively, the high sulphur content, $\delta^{34}S$ variability, MDF $\Delta^{36}S/\Delta^{33}S$ slope and sharp transition between MDF vs. MIF-S sedimentary horizons suggest that other factors such as depth of deposition and/or changes in the contributions of atmospheric vs. riverine sulphur fluxes to the oceans could have played an important role during sedimentation of the Boolgeeda IF[37]. Silicified banded IF at the base of core T1 (sample 172.1, Fig. 3f, Supplementary Fig. 5e), which contains microscopic sulphide aggregates, is also devoid of $^{33}S$-anomalies and has near-zero $\delta^{34}S$ values. This suggests that free oxygen was likely present in the atmosphere before deposition of the ~3.5 m of green siltstone and glaciogenic sediments preserving MDF sulphides (Fig. 3d). The presence of free oxygen is supported by the occurrence of near zero $\Delta^{33}S$ and positive $\delta^{34}S$ sulphides of the underlying ~2.48 Ga Whaleback Shale Member of the Brockman Formation[5,39]. The 3.5 m of MDF-S sedimentary rocks contrast with the bulk of the Boolgeeda IF, which contains anomalous sulphides ($\Delta^{33}S_{bulk} = 0.69 \pm 0.14‰$, $\Delta^{33}S_{SIMS} = 0.62 \pm 0.23‰$) with near-zero $\delta^{34}S$ ($\delta^{34}S_{bulk} = 2.54 \pm 2.74‰$, $\delta^{34}S_{SIMS} = 3.63 \pm 6.41‰$) that plot on the ARA with a $\Delta^{36}S/\Delta^{33}S$ slope of approximately $-1$ (Fig. 3b). This limited range of small $^{33}S$-anomalies and $\delta^{34}S$ values in the Boolgeeda IF overlaps the range observed in the third domain of S-isotopic anomalies identified here (iii), comprising the overlying Kungarra Formation (T2 and top of T1; Figs. 3a and 5) and carbonate of the Kazput Formation (top of T3, Fig. 4), accompanied by a limited range in sulphur content (Fig. 2). This homogeneity in isotope composition throughout ~370 m of core collected over a 3000 m thick sedimentary succession is remarkable considering the different lithologies (terrigenous, glaciogenic and chemical) and their depths of deposition.

**Atmospheric vs continental sulphur reservoirs.** The persistence of monotonous small-magnitude MIF-S signals in Turee Creek Group sedimentary sulphides, punctuated by sulphur isotope perturbations coupled to sulphur enrichments, specifically imply oxidative weathering of the Archaean continental surface. Sulphate produced this way would retain the mass-independent composition of the weathered Archaean sulphides ($\Delta^{33}S$ ~0.5–1‰[9,10]), although some variability would arise as igneous sulphides bearing no MIF-S would also be weathered. Under an oxidizing atmosphere such as in the present, rock weathering should dominate the sulphate flux to the oceans and atmospheric contributions are minimal[40]. In this scenario, changes in the materials being weathered (the source) and in the final depositional environment (the sink) would be reflected in the variability of $\delta^{34}S$ and $\Delta^{33}S$ values of the newly produced sulphur minerals and in sedimentary S concentrations. For example, the two green siltstone and diamictite horizons in the Boolgeeda IF that display high sulphur contents (0.12–4.7 weight %) and $\Delta^{33}S$ ~0‰ are bounded above and below by BIFs with low sulphur content (<1000 ppm) and small positive to near-zero $\Delta^{33}S$. Considering that the BIFs represent deep water deposits with minimal influence from terrigenous inputs, such abrupt change in sulphur

contents and isotope compositions is best explained by a continuous background sulphur source with small positive to near-zero $\Delta^{33}S$ anomalies, as linked to oxidative continental weathering and recorded in the BIFs, and higher rates of sulphate reduction which would enrich sulphur contents (by 1 to several weight %) and result in $\Delta^{33}S = 0$ ‰ recorded in the green siltstone and diamictite horizons. The occurrence of strongly $^{34}S$-depleted sulphides in green siltstones and near zero $\delta^{34}S$ sulphides in the diamictite can be explained by a greater availability of organic matter in the siltstones, which would enhance sulphate reduction. Furthermore, high primary productivity in the Paleoproteroic ocean may have occurred during deposition of the green siltstones, which would lead to temporary euxinic conditions[5], whereas low primary productivity during glaciation would have hampered microbial sulphate reduction.

Support for fluctuations in sulphur sources and sulphate reduction activity during the formation of sedimentary pyrite is also observed on a grain scale. Sample 130.39 at the transition between the Boolgeeda IF and the Kungarra shales (Fig. 3c, Supplementary Fig. 5a) contains $^{34}S$-enriched pyrites ($\delta^{34}S_{core} = 39.86 \pm 5.83‰$) with slightly positive $^{33}S$-anomalies ($\Delta^{33}S_{core} = 0.2 \pm 0.1‰$), which are overgrown by euhedral rims with lower $\delta^{34}S$ values ($\delta^{34}S_{rim} = 13.63 \pm 6.77‰$) but higher $\Delta^{33}S$ ($\Delta^{33}S_{rim} = 0.52 \pm 0.19‰$). The core composition with positive $\Delta^{36}S$ values overlapping the MDF $\Delta^{36}S/\Delta^{33}S$ trend argues for consumption by sulphate reducers of a limited[14,32,33], slightly anomalous, sulphate reservoir. The rim composition forms a linear array trending toward the composition of the sulphides deposited immediately above and below sample 130.39 (Fig. 3a, b). This argues for mixing during diagenesis with a MIF-bearing sulphur reservoir reflecting the ambient water column composition. Similarly, sulphides in the interval below 145 m depth in T3 (domain (ii) of Fig. 2) show the same small positive $\Delta^{33}S$ as the overlying Kazput carbonate and underlying Kungarra Formation ($\Delta^{33}S_{bulk} = 0.59 \pm 0.13‰$, $\Delta^{33}S_{SIMS} = 0.58 \pm 0.18‰$), display small negative $\delta^{34}S$ ($\delta^{34}S_{bulk} = -0.58 \pm 1.95‰$, $\delta^{34}S_{SIMS} = -2.78 \pm 1.76‰$) and, most significantly, overlap the MDF $\Delta^{36}S/\Delta^{33}S$ slope (Fig. 4), again suggesting a mass-dependent sulphate reduction process thriving on an anomalous ($\Delta^{33}S$ ~0.5‰) sulphate source. These characteristics indicate sufficient $O_2$ for weathering of a MIF-S continental sulphide reservoir, coupled to varying degrees of sulphate reduction, during deposition of these sediments. Alternatively, one may argue that MIF-S in Paleoproterozoic sediments, such as in the Turee Creek Group, directly records the results of photochemical processes operating under anoxic conditions, such as occurred during the Archaean. However, this interpretation would require an atmosphere composition that could account for persistently low amplitude $\Delta^{33}S$ values. Furthermore, it would also fail to explain the $\Delta^{36}S/\Delta^{33}S$ MDF slopes of $-7$ and globally asynchronous disappearance of MIF-S in the Paleoproterozoic rock record. The most parsimonious explanation for our observations from Western Australia is that since ca. 2.45 Ga, sulphur isotope systematics in sedimentary sulphides reflect oxidative weathering of Archaean sulphides on the continental surface that delivered sulphate bearing MIF-S, the so-called "memory effect" proposed by Reinhard et al.[10].

**Rethinking the definition of the Great Oxidation Event.** The above observations imply sufficient atmospheric oxygen existed to attenuate atmospheric MIF-S production and drive oxidative weathering on land by 2.45 Ga, and are at odds with current chronologies for the GOE. The transition from MIF-S to MDF-S in the upper Rooihoogte and Duitschland formations in South Africa, which occurred before 2.316 Ga (Fig. 1), has been interpreted as the marker horizon for the end of the GOE[6,7]. However,

this interval is bounded above and below by two major dis-conformities spanning several tens of millions of years[17,18]. It was also recently suggested that the GOE concluded even earlier in South Africa, at 2.33 Ga[8]. Similarly, in North America, a U-Pb age of 2308 ± 8 Ma from the Gordon Lake Formation[17], which is close to the unconformity at the top of the Huronian Supergroup and intruded by the 2.22 Ga Nipissing dike swarm[41], indicates that up to 100 Myr of geologic record is missing. Furthermore, although the Espanola Formation records a large range of MDF-S that was used to constrain the time of the GOE in the Huronian Super-group, several workers identified earlier sedimentary intervals above (Pecors Formation) and beneath (McKim Formation) the glacial diamictite of the Ramsay Lake Formation with strongly attenuated $\Delta^{33}S$ and $\Delta^{33}S/\Delta^{36}S$ systematics that were attributed to increases in $pO_2$ between $10^{-5}$ and $10^{-2}$ PAL before and after the first Huronian glacial interval[14–16]. These indications of increases in atmospheric oxygen at the base of the Huronian succession are consistent with our data.

We propose a new age-calibrated correlation combining the Australian data presented here with data from South Africa and North America (Fig. 1), in which: (i) a rise in atmospheric oxygen occurred around 2.45 Ga ago or earlier[4,5,39], before the first Huronian glacial event (Ramsay Lake Formation in North America and the base of the Duitschland Formation in South Africa); and (ii) the two glacial horizons forming the MBM are 2.31 Ga in age or slightly younger and can be temporally linked within error to the Rooihoogte Formation (2310 ± 9 Ma)[17] and possibly the Gowganda Formation, which is younger than 2308 ± 8 Ma[17]. Our new geochronology and S-isotope data extend the sedimentary record of MIF–S to younger than 2.31 Ga, and clearly establish the MIF-S signal as asynchronous between the Pilbara and Kaapvaal cratons. This result is inconsistent with a purely atmospheric signal that should be globally synchronous; rather, it is consistent with the model of Reinhard et al.[10], which shows that atmospheric oxygenation should be followed by a ~100 Myr long "memory effect" where MIF-S anomalies would have continued to be delivered from weathering catchments to the oceans until exhaustion of anomalous sulphur sources at Earth's surface. The asynchronous disappearance of MIF-S due to weathering is highly plausible, as significant differences in the S-isotope composition of catchments have been shown to exist at the continental scale[42]. An important test of the MIF-S memory effect hypothesis would be whether changes in catchment weathering processes or in sediment provenance might be correlated with shifts in the weak MIF-S signals preserved throughout most of the Turee Creek cores. This is expected considering the short residence of oceanic sulphate in a low sulphate ocean. In any case, the GOE as it was previously defined, by the loss of MIF-S at 2.32 Ga[6] or 2.33 Ga[8] in South African successions, is clearly not applicable to the sedimentary record from Australia. Furthermore, the occurrence of MIF-S memory effects makes it impossible to define the GOE using the terminal MIF-S to MDF-S transition that occurred ca. 2.33–2.32 Ga ago in South Africa and North America, and perhaps even younger in Western Australia, depending on the currently unconstrained age of the uppermost Turee Creek Group sediments. The situation becomes much more nuanced, where depressed MIF-S produc-tion in the atmosphere must be decoupled from the rock cycle, and where sedimentary transport, weathering, preservation and diagenesis may exert greater regional and local control over the trajectory of sedimentary MIF-S signals than global atmospheric oxygenation state. Given such constraints, it might be more logical to define the GOE as the moment that sedimentary MIF-S signals fall below a certain threshold that simple sedimentary recycling cannot reproduce. Our data indicate that this occurred before 2.45 Ga, and arguably, it occurred between 2.50 and 2.48

Ga when sulphur isotope systematics captured the first evidence for changes in the surface atmospheric $O_2$ budget[4,5,39].

## Methods

**Re-Os geochronology.** Five samples of diamictite from the Meteorite Bore Member (T2-252.55, T2-259.3, T2-264.3, T2-271.1 and T2-272.46) and two sets of pyrite separates (two different fractions of sample T2-272.46) were used for Re-Os geochronology (Supplementary Table 1). The reasons for choosing the Meteorite Bore Member diamictites for Re-Os dating are given in the Supplementary Note 2. The Re and Os concentrations and the Os isotopic composition were analysed at the John de Laeter Centre for Isotopic Research at Curtin University in Perth (Australia) using the Carius tube digestion method[43]. Bulk-rock diamictite (matrix only devoid of boulders) were digested using $CrO_3$–$H_2SO_4$[44]. Approximately 3 g of whole-rock sample powder for diamictite was digested in 16 ml of $CrO_3$–$H_2SO_4$ mixture as described by Selby and Creaser[44] to extract preferentially hydrogenous Re and Os. For pyrite, 80–100 mg of pure pyrite separate was digested using inverted *aqua regia* (1 ml of purged double distilled $HNO_3$ and 3 ml of triple distilled HCl). For both digestion methods, the obtained mixture was chilled and sealed in previously cleaned Pyrex™ borosilicate Carius tubes and heated to 220 °C for 60 h. Osmium was extracted from the acid solution by chloroform solvent extraction[45], then back-extracted into HBr, followed by purification via micro-distillation[46]. Rhenium was separated from a portion of the residual solution using anion exchange chromatography.

The purified Os and Re fractions were loaded onto Pt filaments, and measured using N-TIMS on a ThermoFisher Triton™ mass spectrometer using a secondary electron multiplier detector. The measured isotopic ratios were corrected for mass fractionation using $^{192}Os/^{188}Os = 3.092016$, as well as spike and blank contributions. The internal precision of $^{187}Os/^{188}Os$ ratio measured in all samples was better than 0.63% ($2\sigma$, Supplementary Table 1). To monitor long-term instrument reproducibility, an AB-2 Os standard (University of Alberta) was analysed. The AB-2 Os standard yielded 0.10687±0.00012 ($n = 2$, $2\sigma$) during the period of the measurements, which is consistent with the value reported by Selby and Creaser[44] (0.10683860±0.00004). An in-house Re standard solution gave $^{185}Re/^{187}Re = 0.5987±0008$ ($n = 2$, $2\sigma$). The total procedural blank for Os was 0.02 pg for sulphuric acid–chromium oxide procedure and 0.5 pg for inverted *aqua regia* digestion method. For Re, total procedural blank was 3.5 pg for sulphuric acid–chromium oxide digestion method, and 7 pg for inverted *aqua regia* digestion. The $^{187}Os/^{188}Os$ ratios of the blank was 0.194±0.002 for sulphuric acid–chromium oxide method, and 0.202±0.006 for inverted *aqua regia* digestion method. For whole-rock analyses, the total Os analytical blank represented <0.5% of the total Os. Since total blank for both Re and Os was run as part of each batch of dissolutions, appropriate blank correction was applied to each batch. The Re and Os concentrations in pyrite range from 2.9 to 3.6 ppb and from 109 to 206 ppt, respectively (Supplementary Table 1). The Re and Os concentrations of the diamictite samples range from 3.6 to 4.3 ppb and 418 to 870 ppt, correspondingly.

The Re-Os isochron regressions were performed using Isoplot[47], applying the $^{187}Re$ decay constant of $1.666 \times 10^{-11}$ .year$^{-1}$ [47]. The uncertainties for $^{187}Re/^{188}Os$ and $^{187}Os/^{188}Os$ were determined by error propagation, and the error correlation $\rho$ ('rho') (Kendall et al.[49] and references therein). Error correlation between $^{187}Re/^{188}Os$ and $^{187}Os/^{188}Os$ is usually significant for shales and diamictites[49] (0.35–0.57 for our samples), and is therefore recommended for usage on Re-Os isochron regression diagrams.

The plot of isotope data for the five diamictites and the two pyrite separates on the Re-Os isochron diagram (Supplementary Fig. 9a, Supplementary Table 1) defines a best-fit line with an age of 2312.7 ± 5.6 Ma (MSWD = 1.4) and an initial $^{187}Os/^{188}Os$ ratio of 0.151±0.005. The five diamictite samples displayed a similar age of 2316±12 Ma (Supplementary Fig. 9b), with a well-defined initial ratio of 0.149±0.008.

**Pyrite trace element composition.** Trace element abundances of pyrites were analysed by the LA-ICP-MS at the Australian National University. The system consists of an ArF (193 nm wavelength) excimer laser, which is interfaced to a custom-built ablation chamber (HelEx) coupled to an Agilent 7700 quadrupole ICP-MS. Ablation was performed in a He-Ar atmosphere with a pulse energy of ~80 mJ and pulse repetition rate of 5 Hz. Laser beam sizes ranging from 28 to 47 μm were employed. The acquisition time per spectrum was set to 90 s, which comprises 30 s for counting the background and 60 s for counting the ablation signal. Before and after analysis of ~10 unknowns, reference materials were mea-sured for external calibration and to perform instrumental drift corrections. USGS MASS-1 and STDGL-1[50] were used as primary reference materials, and RTS-3, RTS-4 and GXR-1 were run as secondary reference materials. The data were processed using Iolite and $^{57}Fe$ as the internal standard. $^{29}Si$ and/or $^{43}Ca$ were used to estimate the contribution of matrix in each analysis. LA-ICP-MS trace element abundances are reported and summarized in Supplementary Data 1 and Supple-mentary Figs. 7 and 8.

**Pyrite sulphur isotope composition.** A total of 64 and 71 samples were selected for in situ (SIMS) and bulk (fluorination) S-isotope analysis, respectively. Among these, 44 (SIMS) and 32 (bulk) samples were selected in T1, 4 (SIMS) and 20 (bulk)

samples in T2, and 16 (SIMS) and 19 (bulk) samples in T3. A total of 892 analyses were performed by in situ technique (600 analyses in T1, 50 analyses in T2 and 240 analyses in T3). In addition, 42 samples used for in situ S-isotope analysis have been selected for trace element analysis using LA-ICP-MS technique based on pyrite texture, lithology, and stratigraphic unit. Finally, 280 major and trace elements analyses were obtained on bulk powders from the Activation Laboratories (Actlabs) in Ancaster, Ontario Canada. Sulphur concentrations are shown in Supplementary Data 2 and Fig. 2.

**Bulk sulphur isotope analysis.** Approximately 1–3 g of powdered sample were transferred into a 50 ml boiling flask attached to a distillation apparatus similar to that described by Forrest and Newman[51]. Reduced sulphur—consisting primarily of pyrite and, if any acid-volatile of elemental sulphur—was converted to $H_2S$ using a sub-boiling $Cr^{2+}$–HCl mixture[52], transferred using high purity nitrogen and bubbled in a containing silver nitrate trap (0.2 M). The precipitated silver sulphide ($Ag_2S$) was subsequently rinsed and dried before being wrapped in aluminium foil and loaded into a nickel bomb for fluorination with excess $F_2$ (10–100-fold) at 250 °C overnight. Product $SF_6$ was separated from other products using cryogenic and gas chromatography techniques using protocol similar to most laboratories worldwide[53]. Purified $SF_6$ was then quantified and analysed using a dual inlet ThermoFinnigan MAT 253 where $m/z = 127^+$, $128^+$, $129^+$ and $131^+$ ion beams are monitored. The $\delta^{34}S$, $\Delta^{33}S$ and $\Delta^{36}S$ values are presented in the standard delta notation against V-CDT with an analytical reproducibility of ≤0.2‰, 0.02‰ and 0.2‰ (2$\sigma$), respectively (Supplementary Data 3 and Figs. 2–5).

**Ion microprobe analysis of sulphur isotopes.** Sulphur isotope ($^{32}S$, $^{33}S$, $^{34}S$ and $^{36}S$) compositions of sedimentary sulphides (mostly pyrite, $FeS_2$, and in a few rare cases pyrrhotite, FeS) were measured in situ with the SHRIMP-SI ion microprobe at the Research School of Earth Sciences, the Australian National University (ANU; Supplementary Data 4 and Figs. 2–5). Samples were analysed over a 15-month period, between December 2013 and March 2015. A total of 892 spots were measured. All samples were mapped with an automated Leica D6000 microscope in reflected light. Sedimentary sulphides were identified and analysed for their trace metal contents (e.g. Co, Zn and Ni) using the optical and scanning electron microscopy (SEM JEOL JSM-6610A) and energy dispersive X-ray spectroscopy (EDS). Results show that the vast majority of sulphides present in the three drill cores are pyrite containing minor content (less than 1% in weight) of transition metals (Supplementary Fig. 10). Other sulphides include petlandite, chalcopyrite and galena. These were easily identified in reflected light under the microscope and discarded for in situ S isotope analysis. Nevertheless, these sulphides represent a minor component of the different samples investigated as attested by the good overlap between the bulk S-isotope analysis and the mean value of the in situ analysis for each sample (Figs. 2–5). In three samples from drill core T2 (178.375, 341.28 and 365.23) we analysed pyrrhotite (Supplementary Data 4). The reported sulphur isotopic ratios do not account for possible systematic errors in the instrumental mass fractionation (IMF) correction due to the compositional difference between the measured pyrrhotite grains and the reference materials (Ruttan, Balmat, and Maine pyrites). However, by assuming that instrumental bias due to matrix effects is mass dependent, $\Delta^{33}S$ and $\Delta^{36}S$ values should be insensitive to matrix effect corrections. Indeed, the reported $\Delta^{33}S$ and $\Delta^{36}S$ values of pyrrhotites were similar to those of adjacent pyrite grains. Strictly speaking, however, a standard of similar composition should have been used to calculate the S-isotope compositions of those grains.

SHRIMP sulphur isotope measurements were performed with a $Cs^+$ primary beam of ~2 nA. On SHRIMP-SI, caesium ions are generated in a Kimball Physics IGS-4 alkali metal ion gun by heating a Cs zeolite cartridge and focused through an accelerating potential of 5 kV. The sample potential is held at ca. −10 kV resulting in a total primary ion impact energy of 15 keV at the target. For this work, we used Kohler illumination produced by demagnification of a 200 μm Kohler aperture by the final condensor lens operated as an immersion lens producing a ca. 20 × 30 μm pit diameter on the Au-coated target.

Negative secondary ions are accelerated to real ground from the −10 kV sample potential and focused by the quadrupole triplet lenses before passing through the source slit and entering the secondary mass analyser. Source slit was set at 60 μm. The SHRIMP-SI multiple collector was configured to allow simultaneous measurement of $^{32}S^-$, $^{33}S^-$, $^{34}S^-$ and $^{36}S^-$. Secondary ion signals were collected in Faraday cups with $^{32}S^-$, $^{33}S^-$ and $^{34}S^-$ ion currents measured on iFlex electrometers with high-ohmic ($10^{10}$–$10^{12}$ Ω) resistors. $^{32}S^-$ was collected on a $10^{11}$ Ω resistor (50 V range), $^{33}S^-$ and $^{34}S^-$ on $10^{11}$ Ω resistors (5 V range). For measurement of $^{36}S^-$, the iFlex electrometer was operated in charge mode where the feedback resistor is replaced by a 22 pF capacitor[54]. The iFlex electrometers are operated under vacuum with internal temperature control, to deliver optimum performance. Collector slit widths were set at 400 μm for $^{32}S^-$, 150 μm for $^{33}S^-$, 200 μm for $^{34}S^-$ and 300 μm for $^{36}S^-$. Potential isobaric interferences on $^{33}S^-$ and $^{34}S^-$ from $^{32}SH^-$ and $^{33}SH^-$, respectively, were well resolved.

Under the operating conditions described above, the $Cs^+$ primary beam removes the ca. 20 nm Au coating at the analytical spot within a few seconds. In sulphides, the secondary ion current rises steadily for about 4 min, after which the rise slows. During the rapid rise, the $^{33}S^-/^{32}S^-$, $^{34}S^-/^{32}S^-$, and $^{36}S^-/^{32}S^-$ also rise. Data collection commences once the signal has stabilized, normally after 4–5 min.

Each spot analysis took between 15 and 25 min and consisted of 300 s of presputtering, ~100 s of automated steering of secondary ions (i.e., adjustments of the secondary ion beam path in the Y and Z directions within the secondary column, prior to the electrostatic analyser, so as to maximize secondary ion count rates), ~5 s of automated centering of the secondary ions in the collector slits with magnet control, and 400, 600, 800 or 1000 s of data collection (depending on the session). Data acquisition consisted of 2, 3, 4 or 5 sets of 10 measurements (20 s each) with re-optimization of the secondary beam centering in the collector slits carried out between sets.

Typical count rates on $^{32}S^-$, $^{33}S^-$, $^{34}S^-$ and $^{36}S^-$ were about 1–2 × $10^9$, 1–1.5 × $10^7$, 5–9 × $10^7$ and 2–3 × $10^5$ cps, respectively. Background count rates for $^{32-}$ were about 1.5 × $10^5$ cps and for $^{33}S^-$, $^{34}S^-$, and $^{36}S^-$ ca. 2.5–3 × $10^4$ cps. The noise level of the Faraday cup electrometers for the entirety of an analytical session was <2000 cps (1 SD for 20 s integration) for the $10^{11}$ Ω resistors and <300 cps (1 SD for 20 s integration) for the 22 pF capacitor. Background count rates were monitored during the 300 s of presputtering of each individual spot measured.

Analyses of unknown pyrites were bracketed by measurements of pyrite reference materials, Ruttan, Balmat, and Maine. Isotopic ratios were normalized to Ruttan pyrite[55] ($\delta^{34}S = + 1.2 ± 0.2‰$), except for sessions run on December 2013 where Maine pyrite[56] ($\delta^{34}S = –19.96 ± 0.62‰$) was used as the primary reference material. Ruttan pyrite is assumed to have non-mass independent fractionation $\Delta^{33}S$- and $\Delta^{36}S$-values based on their post-Archaean age. Balmat pyrite[55] ($\delta^{34}S = + 15.1 ± 0.2‰$) was also routinely measured. No detectable $\Delta^{33}S$ and $\Delta^{36}S$ mass-independent fractionation signature have been reported for Balmat pyrite[33,57].

Sulphur isotopic ratios have been corrected for instrumental mass fractionation by applying a correction factor calculated from the analyses of the pyrite reference material with known sulphur isotopic ratios ($^{33}S/^{32}S$, $^{34}S/^{32}S$ and $^{36}S/^{32}S$) (Supplementary Data 4). Background counts are subtracted from the total counts prior to ratio calculation. The corrected $^{33}S/^{32}S$, $^{34}S/^{32}S$ and $^{36}S/^{32}S$ ratios are then expressed in delta notation such that ratios are given as permil deviations (‰) from the reference value of the pyrite reference material (Ruttan or Maine) on the V-CDT scale,

$$\delta^{33}S_{V-CDT} = 1000 \times \left[ \frac{(^{33}S/^{32}S)_{corrected}}{(^{33}S/^{32}S)_{reference}} - 1 \right] (‰) \tag{1}$$

$$\delta^{34}S_{V-CDT} = 1000 \times \left[ \frac{(^{34}S/^{32}S)_{corrected}}{(^{34}S/^{32}S)_{reference}} - 1 \right] (‰) \tag{2}$$

$$\delta^{36}S_{V-CDT} = 1000 \times \left[ \frac{(^{36}S/^{32}S)_{corrected}}{(^{36}S/^{32}S)_{reference}} - 1 \right] (‰) \tag{3}$$

Deviations from the Terrestrial Fractionation Line (TFL) have been calculated following Farquhar et al.[1], where

$$\Delta^{33}S = \delta^{33}S - 1000 \times \left( \left( 1 + \frac{\delta^{34}S}{1000} \right)^{0.515} - 1 \right) \tag{4}$$

$$\Delta^{36}S = \delta^{36}S - 1000 \times \left( \left( 1 + \frac{\delta^{34}S}{1000} \right)^{1.90} - 1 \right) \tag{5}$$

The uncertainty on an individual analysis (also defined as internal error) is calculated as the standard deviation of the mean (=standard error, SE) of a set of isotope ratios (2, 3, 4 or 5 sets of 10 measurements each) measured during the course of a single spot analysis. All single-spot errors here reported are quoted at 95% confidence limits ($\sigma_{95\%}$), which corresponds to multiplying the standard error by a two-sided 95% confidence distribution). The internal precision of $\delta^{34}S$, $\Delta^{33}S$ and $\Delta^{36}S$ of individual measurements was usually better than 0.03‰, 0.10‰ and 0.40‰ ($\sigma_{95\%}$), respectively.

The primary normalizing reference material (Ruttan or Maine pyrite) was assessed for analytical reproducibility of the sulphur isotope analyses ($\delta^{34}S$, $\Delta^{33}S$ and $\Delta^{36}S$). At the SHRIMP laboratory at ANU, we adopt the protocol that reproducibility is calculated as the standard deviation (SD) of the entirety of analyses on the reference material over the course of an analytical session. The reproducibility of $\delta^{34}S$, $\Delta^{33}S$ and $\Delta^{36}S$ was generally better than 0.20‰, 0.15‰ and 0.25‰ (1 SD), respectively.

**Data availability**. All data generated or analysed during this study are included in this published article (and its supplementary information files).

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

## Acknowledgements

We thank the Institut de Physique du Globe de Paris, The University of New South Wales and the Geological Survey of Western Australia for supporting the Turee Creek Drilling Project. This work was supported by grants from the UnivEarths Labex Programme at Sorbonne Paris Cité (ANR-10-LABX-0023 and ANR-11-IDEX-0005-02), the Programme National de Planétologie and the São Paulo Research Foundation (FAPESP, grant 2015/16235-2). B.A.K. acknowledges support from the European Union Horizon 2020 research and innovation programme under the Marie Sklodowska-Curie grant agreement No. 708117. J.N.A. and T.R.I. acknowledge support from ARC DP140103393.

## Author contributions

P.P. conceived the Turee Creek Drilling Project. P.P., E.M., C.T. and M.J.V.K. performed the drilling. P.P., E.M. and E.P. performed the logging and selected the samples for trace element analysis. P.P. selected the samples for sulphur isotope analysis. T.C. and P.P. selected the samples for Re-Os dating. P.P. and J.N.A. performed the in situ sulphur isotope analysis. J.N.A. performed the LA-ICP-MS analyses of pyrite. F.B. and P.C. performed the bulk sulphur isotope analysis. S.T. and T.C. performed the Re-Os dating. P.P. and B.A.K. wrote the paper. All authors commented on the manuscript.

## Additional information

**Competing interests:** The authors declare no competing interests.

