## [Peer Review File · Nature Communications]

Editorial note: The figure on page 27 in this Peer Review File is reproduced from Precambrian Research, Volume 256, Martin J. Van Kranendonk, Rajat Mazumder, Kosei E. Yamaguchi, Koji Yamada, Minoru Ikehara; Sedimentology of the Paleoproterozoic Kungarra Formation, Turee Creek Group, Western Australia: A conformable record of the transition from early to modern Earth, 314-343, January 2015, with permission from Elsevier; <https://doi.org/10.1016/j.precamres.2014.09.015>

Reviewers' comments:

Reviewer #1 (Remarks to the Author):

Philippot et al. report new S isotope data and a Re-Os age of "diagenetic" pyrite + mudstones from Western Australia. Their new data help further constrain the temporal trends in S isotope signatures, which in turn provide tighter constraints on the timing of atmospheric oxygenation and relict crustal recycling of mass-independent-fractionation of sulfur isotopes (MIF-S) beyond that provided in previous papers. I am enthusiastic about that aspect of the paper. I am not particularly wild about the authors approach for Re-Os geochronology. With appropriate modification of the manuscript, I think that this will turn out to be a good paper for Nature Communications, and think that it will be well-cited paper.

Lines 37-40: Depending on space constraints, for the abstract there needs to be a clear explanation for why, in a single locality/drillcore, there is alternation between essentially zero MIF-S (two intervals in T1) and low-amplitude but non-zero MIF-S. Oxidative continental weathering of sulfide minerals (and recycling of relict MIF-S signatures) should be occurring in both scenarios for the time interval covered in this study. The sentence on lines 41-42 isn't satisfactory as it does not clearly address the above issue.

Line 51: The authors could be more detailed in their coverage on age constraints for the youngest known S-MIF signal and the oldest known non-S-MIF signal, prior to this study. This introductory paragraph is rather vague about this.

Line 81: Are these early or late diagenetic pyrites? Diagenesis and pyrite formation can occur millions to tens of millions of years after initial deposition. Re-Os dates from macroscopic late diagenetic pyrite nodules are not necessarily informative regarding the timing of deposition, whereas synsedimentary pyrite (e.g., formed in the water column) and early diagenetic pyrite would provide close constraints on the timing of sedimentary deposition.

Lines 88-89: Is this diamictite truly a glacial diamictite? If so, what is the petrographic evidence? I cannot find anything on this in the main text or supplementary information. Not all diamictites have a glacial origin. Some diamictites can be formed by marine sediment gravity flows.

Lines 107-108: Will general readers of Nature Communications understand what "anomalous" and "non-anomalous" means for $\Delta^{33}\text{S}$? Only isotope geochemists or Precambrian geoscientists would recognize this. Please explain briefly in the main text so that general readers do not have to look in the supplementary information for it.

Line 129: "minor sulphur isotope signal" is not the best phrase to use here? Refer specifically to a minimal MIF-S signature.

Lines 159-161: This is a rather vague statement. The two main excursions to $\Delta^{33}\text{S} \sim 0\text{‰}$ both occur in T1 – the oldest part of the studied intervals. Why does $\Delta^{33}\text{S}$ go to 0‰ at these two intervals when clearly we have not yet exhausted (via weathering) the reservoir of continental sulphides with non-zero $\Delta^{33}\text{S}$? The subsequent sentences are not entirely satisfactory given that there are also slightly negative $\Delta^{33}\text{S}$ signatures in T1. Please explain these excursions to zero $\Delta^{33}\text{S}$ more clearly.

Lines 185-186: I note that such signals (attenuated MIF-S isotopes) were also observed transiently in older rocks (namely the 2.5 Ga Mt. McRae Shale; Kaufman et al., 2007; Reinhard et al., 2009), and more persistently in the overlying Brockman Iron Formation by Reinhard et al. (2009; Science). I do not see any incorporation of this important data into this paragraph, and it is needed to provide a complete discussion of temporal trends in S isotope signatures in western Australian sections.

Line 189: Some have interpreted the GOE this way, but others look at the GOE as representing a protracted transition rather than an event (e.g., Lyons et al., 2014; Nature). I think the latter is now more accurate given the data from this study and others. As such, the authors should consider using different wording than “currently interpreted as the marker horizon” as that is not how many of us view the GOE. Indeed, the authors later correctly cast the GOE as having multiple “milestones”.

Line 204: Again, mention the Brockman Iron Formation data.

Line 207: This sentence really ought to be in the abstract!

Line 231: Geochronology suggests 2.06 Ga, not 2.10 Ga, for the end of the Lomagundi Event.

Line 241: If I understood this correctly, a single sample on the Re-Os isochron is being defined as a set of mixed pyrites collected from four different mudstone samples in a single formation. That is unorthodox! What is the reason for doing this? State the thickness of the stratigraphic interval encompassed by the samples defining the Re-Os isochron. Again, there must be a description of the pyrite textures. What can be said about local bottom water redox conditions – at least sediment anoxia is needed for the Re-Os geochronometer to work as hoped. Although there is a good MSWD, I note that if at the time of deposition there was a non-zero slope on the isochron diagram defined by mixed whole rock and late diagenetic pyrite analyses, then the age calculated from present-day isotope ratios would not represent the depositional age. Hence, it is important to demonstrate the timing of pyrite formation through textural analysis. There is some general description in the supplementary information about pyrite textures, but it is paramount to know the textures of specific pyrites chosen for Re-Os analysis. Are syn-sedimentary (formed in a euxinic water column) or early diagenetic pyrites used for Re-Os analysis?

Lines 247-248: The authors use a digestion method that is known to liberate detrital Os from shale matrices (which can potentially yield inaccurate ages because of mixed detrital-authigenic Os isotope compositions). Instead they should have used the chromium-sulfuric acid digestion technique on their mudstone samples (inverse aqua regia is fine for pyrites) that minimizes the release of detrital Os into the digested solution (Selby and Creaser, 2003; Rooney et al., 2011; Kendall et al., 2013). This is important considering that the Re and Os concentrations of the samples are quite low (note the total Os content largely reflects ^{187}Os production via ^{187}Re decay over > 2 Gyr; one can calculate the common Os concentration at the time of formation and compare with estimates of the upper crust Os concentration). Perhaps these concerns could be addressed via regression of pyrite analyses alone, but there are too few pyrite analyses. Finally, a low MSWD is NOT a guarantee of age accuracy. There is a very uneven distribution of data points along the regression line – the slope is heavily controlled by a single, radiogenic, pyrite data point. Hence, the precision and accuracy of this age are not particularly robust despite appearing to be geologically reasonable (i.e., the pre-existing age constraints are rather loose, and the Re-Os age could be off by tens of millions of years and still appear geologically reasonable!).

Line 273: What does “influence of Archean seawater” mean???

The authors have a lot of long paragraphs. Shorter paragraphs, each headed by a topic sentence, would be much easier for the general readers of Nature Communications.

Figure 1: The S isotope record should be extended back to 2.5 Ga. Why exclude all the S isotope data for the Mt. McRae Shale and Brockman Iron Formation, etc? Isn't there S isotope data at 2.5 Ga in the S. African sections? Inclusion of this data would really help further illustrate the temporal trends!

Reviewer #2 (Remarks to the Author):

Philippot and colleagues present new S isotope data and a new Re-Os age from diagenetic sulfides from a series of drillcores through a continuous sequence of sedimentary rocks from Western Australia (Turee Creek Group). Both bulk and in-situ S isotope data from sedimentary sulfides are interpreted in the context of the new Re-Os age constraint and previous geochronologic data to suggest a new framework for the 'Great Oxidation Event' (GOE), a colloquial term for the secular oxygenation of Earth's atmosphere during Paleoproterozoic time.

The extensive new S isotope data, both bulk and in-situ, from a unique archive recently unearthed by the Turee Creek Drilling Project represent an important advance in our understanding of the Precambrian Earth system. The new Re-Os age data, together with the S isotope systematics observed in the Kazput Formation (core T3) in particular, on their own represent a critical new piece of the puzzle in attempts to reconstruct the tempo and mode of Earth's early oxygenation. Although there are naturally debates to be had about timing and mechanism in the S isotope data, given the amount of data presented in the manuscript and the somewhat provocative conclusions, I have very little doubt that the basic result here is robust and important. I would thus be happy to see the paper published in Nature Communications following minor revision (subject to editorial discretion).

One suggestion I would make to the authors is to more explicitly describe the origins of the sulfide phases being analyzed. Many of the supplementary images show cubic pyrites that are clearly post-depositional, and there is some discussion of core-rim differences. I think it would be beneficial if the authors can say something brief and forceful in the main text about the presumed origin of the sulfide phases -- as this will for some readers be the first criticism to be leveled at the interpretation offered in the manuscript. It is likely a mixed assemblage, but in particular can the authors be confident that there hasn't been large-scale overprinting by metamorphic fluids? Better to tackle this head-on then leave the reader to ferret it out in the supplement -- some readers won't, and will remain skeptical (with the result that the impact of the paper may become muted).

Reviewer #3 (Remarks to the Author):

Review of the manuscript by Philippot et al. entitled "Re-defining the Great Oxidation Event with the sulphur record"

The manuscript presents a large dataset of multiple S isotope data for the Turee Creek Group in Western Australia that records the early stage of the GOE. The authors draw two major conclusions: 1) GOE started around 2.45 Ga or earlier; 2) from that time on, local recycling of terrestrial sulfides provided a mass-independent signal to sedimentary successions resulting in a heterogeneous landscape of S isotope record after the beginning of the GOE. The first conclusion is not surprising; see Gumsley et al., 2017 for the most recent publication on this topic (although I am not convinced by the evidence presented in this manuscript and where they set up the transition). The second conclusion in my mind is unsubstantiated and it feels that authors want to fit their data to the prevailing model. There are several geological and geochemical reasons to doubt the second conclusion. First, detrital pyrites has been described throughout the Turee Creek Group (see Krapez et al., 2017; also Kranendonk et al., 2015) so pyrites remained stable under surface conditions during deposition of the Turee Creek Group. This interpretation can be counter-argued that it does not exclude a small terrestrial sulfate flux. More importantly, Fig. 1 of their own manuscript challenges their own arguments. Large MIF-S was reported in the Duitschland and Rooihoogte-Lower Timeball Hill formations, which are unequivocally 100 Ma younger than the beginning of the GOE as placed in this manuscript. This large MIF-S signal cannot represent recycling of continental S, as D33S signals are +4 and even +6 permil (Guo et al., 2006; Luo et al., 2016). This is not a signature of recycling as defined in this manuscript and indeed recently published paper by Gumsley et al. (2017) (not referenced in this manuscript) inferred multiple

rises and falls of MIF-S in association with glacial events. This manuscript does not provide an explanation for multiple glaciations by inferring recycling model.

It is important to understand how authors arrived to these conclusions since geological and geochronologic framework is critical for their interpretation. The generally accepted view, recently summarizing by Krapez et al. (2017) is that the Turee Creek Group was deposited in a short-lived foreland basin that closed by ~ 2.43 Ga or shortly after. Authors challenge this view and argue that 3.5 km of section record over 200 Ma of sedimentation, making this basin a unique bearer of a continuous sedimentary record for this time interval with the slowest depositional rate and longest record of deposition in any other basin of this or even any age. This revision, that might not be obvious to unfocused reader, is based on three pieces of evidence. 1) Diamictite within the Boolgeeda Iron Formation of inferred glacial origin; 2) U-Pb detrital zircon age from the glacial diamictite 1.5 km higher in the section; and 3) Re-Os age for pyrites and mudstones sampled over 100 m of section. Evidence for glacial origin of the diamictite in the Boolgeeda Iron Formation is not strong in my mind, especially if it is not anymore correlative to the well-described glacial Meteorite Bore Member. I found only mention of lamina penetration by a limestone, however overlying lamina are not shown on this figure so it could be a deformational feature (see Kranendonk et al., 2015). These diamictites of uncertain origin could be simply debris flows during transition to a foreland basin. Detrital zircon age is not published; it is only mentioned in the conference abstract with no error bars presented for the age (manuscript states that it is 2340 ± 22 Ma with no data presented and 22 Ma errors might indicate potential complications). This is not strong evidence since the recent comprehensive study by Krapez et al. (2017) did not find any zircons younger than ca. 2.43 Ga in a number of samples covering the whole Turee Creek Group and ca. 2.3 Ga grains were only found in the overlying Beasley River Quartzite. I think it is thus premature to rely on this age. The last piece of evidence is the Re-Os date presented in this paper. However, not being a Re-Os specialist, I see several potential issues with this age. First, it is based on samples collected over 100 m thick section in area with dolerite sills. Normally, samples for Re-Os work are collected over 1-2 m of thickness or better along the single bed to avoid natural variations in initial Os isotope ratio of seawater. Second, methods developed to preferentially attack Re and Os associated with organic matter in Selby's and Creaser's labs were not used so detrital Re and Os contribution could have been substantial. Lastly, considering that 2 points have very similar composition (surprisingly mudstone and a composite of pyrites), it is a 4-point isochron, a bare minimum in Re-Os geochronology. Was uncertainty on the decay constant included in calculations? I therefore consider this date to be uncertain at this stage.

Let's turn argument around and accept author's geologic and geochronologic framework for a moment. Using Fig. 1, diamictite in the Boolgeeda IF would be ca. 2.45 Ga in age and correlative to the oldest Huronian glacial. Diamictite in the Meteorite Bore Member (actually two events based on Kranendonk et al., 2015) is younger than 2.31 Ga and would be correlative with the fourth glacial event of the Paleoproterozoic bracketed between ~ 2.26 and 2.22 Ga in South Africa. There are several critical issues here: 1) in an inferred continuous record of the early Paleoproterozoic 2 Huronian glacials – middle and the upper one – are missing between the Boolgeeda and Meteorite Bore Member; 2) one or two glacial events of the Meteorite Bore Member cannot be correlative to the 2.26-2.22 Ga glacial event in South Africa since they are 50 Ma older than it, implying that at present there is no correlative unit worldwide to this glacial diamictite; 3) If these diamictites are correlative to the 2.26-2.22 Ga glacial event (despite this age difference), it would be even more problematic since carbonates above this glacial do not record the Lomagundi excursion, which must have followed shortly after the 2.26-2.22 Ga glacial event. With all these uncertainties screaming from this new correlation, one could clearly see that the geologic and geochronologic framework that authors subscribed does not fit with the global records that were established over the last 2 decades.

Another critical shortcoming of this manuscript is that the nature of studied sulfides was not described in sufficient detail. The study cover over 3.5 km of succession largely lacking organic rich shales, sulfides hosted in these lithologies could have strong contribution of detrital,

hydrothermal, and authigenic signals. Detrital pyrite was described from this succession (see references above), dolerite sills and metamorphism affected this succession so hydrothermal sulfides are indeed expected and indicated by limited petrography described in SOM. In this case, general statements made in SOM on the origin of sulfides are not convincing, grain-by-grain analysis is required. Figures in SOM do not show convincingly authigenic pyrite.

There are many statements in the manuscript that are either wrong or misrepresenting literature. I provided detailed comments below. In summary, considering a number of uncertainties involved in this study, I feel re-definition of the GOE is not yet warranted.

Lines 29-30: GOE is bracketed by loss of MIF-S and end of the Lomagundi Event at ca. 2.1 Ga (see Holland, 2002).

Lines 32-33: What is this statement based on? Check Gumsley et al. (2017) paper in PNAS

Lines 34-36: Spans 230 Ma? Based on what? It is a foreland basin filled with turbidites, do you know any other foreland basin with 2-3 km of turbidites spanning 230 Ma? This is not geologically reasonable.

Lines 36-37: sulphide is not deposited, it is not clastic in origin.

Lines 37-39: Absence of MIF can be explained by several processes so unique interpretation based on S isotope data alone is not possible. Sulfate aerosols would be a minor flux in oxic atmosphere, where oxidative weathering would dominate.

Lines 39-41: How do you know that small MIF-S reflects oxidative weathering? What if it reflects production in the atmosphere?

Line 41-42: Low sulfate between 2.45-2.22 Ga? But there is good evidence for 2.32 Ga sulfate evaporites in the Gordon Lake Formation of the Huronian Supergroup so sulfate was high at 2.32 Ga if not earlier.

Line 45-46: really? Redox chemistry was dominated by CO₂? What about all other reducing and oxidizing radicals and molecules? See Pavlov and Kasting, 2002.

Lines 49-51: I am confused who said that MIF disappeared by 2.2 Ga ago? There are multiple papers arguing that it disappeared before 2.32 Ga. What am I missing here?

Lines 51-54: Sure, but these papers clearly established age of this transition at ca. 2.32 Ga.

Lines 59-62: These papers also constrain global oxygenation to be older than 2.32 Ga.

Lines 67-70: This statement is based on poor understanding of geology. It states that the Turee Creek Group contains continuous record of 230 Ma years. Just take a realistic sedimentation rate and calculate thickness expected for 230 Ma of deposition. Now, think that most of the Turee Creek Group consists of turbidites that have high sedimentation rate. Add to this that it is retro back-arc basin with a short duration on the order of 20-30 Ma. Finally, none of sited references argue for 3 glacial cycles. Reference 23 is not relevant. Reference 21 argues for one glacial cycle. Reference 22 argues for 2 glacial cycles. How did you get 3? Finally, read Krapez et al., 2017 for tectonic setting and duration of the Turee Creek Group. Neglecting published literature is a bad habit!

Line 74: Be specific: most clastic rocks of the Kungurra Formation are turbidites with high sedimentation rate.

Lines 78-79: This is not correct and biased representation of published literature. See Mueller et al., 2005 Geology paper (with comments and reply), Krapez et al., 2016 Precambrian Research, and Krapez et al., 2017 Precambrian Research. It is clear that 2.2 Ga age does not constrain deposition but provides a maximum age, which by 200 Ma older than the depositional age.

Lines 79-80: This reference is an abstract that present no actual data. Alternatively, there is detailed published study of detrital zircon ages from the Turee Creek Group (Krapez et al., 2017, Precambrian Research) that shows no detrital zircon ages younger than ca. 2.43 Ga. Again, you need to reconcile these data, rather than neglecting them.

Lines 80-96: How do you know that these pyrites are diagenetic in age? What do you mean by 'at the base of the MBM diamictite and its underlying sandstone and carbonate stromatolite?' Below all of these lithologies? Rephrase. If pyrites and mudstones are below all these lithologies, this is not the first age obtained from Paleoproterozoic glaciogenic sediments. Unfortunately, this age is not geologically meaningful. It would place the Meteorite Bore Member to be younger than all 3 Huronian glaciations, but much older than the last Paleoproterozoic glacial event constrained between 2.26 and 2.22 Ga in age. Furthermore, depositional rate of 11 m/Myr for ~1500 m of Kungarra turbidites in a foreland basin is not realistic. The Turee Creek Group was deposited in a separate basin from the Hamersley Group so depositional rates for BIFs (deposited in deep-water setting starved of clastic input) are not relevant to depositional rates for turbidites in the foreland basin. What do you mean by 'the newly identified diamictite horizon recognized here within the upper part of the Boolgeeda Iron Formation?' This diamictite horizon was identified by Krapez, 1996; Martin et al., 1999; and co-authors of this manuscript in earlier publications. And, yes, it cannot be correlated with the MBM since it is stratigraphically lower than it. It is as simple as this. Swanner et al. (2013) never implied that this diamictite reflects older glacial event, they simply stated that it is older in age. Depositional rate of 280 m/Myr was not inferred for the Boolgeeda Iron Formation (Trendall et al., 2004), I do not see this number in this paper. How does your age (which most likely does not constrain deposition) 'indicates that the uppermost Kungarra shales, Koolbye quartzites and Kazput carbonates were continuously deposited until about 2.25 to 2.20 Gyr ago?' This section lacks basic geological logic!

Lines 99-105: There is no discussion what lithologies and types of pyrite were analyzed. This is critical since some lithologies are coarse-grained and might contain detrital pyrite as mentioned in Krapez et al. (2017). There are sills and dikes in the Turee Creek Group and it was deposited in the foreland basin – hydrothermal fluids were moving through the sediments. Finally, some fine-grained lithologies might contain authigenic pyrites. Only the latter would be relevant to the atmospheric evolution.

Lines 107-108: Most of sulfides in the Boolgeeda IF as in other iron formations are not primary in origin – water column and sediments had oxidizing redox inconsistent with sulfidic conditions.

Lines 118-119: Near the Archean-Proterozoic boundary? Really, it is more than 50 Ma younger!

Lines 119-124: What is the origin of pyrite in green siltstone? It is not common for organic matter lean samples to have such high S and pyrite content. Is this chert? What was driving S reduction in absence of organic matter? It is not pyrite precipitated from water column since water column was ferruginous and not sulfidic during deposition of the IF.

Lines 125-127: You can also say that there was enough oxygen for oxidative weathering on the continent to deliver sulfate. Why not? Or that large amount of hydrothermally derived sulfate was flushed to the basin. All these are possibilities, why do you prefer one of them?

Lines 131-133: How limited availability of organic matter would result in unfractionated S isotope values? This must be wrong.

Lines 133-137: This is very vague. Actually neither depth nor change is a process.

Lines 137-141: What about well-known process of sulfidization of iron formation by hydrothermal fluids?

Lines 167-169: How do you know that rims formed during diagenesis vs. via circulation of hydrothermal fluids deriving S by dissolution at stratigraphically lower or higher levels?

Lines 176-182: Why it indicates atmospheric oxygen? You have organic matter poor lithologies and these rarely express large MIF-S signal. Why it would imply asynchronous oxygenation of the atmosphere?

Lines 189-190: Reference 41 is not relevant here.

Lines 193-198: D33S/D36S systematics was not discussed in ref. 16 and refs. 17 and 18 are abstracts so they did not present data.

Line 204: The Deutschland Formation is not correlative to the first Huronian glacial event.

Lines 205-206: One cannot correlate an event with ~ 2.31 Ga age to another event younger than 2.26 Ga since they are different by more than 50 Ma in age.

Lines 206-208: This is truly pathetic. Did you read references in your reference list? Luo et al. (2016) argued for loss of MIF at ~ 2.33 Ga in their title! What is asynchronous?

Line 212: There is still anomalous sulphur source at Earth's surface now. You should be more careful in phrasing.

Lines 224-226: You are inconsistent here with your own definition. Luo et al., (2016) showed D33S up to +4 and +6 permil at 2.33 Ga, if so can you define GOE at 2.45 Ga and explain these data by recycling considering that are from deep-water, open-marine setting?

Lines 226-228: This is odd. How do you imagine this change? Positive D33S values always have matching negative D33S values somewhere. Mixing and recycling can result in diluted or smaller-amplitude D33S signal.

Lines 228-230: What data? How do you know that your samples have this age?

Lines 230-232: Reference 46 has nothing to do with the age or global nature of the Lomagundi Event, this reference argues for the lacustrine origin for this anomaly. Reference 47 is 6 years older than the first use of the GOE acronym.

Line 233: What is geologic marker?

Lines 239-242: Typically, samples less than 1-2 m apart are analyzed with the Re-Os method. You analyzed samples 96 m apart, which cannot have the same initial Os ratio. How do you know that you did not obtain errorchron without geologic meaning? Why method developed by Selby and Creaser for organic-rich shales was not used? This method specifically attacks authigenic Re and Os. In your case, you have also contribution of detrital Re and Os.

Lines 279-281: Do you present here these analyses? If not, why to mention?

Line 287: do you mean 'acid-volatile or elemental sulphur'?

Lines 306-309: Ok so you have sulfides with transition metals, pentlandite, chalcopyrite, and

galena. These are hydrothermal minerals. Why they are not discussed in the main text?

Some of D33S vs D36S diagrams have so much data that clearly data would fit on the ARA and MDF lines.

Supplementary Materials

Lines 38-39: Martin et al., 1999 argued for only one glacial event.

Lines 40-44: What is the evidence for this diamictite at the Boundary Ridge locality have glacial instead of debris flow origin?

Lines 48-50: I am totally confused now. How does this statement correspond with lines 202-204 in the main text arguing for irreversible rise of atmospheric oxygen at 2.45 Ga or earlier? I presume that it means during deposition of the Boolgeeda IF? If so, why your data argue against the same earlier interpretation?

Supplementary Figure 1a Poorly scanned figure with a wrong location for the Boundary Ridge. →
Figure 1b: The section is bounded by two thick →dolerite sills. Would not you expect some hydrothermal fluid circulation and contribution to sulfides in sediments? This is the section from which Re-Os data came.

Supplementary Figure 2: nothing distinctly glaciogenic on these figures.

Supplementary Figure 3: It would be useful to label with different symbols pyrite and mudstone samples.

Lines 121- 134: This is an oversimplified argument. People argued for ages about placer pyrite deposits. Sulfide layers parallel to bedding could be hydrothermal as well as finely disseminated crystals. So there are overgrowths related to hydrothermal or metamorphic processes, this is relevant to discussion in the main text on lines 167-170.

Lines 134-141: What is the source of this sulfur and what reductant was used in absence of abundant organic matter in this sediment? Presence of Fe-oxides and Fe-chlorite indicates late sulfidization. Why it cannot form by later processes? In fact, it seems to be the most logical explanation for me.

Figure 5: None of these pyrites look convincingly diagenetic to me.

Response to referees

Please find below our answers (*in italic*) and modification of the text (*in blue*) to reviewers' comments and the modifications we have made to the ms. In clarifying and editing the text, we have tried to take most of reviewers' concerns into account. You can find a revised version of our ms attached.

Response to Reviewers' comments:

Reviewer #1 (Remarks to the Author):

Philippot et al. report new S isotope data and a Re-Os age of “diagenetic” pyrite + mudstones from Western Australia. Their new data help further constrain the temporal trends in S isotope signatures, which in turn provide tighter constraints on the timing of atmospheric oxygenation and relict crustal recycling of mass-independent-fractionation of sulphur isotopes (MIF-S) beyond that provided in previous papers. I am enthusiastic about that aspect of the paper. I am not particularly wild about the authors approach for Re-Os geochronology. With appropriate modification of the manuscript, I think that this will turn out to be a good paper for Nature Communications, and think that it will be well-cited paper.

Lines 37-40: Depending on space constraints, for the abstract there needs to be a clear explanation for why, in a single locality/drillcore, there is alternation between essentially zero MIF-S (two intervals in T1) and low-amplitude but non-zero MIF-S. Oxidative continental weathering of sulphide minerals (and recycling of relict MIF-S signatures) should be occurring in both scenarios for the time interval covered in this study. The sentence on lines 41-42 isn't satisfactory as it does not clearly address the above issue.

*We agree that this aspect needs to be better addressed but feel that this is not adapted for the abstract. We have developed this aspect in section «**Atmospheric vs continental sulphur reservoirs** », comment Lines 159-161 below.*

Line 51: The authors could be more detailed in their coverage on age constraints for the youngest known S-MIF signal and the oldest known non-S-MIF signal, prior to this study. This introductory paragraph is rather vague about this.

We added the sentence:

The rise in atmospheric oxygen to above 10^{-5} to 10^{-2} PAL coupled with a decline in atmospheric methane⁵⁻⁷ is loosely constrained between ~ 2.50 Gyr, the age of the youngest sediments with no, or only minor $\Delta^{33}\text{S}$ anomalies (top of the McRae Shale Formation and Whaleback Shale Member of the Brockman Formation, Hamersley Group^{8,9}), and ~ 2.32 Ga, the age of the youngest rocks with strong MIF-S (Rooihogte and Timeball Hill formations, South Africa¹⁰⁻¹²).

and deleted:

~~The global disappearance of significant MIF-S by 2.2 Gyr ago is attributed to a major increase in $p\text{O}_2$~~

~~(to above 10^{-5} to 10^{-2} PAL) coupled with a decline in atmospheric methane~~

We also added Kaufman et al., 2007 and Reinhard et al., 2009 lines 54-55.

and Partridge 2008 and Reinhard 2009 in Figure 1.

Line 81: Are these early or late diagenetic pyrites? Diagenesis and pyrite formation can occur millions to tens of millions of years after initial deposition. Re-Os dates from macroscopic late diagenetic pyrite nodules are not necessarily informative regarding the timing of deposition, whereas synsedimentary pyrite (e.g., formed in the water column) and early diagenetic pyrite would provide close constraints on the timing of sedimentary deposition.

All Reviewers rose the question of the timing of pyrite formation. In order to better constrain this important issue we revisited the textural description of pyrite and analyzed the pyrites mounted for in situ sulphur isotope analyses for their trace metal composition using LA-ICP-MS. As shown by various authors, trace metal content appears the best tool to decipher early from late diagenetic pyrites and to evaluate if secondary fluid migration have affected the composition of the original sedimentary pyrite. This represents an important new set of data which has been summarized in Supplementary Figures 5, 6, 7, 8 and 9 and Supplementary Table 2. As shown in Supplementary Figures 8 and 9, the vast majority of the pyrite analyzed show Co/Ni, Zn/Ni and Cu/Ni ratios within the range of typical authigenic pyrite (i.e., between 0.01 and 10, Large et al., 2009). This clearly demonstrates that the pyrite analyzed were not affected by secondary hydrothermal fluid circulation nor local (diabase intrusion) or regional metamorphic overprint, and therefore represent reliable proxies of the processes occurring in the water column and/or during early diagenesis.

The following text has been added to the main text :

Pyrite trace element composition

Trace element abundances of pyrites were analysed by the LA-ICP-MS at the Australian National University. The system consists of an ArF (193 nm wavelength) excimer laser, which is interfaced to a custom-built ablation chamber (HelEx) coupled to an Agilent 7700 quadrupole ICP-MS. Ablation was performed in a He-Ar atmosphere with a pulse energy of ~ 80 mJ and pulse repetition rate of 5 Hz. Laser beam sizes ranging from 28 to 47 μ m were employed. The acquisition time per spectrum was set to 90 s, which comprises 30 s for counting the background and 60 s for counting the ablation signal. Before and after analysis of ~ 10 unknowns, reference materials were measured for external calibration and to perform instrumental drift corrections. USGS MASS-1 and STDGL-1 (Norman et al., 2003)⁵² were used as primary reference materials, and RTS-3, RTS-4, and GXR-1 were run as secondary reference materials. The data were processed using Iolite and ⁵⁷Fe as the internal standard. ²⁹Si and/or ⁴³Ca were used to estimate the contribution of matrix in each analysis. LA-ICP-MS trace element abundances are reported and summarized in Supplementary Table 2 and Supplementary Figs. 8 and 9.

The following text and Supplementary Figures (as well as Supplementary Table 2) have been added to the Supplementary Information :

Textures and chemical composition of sulphides

In all samples, preliminary identification of sulphides was performed by standard petrographic analyses of thin sections (Supplementary Figs. 5, 6, 7). **Distinction between syngenetic/diagenetic pyrites, and detrital and epigenetic pyrites was based on sulphide chemistry, host lithology, and textural features.**

Host lithologies that may contain detrital sulphides identified from their rounded appearance include the Koolbye Formation quartzites at the base of T3 (Figs. 2 and 4) and the 5-meter thick sandstone bed located beneath the Meteorite Bore Member diamictites (base of T2; Figs. 2 and 5). Only a few sulphides were analysed in these rocks and special care was taken to discard rounded grains of potential detrital origin, as these are not expected to preserve information on the sulphur cycle in the

depositional environment of the host rock. Similarly, none of the sulphides present in dropstones of the Meteorite Bore Member and of the Boolgeeda Iron Formation (Supplementary Figs. 2, 4) were analysed either by *in situ* or bulk rock techniques.

Detailed petrographic analyses indicate that the majority of the pyrites studied here show textures indicative of a syngenetic or early diagenetic origin. Examples of such textures include rounded and elongated zoned nodules, concretions, microcrystalline aggregates, framboidal cores, finely disseminated pyrite (<10 µm), small euhedral and subhedral crystals (< 25 µm) aligned with bedding (or conform to soft sediment deformation structures), and bands of closely spaced finely grained pyrite (Supplementary Figures 5, 6, 7). The relative arrangement and proportion of these different textural types in a particular horizon is variable but all forms are interpreted to originate during early diagenesis or, in the case of pyrite framboids, possibly syngenetically within the water column or diagenetically below the sediment-water interface¹³. A few samples, however, show textures indicative of a later generation of diagenetic pyrite. These include overgrowth of coarser euhedral to subhedral pyrites as well as individual large euhedral to subhedral pyrite crystals (> 25 µm). Pyrite overgrowths are regarded as diagenetic rather than from metamorphic and/or hydrothermal events as they occur associated with poorly permeable lithologies (e.g., mudstone) with no evidence of pyrite recrystallisation along schistosity planes. Coarser textures and overgrowth zones (see below) are interpreted to be the result of local re-mobilisation of Fe and S from smaller earlier pyrite generations. Of all samples studied, only one (T1-169.90) is known to record epigenetic pyrite as indicated by the presence of large, inclusion-free, euhedral pyrite crystals in a mm-scale vein.

Supplementary Figure 5. Backscattered electron (BSE) images of syngenetic and diagenetic pyrites analysed in this study. **a**, inclusion-rich nodular pyrite aggregate (sample T1-99.48), **b**, clusters of microcrystalline pyrites in green mudstone (sample T1-146.0), **c**, clusters of elongated anhedral pyrite crystals typically aligned with bedding observed in mudstone (sample T2-341.28), **d**, compacted inclusion-rich pyrite nodule in laminated mudstone (sample T2-365.23), **e and f**, inclusion-rich nodular aggregates in mudstone (samples T3-172.95 and T3-182.90).

Supplementary Figure 6. Backscattered electron (BSE) images of syngenetic and diagenetic pyrites analysed in this study. **a**, inclusion-free overgrowth around syngenetic to early diagenetic pyrite framboids observed in green mudstone (sample T1-130.39), **b**, finely disseminated euhedral to subhedral pyrite crystals (sample T1-152.18), **c**, small euhedral to subhedral pyrite crystals (< 25 µm) aligned with bedding in green mudstone (sample T1-159.30), **d**, small euhedral to subhedral pyrite crystals (< 25 µm) conform to soft sediment deformation structures observed in green mudstone (sample T1-159.47), **e**, band of densely packed microcrystalline pyrite aggregates (sample T1-172.1), **f**, anhedral overgrowth around large euhedral pyrite crystal in carbonate (sample T3-90.75).

Sulphur isotopic signatures provide another line of evidence for a sedimentary (syngenetic/diagenetic) origin of the pyrites. A syngenetic origin is particularly clear for pyrites in the 1.5 m-thick green siltstone horizon of the Boolgeeda IF, showing strong enrichment in sulphur content (up to 5 weight %), together with strongly negative $\delta^{34}\text{S}$ values (down to -35‰) and $\Delta^{33}\text{S} = 0\text{‰}$ (Fig. 3). In

these samples, sulphide occurs in association with quartz forming millimetre-scale layers parallel to the bedding in a matrix mainly composed of Fe-chlorite, Fe-oxides (magnetite and hematite) and apatite (Supplementary Figs. 6c, d and 7). Such layering cannot be formed by secondary processes, but instead reflects periodic delivery of non-anomalous sulphur component to the water column. Finally, authigenic sulphides that incorporate sulphur from the environment of deposition can display ranges of $\delta^{34}\text{S}$ and $\Delta^{33}\text{S}$ values, which are indicative of different sources and processes (photolytic, biological and/or nonbiological redox fractionation reactions at the time of deposition).

Supplementary Figure 7. X-ray maps of sulphide layers of drill core sample T1-159.47. a, b, Photo-micrographs of the petrographic thin section analysed. The red boxes represent zoom in views. Soft sediment deformations are locally preserved in this sample (Supplementary Figure 6d). **BS,** Electron backscattered image, **Fe-S-P,** composite X-ray map showing Fe-oxide (red), pyrite (purple) and apatite (green). Other X-ray maps correspond to sulphur (**S**), silicon (**Si**), iron (**Fe**) and phosphorus (**P**). Note that the sulphide-bearing layers are mainly composed of sulphide and silica. Further support for the syngenetic/diagenetic nature of sulphides is given by inter-element ratios, mostly Co/Ni but also Cu/Ni and Zn/Ni, determined by LA-ICP-MS. Previous studies¹⁴⁻¹⁷ have shown that Co/Ni ratio is an effective chemical indicator for the environment of pyrite formation. Volcanogenic-hydrothermal pyrite is generally characterized by a high Co content with a high Co/Ni ratio¹⁸⁻²⁰. On the other hand, high Ni and $\text{Co/Ni} \leq 2$ have been observed in pyrite that forms in organic matter-rich environments^{21,22}. Large et al. (2014)¹⁶ used Co/Ni ratios as a chemical screening for

sedimentary pyrites. On a Co versus Ni plot, samples which showed a linear array parallel to $\text{Co/Ni} = 1$ and with $\text{Co/Ni} < 2$ were selected as suitable indicators of a diagenetic origin¹⁶. In a recent study, Gregory et al., (2015)¹⁷ have defined several inter-element ratios that can be used as chemical proxies for sedimentary pyrites. Characteristic values and composition limits of sedimentary pyrites were determined to be: $0.01 < \text{Co/Ni} < 2$, $0.01 < \text{Cu/Ni} < 10$, $0.01 < \text{Zn/Ni} < 10$, $0.1 < \text{As/Ni} < 10$, $\text{Ag/Au} > 2$, $1 < \text{Te/Au} < 1000$, $\text{Bi/Au} > 1$, $\text{Sb/Au} > 100$, and $\text{As/Au} > 200$.

Here, we applied to our dataset only the Ni-based ratios as discriminants as they are more robust indicators due to their generally high abundance in pyrite. In addition, Co and Ni have similar ionic radii to Fe and tend to be incorporated into the structure of pyrite. The behaviour of other elements, like As and Au, is more complex due to their mobility over a wide range of redox conditions. Forty-two (42) samples have been selected for trace element analysis based on pyrite texture, lithology, and stratigraphic unit. The Co/Ni ratios measured show that the majority of the samples (40 out of 42) have $\text{Co/Ni} \leq 2$ within errors and all samples have Cu/Ni and $\text{Zn/Ni} < 10$ (Supplementary Table 2, Supplementary Figs. 8 and 9). Some samples show very low trace element abundances, which can be explained by the low trace element abundance observed in the host rocks. Several studies have highlighted that the trace element composition of sedimentary pyrites is highly dependent on the water chemistry as such the trace element abundance variations observed along the stratigraphic profiles (Supplementary Figs. 8) could be explained by changes on water chemistry and redox conditions (this aspect will be discussed in a separate paper). Variations of trace element composition do not show correlation with the different textural pyrite types described previously.

Supplementary Figure 8. Co/Ni, Zn/Ni, and Cu/Ni depth profiles of pyrite from Turee Creek drill cores (T1, T2, and T3). Red circles correspond to individual (spot) analyses carried out with LA-ICP-MS. The grey shaded areas represent the composition limits defined for sedimentary pyrites ($0.01 < \text{Co/Ni} < 2$, $0.01 < \text{Cu/Ni} < 10$, $0.01 < \text{Zn/Ni} < 10$). Despite the large range observed on Co/Ni, Zn/Ni, and Cu/Ni ratios measured, most of the samples analysed show inter-element ratios within the range of sedimentary pyrite^{17,22}.

Supplementary Figure 9. Binary plots for Co, Cu, and Zn vs Ni for pyrites from Turee Creek drill cores (T1, T2, and T3). Red circles correspond to individual (spot) analyses carried out with LA-ICP-MS. Cobalt correlates moderately well with Ni, with most of the samples from drill cores T1 (a), T2 (b), and T3 (c) plotting along the $\text{Co/Ni} = 1$ line or on a linear array parallel to $\text{Co/Ni} = 1$. A moderate to weak correlation is observed for Cu vs Ni (d, e, f) and Zn vs Ni (g, h, i), probably due to the scatter of the analytical data, which could be related to different trace element incorporation mechanisms into pyrite (i.e., trace element held within the pyrite structure or as nano-inclusions^{16,17}).

Lines 88-89: Is this diamictite truly a glacial diamictite? If so, what is the petrographic evidence? I cannot find anything on this in the main text or supplementary information. Not all diamictites have a glacial origin. Some diamictites can be formed by marine sediment gravity flows.

The evidence comes from two ~5 cm pebbles and several cm-scale pebbles isolated in a fine siltstone/mudstone in a meter-scale layer. These occur within the BIF, at about 20 meters below the contact with the overlying Kungarra Fm. This diamictite layer shows many similarities with the one reported by Martin (1999), who describe m-scale diamictite horizons in two localities north of the Hardey Syncline (Yeera Bluff and Duck Creek Syncline, i.e., Boundary Ridge locality of Williford et al (2011) and Van Kranendonk et al (2015)), for which there is clear evidence for dropstones with penetrating lower contact in mudstones and siltstones (see his Figure 5A). However, textural relationship between pebbles and host matrix cannot be observed in the 5 cm-large drill core section, and we agree that we cannot distinguish a glacial origin from a sediment gravity flow. Nevertheless, these two interpretations are not mutually exclusive. Martin (1999) interpreted the diamictites, sandstones, and laminated siltstones from the Yeera Bluff and Duck Creek Syncline to represent distal diamictites deposited from sediment gravity flows, sediment plumes, and rain-out from icebergs (see his Figure 9). This together with the geological context (same position of the diamictite horizons near the top of the Boolgeeda Iron Formation) and petrographical description (similar texture and type of pebbles presented in the Supplementary Information) strongly support a same glacial origin for the Boolgeeda Iron Fm. diamictite.

Importantly, because of the apparent absence of glaciogenic facies in the Boolgeeda Iron Formation in the Hardey syncline (our TCDP 1 core), Martin (1999) interpreted the different glaciogenic facies associations identified in the Yeera Bluff, Duck Creek and Hardey Syncline to be coeval and related to a single period of glaciation. To explain that the Yeera Bluff and Duck Creek diamictites occur in the Boolgeeda BIF and not in the Kungarra, he proposed that the banded iron formation deposited in Yeera Bluff and Duck Creek is a facies equivalent of fine-grained clastic deposits in the lower Kungarra Formation in the Hardey syncline. Lateral variations in the facies associations from south to north were interpreted to reflect a gradation from ice-proximal to ice-distal depositional environments.

The finding of a diamictite layer in the Boolgeeda Iron Fm. of the Hardey Syncline and our new U-Pb detrital zircon age constraints (Caquineau et al, 2017, in review) allow a different interpretation. Results obtained on the diamictites in TCDP1 (Hardey Syncline) and from the Boundary Ridge locality show a minimum concordant age of 2.45 Gyr (Figure 1 below). This implies that these horizons are older than the MBM by about 100 Ma (see further discussion about age constraints in the TCG below) and are best attributed to the first Paleoproterozoic glacial event (Ramsay lake and Makganeyne glacial events according to the new age of Gumsley et al., 2017).

Figure 1: Detrital zircon U-Pb age from the two Boolgeeda Iron Formation diamictites (TCDP 1 and Boundary Ridge locality of the Duck Creek Syncline) showing a concordant maximum age of 2.45 Ga, in agreement with the age of the underlying Woongarra Rhyolites (Caquineau et al., in review).

In order to strengthen the description of the diamictite we added in the Supplementary Information :

The 2 m-thick diamictite horizon located within the Boolgeeda Iron Formation at about 20 meters below the contact with the overlying Kungarra Formation of our T1 drill core (Supplementary Fig. 2 d,e,f) shows the same textural characteristics as the one identified at the Boundary Ridge locality (Supplementary Fig. 2 g,h). The evidence comes from several ~5 cm pebbles and several cm-scale pebbles isolated in a siltstone matrix in a meter-scale layer. This clearly demonstrates that the Boundary Ridge diamictite represents a distinct glacial event separated from the overlying MBM by nearly 1500 metres of the Kungarra shales. Our new Re-Os sulphide age of ~2.31 Gyr (Supplementary Fig. 3) for the base of the MBM supports this interpretation, namely that the deep basin glaciomarine horizon identified in the Boolgeeda IF is much older and most likely correlative with the first Huronian glacial event (Ramsay Lake) at about 2.45 Gyr. This interpretation is further confirmed by a

new U-Pb age of 2.45 Ga obtained on detrital zircons extracted from the diamictite horizons of the Boundary Ridge and TCDP1 core (Hardey Syncline)¹¹. It also indicates that the other two or three glacial deposits present in the Kungarra Formation are ~2.31 Gyr old or slightly younger. Accordingly, our data argue against the interpretation that the disappearance of MIF-S anomalies and occurrence of strongly depleted $\delta^{34}\text{S}$ values identified at the top of the Boolgeeda IF at the Boundary Ridge locality represents the MIF to MDF transition as it was interpreted in Williford et al., (2011)⁸.

Lines 107-108: Will general readers of Nature Communications understand what “anomalous” and “non-anomalous” means for $\Delta^{33}\text{S}$? Only isotope geochemists or Precambrian geoscientists would recognize this. Please explain briefly in the main text so that general readers do not have to look in the supplementary information for it.

We change the sentence:

(i) strongly variable $\delta^{34}\text{S}$ in both anomalous and non-anomalous $\Delta^{33}\text{S}$ sulphides in the interval below 130 m depth in T1

into

(i) strongly variable $\delta^{34}\text{S}$ sulphides showing either no ($\Delta^{33}\text{S} = 0\text{‰}$), or slightly positive or negative $\Delta^{33}\text{S}$ values in the interval below 130 m depth in T1

Line 129: “minor sulphur isotope signal” is not the best phrase to use here? Refer specifically to a minimal MIF-S signature.

We changed minor sulphur isotope signal into a minimal deviation from mass dependent fractionation signal

Lines 159-161: This is a rather vague statement. The two main excursions to $\Delta^{33}\text{S} \sim 0\text{‰}$ both occur in T1 – the oldest part of the studied intervals. Why does $\Delta^{33}\text{S}$ go to 0‰ at these two intervals when clearly we have not yet exhausted (via weathering) the reservoir of continental sulphides with non-zero $\Delta^{33}\text{S}$? The subsequent sentences are not entirely satisfactory given that there are also slightly negative $\Delta^{33}\text{S}$ signatures in T1. Please explain these excursions to zero $\Delta^{33}\text{S}$ more clearly.

We added the following paragraph line 163-175 to explain these excursions:

The two green siltstone and diamictite horizons in the Boolgeeda IF that display high sulphur contents (0.12 to 4.7 weight %) and $\Delta^{33}\text{S} \sim 0\text{‰}$ are bounded above and below by BIFs with low sulphur content (<1000 ppm) and small positive to near zero $\Delta^{33}\text{S}$. Considering that the BIFs represent deep water deposits with minimal influence from terrigenous inputs, such abrupt change in sulphur contents and isotope compositions is best explained by the contribution of two competing sulphur sources; (i) a continuous background sulphur source characterized by low sulphur content (< 1000 ppm) and small positive to near zero $\Delta^{33}\text{S}$ anomalies linked to oxidative continental weathering recorded in the BIFs, and (ii) a transient sulphur source of atmospheric origin characterized by high sulphur content (1 to several weight %) and $\Delta^{33}\text{S} = 0 \text{‰}$ recorded in the green siltstone and diamictite horizons. The occurrence of strongly ^{34}S -depleted sulphides in green siltstones and near zero $\delta^{34}\text{S}$ sulphides in the diamictite can be explained by the availability of organic matter. High primary productivity in the Paleoproterozoic ocean may have occurred during deposition of the green siltstones, which would lead to temporary euxinic conditions⁴¹, whereas low primary productivity during glaciation would have hampered microbial sulphate reduction. The recognition that other green siltstone layers from the same stratigraphic succession do not show high sulphur contents and $\Delta^{33}\text{S} = 0 \text{‰}$ suggests that the transient expression in the sedimentary record of the atmospheric sulphur source can be best accounted for by the episodic nature of volcanic SO_2 production rather than by fluctuation in atmospheric oxygen levels.

Support for the occurrence of two main sources of sulphur in the formation of sedimentary pyrite is also observed on a grain scale. Sample 130.39...

Lines 185-186: I note that such signals (attenuated MIF-S isotopes) were also observed transiently in older rocks (namely the 2.5 Ga Mt. McRae Shale; Kaufman et al., 2007; Reinhard et al., 2009), and more persistently in the overlying Brockman Iron Formation by Reinhard et al. (2009; Science). I do not see any incorporation of this important data into this paragraph, and it is needed to provide a complete discussion of temporal trends in S isotope signatures in western Australian sections.

Kaufmann et al (2007) and Reinhard et al (2009) have been added twice in the introduction paragraph (lines 51 and 55), in Figure 1 and, in line 204 when we introduce the notion that the rise of oxygen could be earlier than 2.45 Ga and in line 225-226. We also added Reinhard et al (2009) in the paragraph above dealing with temporary euxinic and sulphidic conditions.

Line 189: Some have interpreted the GOE this way, but others look at the GOE as representing a protracted transition rather than an event (e.g., Lyons et al., 2014; Nature). I think the latter is now more accurate given the data from this study and others. As such, the authors should consider using different wording than “currently interpreted as the marker horizon” as that is not how many of us view the GOE. Indeed, the authors later correctly cast the GOE as having multiple “milestones”.

We are specifically referring here to the sedimentary successions in South Africa, specifically the recent papers of Luo et al (2016) and Guo et al (2009) following the work of Bekker et al (2004). The GOE as a whole is addressed below. We feel it is important to emphasize the notion of « marker horizon » here as it represents a deeply rooted belief for most authors (e.g., Luo et al paper was published May 2016). To make it more specific, however, we modified the sentence by adding « the end of the GOE », so that :

The transition from MIF-S to MDF-S in the upper Rooihogte and Deutschland formations in South Africa, which occurred before 2.316 Gyr (Fig. 1), is currently interpreted as the marker horizon for the end of the GOE.

Line 204: Again, mention the Brockman Iron Formation data.

Done

Line 207: This sentence really ought to be in the abstract!

Done

Line 231: Geochronology suggests 2.06 Ga, not 2.10 Ga, for the end of the Lomagundi Event.

Changed to 2.06 Gyr

Line 241: If I understood this correctly, a single sample on the Re-Os isochron is being defined as a set of mixed pyrites collected from four different mudstone samples in a single formation. That is unorthodox! What is the reason for doing this? State the thickness of the stratigraphic interval encompassed by the samples defining the Re-Os isochron. Again, there must be a description of the pyrite textures. What can be said about local bottom water redox conditions – at least sediment anoxia is needed for the Re-Os geochronometer to work as hoped. Although there is a good MSWD, I note that if at the time of deposition there was a non-zero slope on the isochron diagram defined by mixed whole rock and late diagenetic pyrite analyses, then the age calculated from present-day isotope ratios would not represent the depositional age. Hence, it is important to demonstrate the timing of pyrite formation through textural analysis. There is some general description in the supplementary information about pyrite textures, but it is paramount to know the textures of specific pyrites chosen

for Re-Os analysis. Are syn-sedimentary (formed in a euxinic water column) or early diagenetic pyrites used for Re-Os analysis?

The reason of mixing pyrite from 4 different samples is due to the paucity of pyrite in individual samples. We were careful in choosing the same lithology so as to avoid mixing of pyrite of different origin. With regards of the pyrite texture and origin we added a lengthy new discussion and new results in the Supplementary Information (see above comment, line 81).

Lines 247-248: The authors use a digestion method that is known to liberate detrital Os from shale matrices (which can potentially yield inaccurate ages because of mixed detrital-authigenic Os isotope compositions). Instead they should have used the chromium-sulphuric acid digestion technique on their mudstone samples (inverse aqua regia is fine for pyrites) that minimizes the release of detrital Os into the digested solution (Selby and Creaser, 2003; Rooney et al., 2011; Kendall et al., 2013). This is important considering that the Re and Os concentrations of the samples are quite low (note the total Os content largely reflects ^{187}Os production via ^{187}Re decay over > 2 Gyr; one can calculate the common Os concentration at the time of formation and compare with estimates of the upper crust Os concentration). Perhaps these concerns could be addressed via regression of pyrite analyses alone, but there are too few pyrite analyses. Finally, a low MSWD is NOT a guarantee of age accuracy. There is a very uneven distribution of data points along the regression line – the slope is heavily controlled by a single, radiogenic, pyrite data point. Hence, the precision and accuracy of this age are not particularly robust despite appearing to be geologically reasonable (i.e., the pre-existing age constraints are rather loose, and the Re-Os age could be off by tens of millions of years and still appear geologically reasonable!).

Reviewers #1 and #3 pointed out that the inverse aqua regia digestion technique used for Re-Os dating is adapted for pyrite separates but not for bulk samples. For this reason, and in order to confirm unambiguously the Re-Os age, we performed additional measurements using the chromium-sulphuric acid digestion technique of Selby and Creaser, (2003). Taken together, the data indicate a Re-Os age of 2309.0 ± 9.2 Ma. Taken separately, the data using the chromium-sulphuric acid digestion technique indicate a Re-Os age of 2314 ± 19 Ma. The two ages are similar within error to the Re-Os of 2310.2 ± 6.8 Ma obtained in the first version of the manuscript using the inverse aqua regia digestion technique. This new data together with the detrital zircon U-Pb age of 2.34 Ga of Caquineau et al (submitted) obtained on similar samples unambiguously confirm that the Meteorite Bore Member is 2.31 Ga old.

We modified the Methods Section as :

Re-Os geochronology

Three samples of diamictite from the Meteorite Bore Member (T2-169.05, T2-205.94 and T2-272.46), two samples of mudstone/siltstone from the Kungarra Formation (T2-357 and T2-368) and three sets of pyrite separates (2 different fractions of sample T2-272.46 and a set of pyrites collected in 4 different mudstones of the Kungarra Formation, T2-332-351-357-360) were used for Re-Os geochronology (Supplementary Table 1). The reason for mixing pyrites from four different samples of mudstone/siltstone of the Kungarra Formation underlying the Meteorite Bore Member is due to the paucity of pyrite.

The Re and Os concentrations and the Os isotopic composition were analysed at the John de Laeter Centre for Isotopic Research at Curtin University in Perth (Australia) using the Carius tube digestion method⁴⁸. Samples were digested using two different mediums: reverse aqua regia and $\text{CrO}_3 - \text{H}_2\text{SO}_4$ ⁴⁹ (Ref 49). For the first method, approximately 1 g of whole-rock sample powder for shales, and 80 to 100 mg of pure pyrite separate were mixed with appropriate amounts of ^{185}Re and ^{190}Os spikes. The acid digestion was carried out using concentrated acids (3 ml of purged double-distilled HNO_3 and 6 ml of triple distilled HCl). The analysis of these samples were replicated by starting digestion using $\text{CrO}_3 - \text{H}_2\text{SO}_4$ mixture as described by Selby and Creaser (2003)⁴⁹ to extract preferentially hydrogenous Re and Os. For this technique, 2 to 3 g of sample powder were digested in

16 ml of $\text{CrO}_3 - \text{H}_2\text{SO}_4$ mixture⁴⁹. For both methods, the obtained mixture was chilled and sealed in previously cleaned PyrexTM borosilicate Carius tubes and heated to 220°C for 60 h. Osmium was extracted from the acid solution by chloroform solvent extraction⁵⁰, then back-extracted into HBr, followed by purification via microdistillation⁵¹. Rhenium was separated from a portion of the residual solution using anion exchange chromatography.

The purified Os and Re fractions were loaded onto Pt filaments, and measured using N-TIMS on a ThermoFisher TritonTM mass spectrometer using a secondary electron multiplier detector. The measured isotopic ratios were corrected for mass fractionation using $^{192}\text{Os}/^{188}\text{Os}=3.092016$, as well as spike and blank contributions. The internal precision of $^{187}\text{Os}/^{188}\text{Os}$ ratio measured in all samples was better than 0.15% (2σ). To monitor long-term instrument reproducibility, an AB-2 Os standard (University of Alberta) was analysed. The AB-2 Os standard yielded 0.10687 ± 0.00012 ($n=2$, 2σ) during the period of the measurements, which is consistent with the value reported by Selby & Creaser (2003)⁴⁹ (0.10683860 ± 0.00004). An in-house Re standard solution gave $^{185}\text{Re}/^{187}\text{Re}=0.5987\pm 0.0008$ ($n=2$, 2σ). The total procedural blank was 0.5 pg for Os ($n=2$) and 7 pg for Re. The $^{187}\text{Os}/^{188}\text{Os}$ ratios of the blank was 0.202 ± 0.006 . For whole-rock analyses, the total Os analytical blank represented less than 0.5% of the total Os. Since total blank for both Re and Os was run as part of each batch of dissolutions, appropriate blank correction was applied to each batch.

The Re and Os concentrations in pyrite range from 1.9 to 3.6 ppb and from 109 to 262 ppt, respectively (Supplementary Table 1). The Re and Os concentrations in mudstones are more variable, ranging from 0.9 to 1.9 ppb and 122 to 1,980 ppt, correspondingly. Replicate analyses of the samples show variable Re and Os concentrations. This can be explained by the presence of variable quantities of sulphides, possibly due to ‘nugget’ effect.

The plot of isotope data for eight mudstone-siltstone (including duplicates) and three pyrite separates on the Re-Os isochron diagram (Supplementary Figure 3a, Supplementary Table 1) defines a best-fit line with an age of 2309.0 ± 9.2 Ma (MSWD = 3.3) and an initial $^{187}\text{Os}/^{188}\text{Os}$ ratio of 0.14 ± 0.03 . The data indicate that some scatter is present in the system (MSWD>1), probably due to variable initial $^{187}\text{Os}/^{188}\text{Os}$ ratios in the detrital material and/or possible influence of Archaean seawater, which may have affected the Os budget in the sediments, due to reequilibration with hydrogenous Os from the ocean column. Four samples analysed using $\text{CrO}_3\text{-H}_2\text{SO}_4$ medium preferentially released hydrogenous Os and Re, and showed less scatter (MSWD = 1.06) with similar age of 2314 ± 19 Ma (Supplementary Figure 3b), with a relatively well-defined initial ratio of 0.15 ± 0.01 .

We modified the Supplementary Figure 3 and Supplementary Table 1 as:

Supplementary Figure 3. Re-Os age. Re-Os isochron plot for **(a)** all samples (three pyrite separates and eight shale samples, including replicates). **(b)** Plot showing only Re and Os data for bulk samples determined using $\text{CrO}_3 - \text{H}_2\text{SO}_4$ digestion medium. Error bars are $\pm 2\sigma$. Regression based on Isoplot¹².

Supplementary Table 1. Re-Os concentrations and Os isotopic compositions for bulk samples and pyrite separates.

Sample	Lithology	Re ppb	Total Os, ppt	$^{187}\text{Re}/^{188}\text{Os}$	$^{187}\text{Os}/^{188}\text{Os}$
T2 – 272.46	Pyrite	2.89	109	865.9	34.090(9)
T2 – 272.46 repeat	Pyrite	3.58	206	147.1	5.940(4)
T2-332,357,360,351	Pyrite	1.94	262	41.9	1.710(7)
T2-169.05	Diamictite	15.65	1872	50.8	2.140(7)

T2 – 169.05*	Duplicate*	16.59	1980	50.6	2.129(8)
T2 – 205.94*	Diamictite	3.46	234	86.1	3.546(5)
T2 – 272.46*	Diamictite	4.05	418	54.2	2.288(5)
T2 – 357	Mudstone	1.86	168	73.2	3.040(6)
T2 -357	Duplicate	5.32	441	81.5	3.280(3)
T2 – 368	Siltstone-mudstone	0.85	122	39.8	1.630(7)
T2 – 368*	Duplicate	4.00	1508	12.5	0.645(8)

*CrO₃ – H₂SO₄ digestion.

Number in parenthesis indicates the uncertainty as 2σ in the measured Os isotopic ratio.

Line 273: What does “influence of Archean seawater” mean???

*Seawater ¹⁸⁷Os/¹⁸⁸Os (as recorded by the initial ¹⁸⁷Os/¹⁸⁸Os [*I*_{Os}] of Re-Os isochron regressions, ~ 0.1517 and 0.117) reflects a balance between the riverine flux of radiogenic Os from oxidative weathering of upper continental crust (present-day value of ca. 1.0-1.5) and the flux of unradiogenic Os from hydrothermal alteration of oceanic crust and peridotites, and cosmic dust (ca. 0.11-0.13). During the Early Paleoproterozoic, oxidative weathering of crustal sulphides may represent a measurable, albeit minor source of Re and Mo relative to the hydrothermal and extraterrestrial fluxes. Because the Os budget of the Archean continental crust should be radiogenic due to the relatively high Re/Os ratio of continental rocks, it produces ingrowth of radiogenic ¹⁸⁷Os over time. In the present case, the data can reflect oxidative weathering of the Archean continental crust displaying an initial ¹⁸⁷Os/¹⁸⁸Os similar to DM (¹⁸⁷Os/¹⁸⁸Os ca. 0.11-0.12), which is unlikely, or indicates that the Os budget in the analysed sediments was overwritten by hydrogenous Os from the ocean column.*

We modified the text as :

The data indicate that some scatter is present in the system (MSWD>1), probably due to variable initial ¹⁸⁷Os/¹⁸⁸Os ratios in the detrital material and/or possible influence of Archean seawater, which may have affected the Os budget in the sediments analysed due to reequilibration with hydrogenous Os from the ocean column.

The authors have a lot of long paragraphs. Shorter paragraphs, each headed by a topic sentence, would be much easier for the general readers of Nature Communications.

We feel that the organization of the paper reflects the guidelines of Nature Communications.

Figure 1: The S isotope record should be extended back to 2.5 Ga. Why exclude all the S isotope data for the Mt. McRae Shale and Brockman Iron Formation, etc? Isn't there S isotope data at 2.5 Ga in the S. African sections? Inclusion of this data would really help further illustrate the temporal trends!

We modified Figure 1 including the data of Partridge et al (2008) on the Whaleback Shale Member at the base of the Brockman Fm and that of Kaufman et al. (2007) and Reinhard et al. (2009) on the McRae Shale such as :

Figure 1. New age-calibrated correlation of Late Archean - Paleoproterozoic sedimentary successions from North America, South Africa and Australia. The Re-Os sulphide age from this study is shown as 2.31 Gyr*. Other age constraints are labelled accordingly. Ages with no reference can be found in Rasmussen et al.¹⁹. The inferred Great Oxidation Event (GOE*) from this study is shown as a ~300 Myr interval based on combined age, sulphur isotope and stratigraphic data. Other GOEs inferred from previous studies are labelled accordingly. The vertical arrow labelled Makganyene⁶⁰ refers to the range of age inferred for the Makganyene glacial deposit based on a new U-Pb age of the Ongeluk Volcanics, Transvaal Supergroup, South Africa⁶⁰. MIF, mass-independent fractionation of sulphur isotopes; MDF, mass-dependent fractionation of sulphur isotopes. The mixed light blue-red domain labelled (MDF) shown at the top of the Turee Creek Group (this study), in the Whaleback Member of the Brockman Formation⁴⁰ and McRae Shale Formation⁴¹ of the Hamersley Group, and at the base of the Huronian sedimentary column¹⁶⁻¹⁸, corresponds to sedimentary horizons in which strongly attenuated $\Delta^{33}\text{S}$ and $\Delta^{33}\text{S}/\Delta^{36}\text{S}$ systematics are attributed to increases in $p\text{O}_2$ above 10^{-5} to 10^{-2} PAL (see text). Abbreviations, Hamersley Basin: McRS, McRae Shale Formation, Br, Brockman Formation, WW, Weeli Wolli Formation, Wo, Woongarra Rhyolite, Boo, Boolgeeda Iron Fm., Kun, Kungarra Fm., MBM, Meteorite Bore Member, Ko, Koolbye Fm., Ka, Kazput Fm., BRG, Beasley River Quartzite, CSB, Cheela Springs Basalt. Huronian Supergroup: Li, Livingstone Creek Fm., Th, Thessalon Fm., Ma, Matinenda Fm., McK, McKim Fm., RL, Ramsay Lake Fm., Pe, Pecors Fm., Mi, Mississagi, Fm., Br, Bruce Fm., Es, Espanola Fm., Se, Serpent Fm., Go, Gowganda Fm., Lo, Lorrain Fm., GL, Gordon Lake Fm., BR, Bar River Fm. Eastern Transvaal Basin: Mal, Malmani Fm., Pen, Penge Iron Fm., Tongwane Fm., Dui, Duitschland Fm., TH, Timeball Hill Fm., Bo, Boshhoek Fm., HV, Hekpoort Volcanics.

Reviewer #2 (Remarks to the Author):

Philippot and colleagues present new S isotope data and a new Re-Os age from diagenetic sulphides from a series of drillcores through a continuous sequence of sedimentary rocks from Western

Australia (Turee Creek Group). Both bulk and in-situ S isotope data from sedimentary sulphides are interpreted in the context of the new Re-Os age constraint and previous geochronologic data to suggest a new framework for the 'Great Oxidation Event' (GOE), a colloquial term for the secular oxygenation of Earth's atmosphere during Paleoproterozoic time.

The extensive new S isotope data, both bulk and in-situ, from a unique archive recently unearthed by the Turee Creek Drilling Project represent an important advance in our understanding of the Precambrian Earth system. The new Re-Os age data, together with the S isotope systematics observed in the Kazput Formation (core T3) in particular, on their own represent a critical new piece of the puzzle in attempts to reconstruct the tempo and mode of Earth's early oxygenation. Although there are naturally debates to be had about timing and mechanism in the S isotope data, given the amount of data presented in the manuscript and the somewhat provocative conclusions, I have very little doubt that the basic result here is robust and important. I would thus be happy to see the paper published in Nature Communications following minor revision (subject to editorial discretion).

One suggestion I would make to the authors is to more explicitly describe the origins of the sulphide phases being analyzed. Many of the supplementary images show cubic pyrites that are clearly post-depositional, and there is some discussion of core-rim differences. I think it would be beneficial if the authors can say something brief and forceful in the main text about the presumed origin of the sulphide phases -- as this will for some readers be the first criticism to be leveled at the interpretation offered in the manuscript. It is likely a mixed assemblage, but in particular can the authors be confident that there hasn't been large-scale overprinting by metamorphic fluids? Better to tackle this head-on then leave the reader to ferret it out in the supplement -- some readers won't, and will remain skeptical (with the result that the impact of the paper may become muted).

Same comment as line 81 above.

Reviewer #3 (Remarks to the Author):

Review of the manuscript by Philippot et al. entitled "Re-defining the Great Oxidation Event with the sulphur record"

The manuscript presents a large dataset of multiple S isotope data for the Turee Creek Group in Western Australia that records the early stage of the GOE. The authors draw two major conclusions: 1) GOE started around 2.45 Ga or earlier; 2) from that time on, local recycling of terrestrial sulphides provided a mass-independent signal to sedimentary successions resulting in a heterogeneous landscape of S isotope record after the beginning of the GOE. The first conclusion is not surprising; see Gumsley et al., 2017 for the most recent publication on this topic (although I am not convinced by the evidence presented in this manuscript and where they set up the transition). The second conclusion in my mind is unsubstantiated and it feels that authors want to fit their data to the prevailing model. There are several geological and geochemical reasons to doubt the second conclusion. First, detrital pyrites has been described throughout the Turee Creek Group (see Krapez et al., 2017; also Kranendonk et al., 2015) so pyrites remained stable under surface conditions during deposition of the Turee Creek Group.

It is important to make the distinction that the detrital pyrite mentioned in Krapez et al (2017) are interpreted as such, and probably based on outcrop samples. We found two sentences on page 71 'Albeit typically weathered, sandstones contain rounded grains of detrital pyrite » and page 72 « Although highly weathered and strongly cleaved, the diamictites and sandstones contain rounded grains of detrital pyrite.» As mentioned by these authors, pyrite visible in surface samples are strongly weathered. We fully agree with this. In surface samples, rounded pyrite are commonly observed due to weathering and can indeed be misinterpreted as detrital pyrite. In fact this is the reason why we performed drilling: to obtain unweathered sample adapted to in depth petrographical and chemical and isotopic analyses.

Van Kranendonk et al (2015) did not report any evidence for detrital pyrite but mentioned the early

work of Williford et al (2011), who reported petrographic description and S-isotope analysis of pyrite from the 7 m-thick section of the Boundary Locality located 100 km to the north east of the Hardey syncline. This reduced section comprises the top of the Boolgeeda Iron Formation together with a few meters of terrigenous rocks, mudstones and glacial diamictites. These authors show evidence for the presence of detrital pyrites in two sandstone samples. These detrital pyrites show highly heterogeneous D33S values between -4 and + 15%. Williford et al (2011) did not take these few grains into consideration in their interpretation of their S-isotope results.

We have investigated more than 250 petrographic thin sections from the 3 drill cores, which on the contrary of surface samples are not weathered, therefore minimizing misinterpretation of the nature of the observed pyrite grains. In the Supplementary Material we wrote:

Host lithologies that may contain detrital sulphides identified from their rounded appearance include the Koolbye Formation quartzites at the base of T3 (Figs. 2 and 4) and the 5-meter thick sandstone bed located beneath the Meteorite Bore Member diamictites (base of T2; Figs. 2 and 5). Only a few sulphides were analysed in these rocks and special care was taken to discard rounded grains of potential detrital origin, as these are not expected to preserve information on the sulphur cycle in the depositional environment of the host rock. Similarly, none of the sulphides present in dropstones of the Meteorite Bore Member and of the Boolgeeda Iron Formation (Supplementary Figs. 2, 4) were analysed either by in situ or bulk rock techniques.

So indeed, we agree that detrital pyrites are present in some of the rocks investigated, but these are rare and localized in specific lithologies such as sandstone and quartzite. Importantly, the occurrence of detrital pyrites in these rocks is not contradictory with the rise of atmospheric oxygen. Rapid erosion of continental surfaces leading to sandstone formation in an atmospheric environment with a pO_2 between 10^{-5} and 10^{-2} PAL could locally preserved some detrital grains.

This interpretation can be counter-argued that it does not exclude a small terrestrial sulphate flux. More importantly, Fig. 1 of their own manuscript challenges their own arguments. Large MIF-S was reported in the Duitschland and Rooihogte-Lower Timeball Hill formations, which are unequivocally 100 Ma younger than the beginning of the GOE as placed in this manuscript. This large MIF-S signal cannot represent recycling of continental S, as D33S signals are +4 and even +6 permil (Guo et al., 2006; Luo et al., 2016). This is not a signature of recycling as defined in this manuscript and indeed recently published paper by Gumsley et al. (2017) (not referenced in this manuscript) inferred multiple rises and falls of MIF-S in association with glacial events. This manuscript does not provide an explanation for multiple glaciations by inferring recycling model.

The presence of a damped but measurable mass-independent sulphur isotope signal in samples that are younger than 2.45 Ga but older than 2.0 Ga were recognized by Farquhar et al (2000), Farquhar and Wing (2003) and ongoing papers following their pioneering work on MIF-S pyrites. This observation has been confirmed by the vast majority of studies including that of Guo et al (2009, not 2006), who reported MIF-S values between about 0.2 and 1.5 ‰, which are within the range of the one reported here (we could not find data at +4 and +6 ‰ in Guo et al paper).

Luo et al (2016), indeed reported a few MIF-S values > 2‰. Most of these data are from sandstone and therefore could be detrital pyrite, very much like the one reported by Williford et al (2011) in the two sandstone layers located at the top of the Boolgeeda Iron Formation. These detrital pyrites show strongly positive and heterogeneous MIF-S values (see above comment). Besides, the analyses reported by Luo et al. (2016) are bulk rock analyses and therefore most likely represent a mixture of different pyrite generations. Importantly, note that the study of Luo et al (2016) is lacking petrographical description of pyrite.

Here we report the largest S-isotope dataset ever reported on the Paleoproterozoic (more than 1000 analyses). We clearly demonstrate that the MIF-S signal is restricted to low $\Delta^{33}S$ values (0 to 1.5 ‰). This is a robust and statistically relevant dataset, not interpretation based on a few analyses.

The paper of Gumsley et al (2017) was published February 21, 2017, that is one month after the submission of our paper to Nature Communications, so we were not aware of this paper. This paper is now cited in Figure 1 and the age of the Makganyene event is shown in Figure 1.

It is important to understand how authors arrived to these conclusions since geological and geochronologic framework is critical for their interpretation. The generally accepted view, recently summarizing by Krapez et al. (2017) is that the Turee Creek Group was deposited in a short-lived foreland basin that closed by ~2.43 Ga or shortly after.

This is not the generally accepted view, only that of Krapez et al. (2017). Most authors consider the basin to be long-lived (e.g., Martin, 2000). We have found 2.34 Gyr old detrital zircons at the bottom of the MBM diamictites (see comment below). This new age and our Re-Os age imply that the basin was open at 2.31 Gyr soon before and during the deposition of the MBM, top of the Kungarra Formation, and Koolbye quartzites and Kazput carbonates, representing a total of more than 1500 meters of sediments.

Authors challenge this view and argue that 3.5 km of section record over 200 Ma of sedimentation, making this basin a unique bearer of a continuous sedimentary record for this time interval with the slowest depositional rate and longest record of deposition in any other basin of this or even any age.

The depositional rates used in this paper are those reported by Trendall et al (2004) for the Harmersley Group underlying the Turee Creek Group. A long-lived basin of ~250 Myr duration for the Turee Creek Group has been proposed by Martin (2000). This is not new. Our new age constraints support this model and argues against the short-lived model of Krapez et al (2017).

This revision, that might not be obvious to unfocused reader, is based on three pieces of evidence. 1) Diamictite within the Boolgeeda Iron Formation of inferred glacial origin; 2) U-Pb detrital zircon age from the glacial diamictite 1.5 km higher in the section; and 3) Re-Os age for pyrites and mudstones sampled over 100 m of section.

Evidence for glacial origin of the diamictite in the Boolgeeda Iron Formation is not strong in my mind, especially if it is not anymore correlative to the well-described glacial Meteorite Bore Member. I found only mention of lamina penetration by a lonestone, however overlying lamina are not shown on this figure so it could a deformational feature (see Kranendonk et al., 2015).

The Figure above is taken from Van Kranendonk et al (2015, Figure 21). It is defined as: Outcrop photographs of facies association 4 from the Boundary Ridge locality: (A) rhyolite dropstone with penetrating lower contact in very finely laminated mudstone, from 10 cm above the top contact of the underlying transitional chert unit.

See also Figure 5 of Martin (1999), who show clear evidence of dropstones for the diamictite horizon of the Boolgeeda Iron Formation from the Duck Creek Syncline and Yeera Bluff Localities.

These diamictites of uncertain origin could be simply debris flows during transition to a foreland basin. Detrital zircon age is not published; it is only mentioned in the conference abstract with no error bars presented for the age (manuscript states that it is 2340 +/- 22 Ma with no data presented and 22 Ma errors might indicate potential complications). This is not strong evidence since the recent comprehensive study by Krapez et al. (2017) did not find any zircons younger than ca. 2.43 Ga in a number of samples covering the whole Turee Creek Group and ca. 2.3 Ga grains were only found in the overlying Beasley River Quartzite. I think it is thus premature to rely on this age.

The 2.34 Gyr age is a strongly reliable concordant U-Pb age and there is no reason to ignore this important piece of information.

The last piece of evidence is the Re-Os date presented in this paper. However, not being a Re-Os specialist, I see several potential issues with this age. First, it is based on samples collected over 100 m thick section in area with dolerite sills. Normally, samples for Re-Os work are collected over 1-2 m of thickness or better along the single bed to avoid natural variations in initial Os isotope ratio of seawater. Second, methods developed to preferentially attack Re and Os associated with organic matter in Selby's and Creaser's labs were not used so detrital Re and Os contribution could have been substantial. Lastly, considering that 2 points have very similar composition (surprisingly mudstone and a composite of pyrites), it is a 4-point isochron, a bare minimum in Re-Os geochronology. Was uncertainty on the decay constant included in calculations? I therefore consider this date to be uncertain at this stage.

New Re-Os ages have been obtained using the chromium-sulphuric acid digestion technique confirming the 2.31 Gyr age (see above), which are in good agreement with the maximum 2.34 Gyr U-Pb age obtained on detrital zircon on the same samples.

Let's turn argument around and accept author's geologic and geochronologic framework for a moment. Using Fig. 1, diamictite in the Boolgeeda IF would be ca. 2.45 Ga in age and correlative to the oldest Huronian glacial. Diamictite in the Meteorite Bore Member (actually two events based on Kranendonk et al., 2015) is younger than 2.31 Ga and would be correlative with the fourth glacial event of the Paleoproterozoic bracketed between ~2.26 and 2.22 Ga in South Africa. There are several critical issues here: 1) in an inferred continuous record of the early Paleoproterozoic 2 Huronian glacials – middle and the upper one – are missing between the Boolgeeda and Meteorite Bore Member; 2) one or two glacial events of the Meteorite Bore Member cannot be correlative to the 2.26-2.22 Ga glacial event in South Africa since they are 50 Ma older than it, implying that at present there is no correlative unit worldwide to this glacial diamictite; 3) If these diamictites are correlative to the 2.26-2.22 Ga glacial event (despite this age difference), it would be even more problematic since carbonates above this glacial do not record the Lomagundi excursion, which must have followed shortly after the 2.26-2.22 Ga glacial event. With all these uncertainties screaming from this new correlation, one could clearly see that the geologic and geochronologic framework that authors subscribed does not fit with the global records that were established over the last 2 decades.

1) this is not correct only one glaciogenic diamictite is missing the Bruce Formation in the Huronian and the Basal Duitschland in the Transvaal.

Indeed, our data show that the model for the GOE and associated glacial events is not correct. Recent dating by Gumley et al (2017) of the Ongeluk Volcanics showed that the Makganyene glacial deposit is ~ 100 Myr older than previously estimated (between 2.46 and 2.43 Gyr), an age similar to Ramsay Lake and Boolgeeda Iron Fm (see above comment on the zircon U-Pb age of the Boolgeeda diamictites). Our new Re-Os and U-Pb age for the MBM implies that it must be ~2.31 Gyr old or younger. The MBM can be correlated within error to the Rooihogte Fm (2310 ± 9 Ma; Rasmussen et al. 2013) and possibly the Gowganda Fm, which is older than 2308 ± 8 Ma (Rasmussen et al. 2013). Considering the large number of unconformities, additional data in South Africa and Canada are required, however, before a clear picture of the chronology of deposition of glacial deposits can

emerge. Specifically, controversies remain concerning the number and age of glacial deposits in South Africa. Some authors consider three diamictites at the top of the Timeball Hill Fm., base of the Rooihoogte Fm. and base of the Duistschland Fm. (e.g., Rasmussen et al., 2013 and references therein). Others consider only two glacial events diamictites at the top of the Timeball Hill Fm., and base of the Duistschland Fm. (e.g., Hoffmann et al., 2013, Luo et al., 2016, and references therein; correlation used in this paper, Figure 1).

Another critical shortcoming of this manuscript is that the nature of studied sulphides was not described in sufficient detail. The study cover over 3.5 km of succession largely lacking organic rich shales, sulphides hosted in these lithologies could have strong contribution of detrital, hydrothermal, and authigenic signals. Detrital pyrite was described from this succession (see references above), dolerite sills and metamorphism affected this succession so hydrothermal sulphides are indeed expected and indicated by limited petrography described in SOM. In this case, general statements made in SOM on the origin of sulphides are not convincing, grain-by-grain analysis is required. Figures in SOM do not show convincingly authigenic pyrite.

We obtained more than 250 thin sections from the 3 drill cores and performed detailed petrographic observations. Metamorphic overprint is low greenschist facies conditions. As demonstrated by our new trace element abundances of pyrite there is no evidence of fluid circulation/hydrothermal overprint throughout the cores. Dolerite emplacement did not affect the layering nor mineral assemblage (see comment of Supplementary Figure 1b below). This is the first detail study of the petrography of the Turee Creek Group.

There are many statements in the manuscript that are either wrong or misrepresenting literature. I provided detailed comments below. In summary, considering a number of uncertainties involved in this study, I feel re-definition of the GOE is not yet warranted.

Lines 29-30: GOE is bracketed by loss of MIF-S and end of the Lomagundi Event at ca. 2.1 Ga (see Holland, 2002).

It is old fashioned to use the end of the Lomagundi to bracket the GOE. Since the early 00', the rise of atmospheric oxygen is considered to be best constrained by the disappearance of the mass independent fractionation of sulfur isotopes, that is between ca. 2.45 and 2.32 Ga.

We changed 2.45 and 2.2 billion years into 2.50 and 2.32 billion years to clear things out and to account for the comments of Reviewers 1 and 3 (see below the comment line 49-51).

Lines 32-33: What is this statement based on? Check Gumsley et al. (2017) paper in PNAS

The paper of Gumsley et al. (2017) was not published when we send our ms for review. This paper does not provide indication on S isotopes only a new age of the Ongeluk Volcanics, which implies reconsidering the age of the Makganyene glacial event. GOE is defined on the basis of S-isotope data, not age of glacial events.

Lines 34-36: Spans 230 Ma? Based on what? It is a foreland basin filled with turbidites, do you know any other foreland basin with 2-3 km of turbidites spanning 230 Ma? This is not geologically reasonable.

The Turee Creek Group is partly composed of siliciclastic rocks (mudstones, siltstones and sandstones) but contains abundant carbonates and diamictites. Sedimentation rates used in this study are from Trendall et al (2004) estimated on carbonate and siliciclastic sediments from the underlying Hamersley Group. The model of Krapez et al (2017) is one model among many others. See review of the different geodynamic settings proposed for the Turee Creek Group in Van Kranendonk et al. (2015).

Lines 36-37: sulphide is not deposited, it is not clastic in origin.

We refer here to a geochemical signal (MIF-S sulphide)

Lines 37-39: Absence of MIF can be explained by several processes so unique interpretation based on S isotope data alone is not possible. Sulphate aerosols would be a minor flux in oxic atmosphere, where oxidative weathering would dominate.

We added a paragraph on the significance of non MIF-S intervals lines 163-175 (see comment Line 159-161 of Reviewer 1 above).

Lines 39-41: How do you know that small MIF-S reflects oxidative weathering? What if it reflects production in the atmosphere?

This is explain in the discussion section :

« Alternatively, one may argue that MIF-S of sediments record photochemical processes under anoxic conditions, i.e. much like during the Archaean, yet with likely changes in atmosphere composition to account for the low amplitude $\Delta_{33}\text{S}$ values. This is unlikely, however, because it would imply asynchronous oxygenation of the atmosphere. This model would also fail in accounting for the MDF slopes of -7 between $\Delta^{33}\text{S}$ and $\Delta^{36}\text{S}$. »

Line 41-42: Low sulphate between 2.45-2.22 Ga? But there is good evidence for 2.32 Ga sulphate evaporites in the Gordon Lake Formation of the Huronian Supergroup so sulphate was high at 2.32 Ga if not earlier.

This is a relative estimate compared to modern day to explain the variability of the amount of sulphate delivered to the ocean during this transitional period associated with the rise of atmospheric oxygen to value comprised between 10^{-5} to 10^{-2} PAL, that is at least 100 time lower than present day.

Line 45-46: really? Redox chemistry was dominated by CO₂? What about all other reducing and oxidizing radicals and molecules? See Pavlov and Kasting, 2002.

We wrote was dominated by carbon dioxide and methane (Kasting, 1993). Methane (CH₄) is a reducing gas. These are the two main species considered in most atmospheric models, including that of Pavlov and Kasting (2002).

Lines 49-51: I am confused who said that MIF disappeared by 2.2 Ga ago? There are multiple papers arguing that it disappeared before 2.32 Ga. What am I missing here?

Following Reviewer 1 and 3 comments, we deleted the sentence

~~The global disappearance of significant MIF-S by 2.2 Gyr ago is attributed to a major increase in $p\text{O}_2$ (to above 10^{-5} to 10^{-2} PAL) coupled with a decline in atmospheric methane~~

and added :

The rise in atmospheric oxygen to above 10^{-5} to 10^{-2} PAL coupled with a decline in atmospheric methane⁵⁻⁷ is loosely constrained between ~ 2.50 Gyr, the age of the youngest sediments with no, or only minor $\Delta^{33}\text{S}$ anomalies (top of the McRae Shale Formation and Whaleback Shale Member of the Brockman Formation, Hamersley Group^{8,9}), and ~2.32 Ga, the age of the youngest rocks with strong MIF-S (Rooihogte and Timeball Hill formations, South Africa¹⁰⁻¹²).

Lines 51-54: Sure, but these papers clearly established age of this transition at ca. 2.32 Ga.

Yes they do, but we show that this is not correct.

Lines 59-62: These papers also constrain global oxygenation to be older than 2.32 Ga.

Before the paper of Gumsley et al (2017), several authors considered that only the 2.3 to 2.2 Gyr old glacial event (Makganyene diamictites) was global (Snowball Earth, see Evans et al. (1997) and Kirschvink et al. (2000) for example).

Lines 67-70: This statement is based on poor understanding of geology.

It states that the Turee Creek Group contains continuous record of 230 Ma years. Just take a realistic sedimentation rate and calculate thickness expected for 230 Ma of deposition.

Martin et al (2000) proposed that sedimentation of the Turee Creek Group was a long-lived process spanning ~250 Myr. Trendall et al. (2004) defined a variety of sedimentation rates for the underlying Hamersley Group sedimentary successions such as: Shales, 5m/Ma, carbonates 12m/Ma, BIF 180 to 225 m/Ma. These are well constrained rates using a large dataset of U-Pb ages. These are realistic constraints provided by renowned geologists. Their interpretations are in good agreement with our results.

Now, think that most of the Turee Creek Group consists of turbidites that have high sedimentation rate.

The Turee Creek Group comprises abundant carbonate in Kazput (more than 400 meters), abundant mudstone (more than 500 meters) in Kungara and more than 500 meters of diamictites. Some siliciclastic sediments can be attributed to turbidites. There is abundant mudstones similar to the one described in the underlying Hamersley Group. This together with the sedimentation rates proposed by Trendall et al (2004) for the Hamersley Group clearly show that the « turbidite model » does reflect geological observations.

Add to this that it is retro back-arc basin with a short duration on the order of 20-30 Ma.

This is the model defended by Krapez (1996), Krapez et al., (2017), which is one among many others (see for example Martin et al (2000) who argued for a long-lived (250 Ma) foreland basin, the McGrath Trough). The age constraints of Krapez et al (2017) are not statistically representative. We have clear age data showing that there is 2.34 Gyr old detrital zircons in the MBM diamictites from the Turee Creek Group. The reason for this discrepancy is likely due to the statistical relevance of the analyses and the fact that our data is based on detailed petrographic analyses of drill core samples, which preserved pristine sedimentary features.

For the sake of comparison, Krapez et al. have analyzed 6 samples from the TCG and obtained a total of 311 detrital zircon analysis. The rocks analyzed include 2 quartzites unit overlying the Kazput Formation (quartzite 1 and 2; not shown in our paper but see Krapez et al., 2017), the quartzite forming the Koolbye Formation and the sandstone bar underlying the MBM diamictites. Note that there is major controversies on the geological meaning of Quartzites 1 and 2, which may not be belonging to the TCG, rather to the overlying BRQ.

In contrast, we have investigated 18 samples spanning the entire TCG for a total of 1513 detrital zircons analysed. The samples are not from major quartzites units but from minor sandstone layers in diamictites from the MBM and the Boolgeeda Iron Formation (both at the Hardey Syncline and the Boundary Iron Formation), minute detrital layers in carbonates, placers in carbonate microbialites at the base of the MBM. Statistically we show that the range of zircon ages is much wider than the one reported by Krapez et al (from 4 to 2.34 Ga). Using Lu-Hf systematics obtained on zircon showing concordant ages (> 500 analyses) we also show that the source of the zircons is heterogeneous and therefore not local, which can be best explained by a long sedimentation process of 200 to 250 Ma. These results are part of two papers by Caquineau et al., one recently submitted to Precambrian

Research and another one in preparation.

Below are the age results obtained on diamictites from the MBM (same samples used for Re-Os geochronology; concordant minimum age of 2.34 Ga) and the Boolgeeda Iron Formation (both at the Boundary Ridge and Hardey Syncline Localities; concordant minimum age of 2.45 Ga). This clearly implies that there is a minimum of 100 Myr between sedimentation of the Boolgeeda Iron Formation and the MBM diamictites, that is half of the Turee Creek succession. This clearly demonstrates that the duration of sedimentation cannot be 20-30 Ma, rather ~200 Ma or more.

Left : Detrital zircon U-Pb age from the MBM diamictites showing a concordant minimum age of 2.34 Ga. Right : Detrital zircon U-Pb age from the Boolgeeda Iron Formation diamictites showing a concordant minimum age of 2.45 Ga, in agreement with the age of the underlying Woongarra Rhyolites.

Finally, none of the cited references argue for 3 glacial cycles. Reference 23 is not relevant. Reference 21 argues for one glacial cycle. Reference 22 argues for 2 glacial cycles. How did you get 3? Finally, read Krapez et al., 2017 for tectonic setting and duration of the Turee Creek Group. Neglecting published literature is a bad habit!

We modified the sentence

“is characterised by a continuous marine sedimentary succession spanning the GOE and up to three glacial events²¹⁻²³ (Fig. 1; Supplementary Figs. 1, 2; Supplementary Information, ref. 21-23)”

into

“is characterised by a continuous marine sedimentary succession spanning the GOE and two²¹ or possibly three²² glacial events (Fig. 1; Supplementary Figs. 1, 2; Supplementary Information, ref. 21,22).”

It is to be emphasized here that we were not aware of Krapez et al (2017) paper when submitted our paper. Our paper was submitted January 17, 2017 that is several months before the publication of Krapez et al (2017). Krapez et al paper 2017 was received January 23, accepted for publication March 5, and available online 18 March, 2017. Note also, that the review of our paper was delayed by 3 months due to a late reviewer...

Line 74: Be specific: most clastic rocks of the Kungarra Formation are turbidites with high sedimentation rate.

Our aim here is not to present the detail of the stratigraphy of the Kungarra Fm. Below is a summary of the lithologies present in the Kungarra Fm after Van Kranendonk et al (2015). These are clearly not representative of a Bouma sequence. Turbidites are indeed present but represent only part

of the clastic sediments.

Table 1
Facies summaries and interpretations, Kungarra Formation, Turee Creek Group.

Facies	Description	Interpretation
Facies association 1		
Facies A	Chert-ferruginous chert interbedded with greenish-brown shale	Deepwater chemical precipitate
Facies B	Massive mudstone with occasional siltstone interbeds	Off-shore deposit (below wave base)
Facies C	Massive fine-grained sandstone interbedded with mudstone with lower sharp, and upper gradational, contacts	Off-shore deposit; rapid deposition from turbulent suspension (Lowe, 1975; Bose et al., 1997) below wave base
Facies D	Massive to parallel laminated fine-grained sandstone; sandy laminae are bounded by very thin double mud layers	Tidally influenced offshore turbidite (cf. Shanmugam, 2003; Mazumder and Arima, 2013) formed below wave base
Facies E	Fine-grained sandstone with current ripples	Off-shore deposit formed below wave base
Facies association 2		
Facies A	Fine-grained, well-sorted, symmetric to near-symmetric rippled sandstone	Wave agitated shallow-marine deposit (cf. De Raaf et al., 1977; Johnson and Baldwin, 1996)
Facies B	Fine-grained, massive to parallel-laminated sandstone; bed tops bear wave ripples	Wave reworked shallow-marine deposit
Facies C	Medium-grained sandstone with convolute lamination	Storm influenced shallow-marine deposit (Johnson, 1977; Bose, 1983; Leeder, 1999)
Facies D	Medium-grained hummocky cross-stratified (HCS) sandstone; no wave reworking on top of HCS beds	Storm deposit formed between storm and fair-weather wave bases (cf. Bose et al., 1997)
Facies E	Very fine-grained muddy sandstone with climbing ripple-lamination, double mud drapes and combined flow ripples	Tide-wave interactive sub-tidal deposit
Facies F	Coarse-grained large-scale cross-bedded sandstone with shore-parallel paleocurrent	Longshore bar deposit
Facies G	Massive and/or parallel to ripple cross-laminated carbonate with occasional desiccation cracks	Intertidal deposit to beachrock
Facies H	Stromatolitic carbonate; characterised by domical stromatolites consisting of crinkly microbial laminations	Shallow-marine deposit
Facies association 3		
Facies A	Thickly bedded, massive, matrix supported sandstone with randomly oriented subangular to subrounded clasts with faceted and striated faces; outsize clasts are common; conglomeratic at places	Glacial diamictite (cf. Martin, 1999)
Facies association 4		
Facies A	Thinly bedded, dark green mudstone with outsize clasts (up to 30 cm)	Glacial diamictite formed by melting of floating ice sheet (cf. Martin, 1999)
Facies B	Quartz-rich sandstone with angular to well-rounded outsize clasts; large-scale slump structures	Glaciogenic deposit reworked by turbidity current
Facies C	Conglomerate with subrounded to moderately well rounded predominantly carbonate clasts; some clasts display a clear penetrative fabric into underlying mudstone	Glaciogenic deposit reworked locally by mass transport process
Facies D	Pale cream, fine-grained carbonate (calclutite) with very fine-scale bedding; fine dolomite rhombs embedded in a silty matrix; overlies conglomerate facies C	Carbonate platform deposit exposed during glacial retreat
Facies E	Polymictic conglomerate/pebbly sandstone; characterised by well-rounded to subangular clasts; sandstone locally displays rhythmic bedding	Conglomerates are sediment gravity flow deposits and the sandstones are turbidites
Facies F	Medium-grained, massive, quartz-rich sandstone overlying the conglomerate	Shallow-marine deposit formed during sea level fall at higher flow regime
Facies association 5		
	Mudstone interbedded with thin units of ferruginous chert, an Mn-rich ferruginous unit, and beds of calcarenite conformably overlie glaciogenic sedimentary rocks of the Meteorite Bore Member	Relatively deep water deposits, below the storm wave base

Lines 78-79: This is not correct and biased representation of published literature. See Mueller et al., 2005 Geology paper (with comments and reply), Krapez et al., 2016 Precambrian Research, and Krapez et al., 2017 Precambrian Research. It is clear that 2.2 Ga age does not constrain deposition but provides a maximum age, which by 200 Ma older than the depositional age.

We changed

«and another U-Pb zircon age of $2,209 \pm 15$ Myr from a volcanoclastic breccia in the Cheela Springs Basalt »

into

“and a Pb-Pb baddeleyite age of $2,208 \pm 10$ Myr from a dolerite sill intruding the Meteorite Bore Member in the Hardey Syncline³⁰”

Lines 79-80: This reference is an abstract that present no actual data. Alternatively, there is detailed published study of detrital zircon ages from the Turee Creek Group (Krapez et al., 2017, Precambrian Research) that shows no detrital zircon ages younger than ca. 2.43 Ga. Again, you need to reconcile

these data, rather than neglecting them.

We changed

Detrital zircons from the MBM indicate a maximum age of deposition of ca. 2,340 Myr for the diamictites

into

Detrital zircon U-Pb ages from quartzite horizons of the Turee Creek Group have led Krapez et al., (2017)²² to suggest that the Turee Creek basin was closed by about 2,430 Myr. However, new U-Pb age constraints from detrital zircons collected at the base of the MBM indicate a maximum age of deposition of ca. 2,340 Myr for the diamictites³¹. This together with the occurrence of a ~1500 meter-thick sequence above the MBM strongly support the existence of a long-lived basin (~2450-2200 Myr) spanning at least 250 Myr of sedimentation³².

Lines 80-96: How do you know that these pyrites are diagenetic in age?

See above comments

What do you mean by ‘at the base of the MBM diamictite and its underlying sandstone and carbonate stromatolite?’ Below all of these lithologies? Rephrase. If pyrites and mudstones are below all these lithologies, this is not the first age obtained from Paleoproterozoic glaciogenic sediments.

We changed

at the base of the MBM diamictite and its underlying sandstone and carbonate stromatolite

into

at the base of the MBM diamictite and underlying sandstone and carbonate stromatolite

Unfortunately, this age is not geologically meaningful. It would place the Meteorite Bore Member to be younger than all 3 Huronian glaciations, but much older than the last Paleoproterozoic glacial event constrained between 2.26 and 2.22 Ga in age.

This is a well constrained age using two different methods. The consequence of this age is that the short-lived basin model of Krapez et al (2017) is incorrect. This is the fate of models, to evolve as more new data are published (see for example the recent paper of Gumsley et al (2017) who showed that the geological model of the Griqualand West Basin was wrong based on a new age constraint for the Ongeluk Volcanics overlying the Makganyene diamictites).

Furthermore, depositional rate of 11 m/Myr for ~1500 m of Kungarra turbidites in a foreland basin is not realistic. The Turee Creek Group was deposited in a separate basin from the Hamersley Group so depositional rates for BIFs (deposited in deep-water setting starved of clastic input) are not relevant to depositional rates for turbidites in the foreland basin.

Depositional rates for BIFs (~180 m/Myr) are different from carbonate (12 m/Myr) and shales (5 m/Myr). There is abundant carbonates and shales in the Turee Creek Group arguing for a long-lived sedimentation record of ~ 250 Myr as previously proposed by numerous authors (e.g., Martin et al, 2000).

What do you mean by ‘the newly identified diamictite horizon recognized here within the upper part of the Boolgeeda Iron Formation?’ This diamictite horizon was identified by Krapez, 1996; Martin et al., 1999; and co-authors of this manuscript in earlier publications. And, yes, it cannot be correlated with the MBM since it is stratigraphically lower than it. It is as simple as this. Swanner et al. (2013) never implied that this diamictite reflects older glacial event, they simply stated that it is older in age.

Depositional rate of 280 m/Myr was not inferred for the Boolgeeda Iron Formation (Trendall et al., 2004), I do not see this number in this paper. How does your age (which most likely does not constrain deposition) ‘indicates that the uppermost Kungarra shales, Koolbye quartzites and Kazput carbonates were continuously deposited until about 2.25 to 2.20 Gyr ago?’ This section lacks basic geological logic!

Several points in this comment were already addressed in large extend above and will not be reconsidered here.

This is the first time that diamictites in the Boolgeeda Iron Formation are reported at the Hardey Syncline locality. Previous description were made at Duck Creek Syncline (Boundary Ridge) and Yeera Bluff localities. The occurrence of diamictites at the Hardey Syncline locality demonstrates that there is an older glacial event predating the MBM event. With the exception of Swanner et al (2013), previous authors linked the Boolgeeda diamictites of the Duck Creek Syncline and Yeera Bluff localities to the MBM.

Indeed, Trendall et al (2004) report a depositional rate of 225 m/Myr for the Boolgeeda Iron Formation not 280 m/Myr. We made a confusion mixing the rate estimated for the Weeli Wolli Fm. (180 m/Myr) and that of the Boolgeeda Iron Formation (225 m/Myr). Note that this does not change the conclusion of the paper.

Lines 99-105: There is no discussion what lithologies and types of pyrite were analyzed. This is critical since some lithologies are coarse-grained and might contain detrital pyrite as mentioned in Krapez et al. (2017). There are sills and dikes in the Turee Creek Group and it was deposited in the foreland basin – hydrothermal fluids were moving through the sediments. Finally, some fine-grained lithologies might contain authigenic pyrites. Only the latter would be relevant to the atmospheric evolution.

All pyrite studied here are early diagenetic and in some cases (green siltstone of the Boolgeeda Iron Fm.) likely authigenic. None of the pyrite analyzed here are detrital. There is no evidence of fluid circulation attending dolerite emplacement (see comment below) nor regional metamorphism.

Lines 107-108: Most of sulphides in the Boolgeeda IF as in other iron formations are not primary in origin – water column and sediments had oxidizing redox inconsistent with sulphidic conditions.

We do not agree with this. See above comments.

Lines 118-119: Near the Archean-Proterozoic boundary? Really, it is more than 50 Ma younger!

Lines 119-124: What is the origin of pyrite in green siltstone? It is not common for organic matter lean samples to have such high S and pyrite content. Is this chert? What was driving S reduction in absence of organic matter? It is not pyrite precipitated from water column since water column was ferruginous and not sulphidic during deposition of the IF.

See above comment (Reviewer 1, line 163-175).

Lines 125-127: You can also say that there was enough oxygen for oxidative weathering on the continent to deliver sulphate. Why not? Or that large amount of hydrothermally derived sulphate was flushed to the basin. All these are possibilities, why do your prefer one of them?

We discuss the role of oxidative weathering in the text. There is no evidence for hydrothermally-derived sulphate in the Turee Creek as indicated by the chemical composition of pyrite.

Lines 131-133: How limited availability of organic matter would result in unfractionated S isotope values? This must be wrong.

This is not wrong but the basis of the activity microbial sulphate reduction which are heterotrophs. Organic matter availability is a key factor when interpreting S-isotope systematics (see the recent paper of Crowe et al., (2014) published in Science + abundant references on the subject therein).

Lines 133-137: This is very vague. Actually neither depth nor change is a process.

We replaced processes by factors

Lines 137-141: What about well-known process of sulphidization of iron formation by hydrothermal fluids?

High sulphur contents in the Boolgeeda is localized and restricted to pyrite layers lining the stratification in green siltstone and mudstone/siltstone from the diamictite horizon. Fluid processes would affect different lithologies and not be restricted to layer-parallel features, which are typical of authigenic pyrite. The trace metal abundances of pyrite show that they were not derived from hydrothermal fluids.

Lines 167-169: How do you know that rims formed during diagenesis vs. via circulation of hydrothermal fluids deriving S by dissolution at stratigraphically lower or higher levels?

Pyrite chemical composition tells us that there is no fluid circulation in the Boolgeeda BIFs. Fluid circulation are characterized by specific features (veinlets, pervasive penetration along schistosity plane...). None of these features are present in the drill core samples.

Lines 176-182: Why it indicates atmospheric oxygen? You have organic matter poor lithologies and these rarely express large MIF-S signal. Why it would imply asynchronous oxygenation of the atmosphere?

A MIF-S signal is independent of organic matter per definition. We don't know of any literature discussing the link between organic matter abundance and MIF-S signal. Strictly speaking, if the MIF-S signal reflects a photolytic process, than the occurrence of a variety of sedimentary intervals of different ages in different continents should indicate that the rise of oxygen is asynchronous (i.e., not occurring at the same time).

Lines 189-190: Reference 41 is not relevant here.

Reference 41 has been deleted and replaced by Hoffman (2013)²².

Lines 193-198: D33S/D36S systematics was not discussed in ref. 16 and refs. 17 and 18 are abstracts so they did not present data.

These are minor details. We can modify the sentence if requested by the Editor, but this will affect the style. The abstracts 17 and 18 are discussing $\Delta^{33}\text{S}/\Delta^{36}\text{S}$ systematics.

Line 204: The Duitschland Formation is not correlative to the first Huronian glacial event.

Some authors have suggested so (see Williford et al., 2011) for example.

Lines 205-206: One cannot correlate an event with ~2.31 Ga age to another event younger than 2.26 Ga since they are different by more than 50 Ma in age.

We tune down this interpretation and wrote :

ii) the two major glacial horizons forming the MBM are 2.31 Gyr in age or slightly younger and can be temporally linked within error to the Rooihogte Formation ($2310 \pm 9 \text{ Ma}$)¹⁹ and possibly the

Gowganda Formation, which is younger than $2308 \pm 8 \text{ Ma}^{19}$.

Lines 206-208: This is truly pathetic. Did you read references in your reference list? Luo et al. (2016) argued for loss of MIF at $\sim 2.33 \text{ Ga}$ in their title! What is asynchronous?

*The core of this paper is to show that the disappearance of MIF-S in the sedimentary signal IS DECOUPLED from the ATMOSPHERIC SIGNAL. The core of Luo et al. data is to say that the disappearance of the MIF-S signal in a 3 m thick sedimentary section is linked to the rise of O₂. But we show that we have at least 2 and possible 3 absence of MIF-S signal in the 700 meters of core investigated. So the disappearance of a signal in 3 meters of core is **NOT RELEVANT** to the rise of atmospheric oxygen on Earth!*

Line 212: There is still anomalous sulphur source at Earth's surface now. You should be more careful in phrasing.

Anomalous sulphur sources are found in ice cores due to large volcanic explosion. These are atmospheric issues not related to oceanic environment where they are diluted.

Lines 224-226: You are inconsistent here with your own definition. Luo et al., (2016) showed D33S up to +4 and +6 permil at 2.33 Ga, if so can you define GOE at 2.45 Ga and explain these data by recycling considering that are from deep-water, open-marine setting?

Luo show about 20 analyses greater than 1‰. ‰. Again, we emphasize that most of the data from Luo et al., 2016 are from sandstone and therefore it could be detrital pyrite that gives the large $\Delta^{33}\text{S}$ in their dataset. All the data of Guo are lower than 1.6 ‰. This is not statistically relevant and may reflect that some pyrite are not fully dissolved into sulphate, which is to be expected in an atmosphere containing 1000 less oxygen than today (see above comment), where surface waters could also still be anoxic in many cases.

Lines 226-228: This is odd. How do you imagine this change? Positive D33S values always have matching negative D33S values somewhere. Mixing and recycling can result in diluted or smaller-amplitude D33S signal.

No, the crux of the $\Delta^{33}\text{S}$ signal is that it is not balanced. There is much more positive values recorded than negative ones. This has been one of the main question discussed by Farquhar and co-workers (2000, 2001, 2003....). Mixing can explain many things but this requires documenting significantly positive and significantly negative MIF-S reservoirs, which is not the case.

Lines 228-230: What data? How do you know that your samples have this age?

See above

Lines 230-232: Reference 46 has nothing to do with the age or global nature of the Lomagundi Event, this reference argues for the lacustrine origin for this anomaly. Reference 47 is 6 years older than the first use of the GOE acronym.

We replaced Schidlowski et al (1976) by Schidlowski et al (1975). We refer to the updated datation of the GOE by Karhu and Holland (1996).

Line 233: What is geologic marker?

Marker of oxidation that are geologically relevant (e.g., red beds), i.e., different from a geochemical marker such as MIF-S.

Lines 239-242: Typically, samples less than 1-2 m apart are analyzed with the Re-Os method. You analyzed samples 96 m apart, which cannot have the same initial Os ratio. How do you know that you did not obtain errorchron without geologic meaning? Why method developed by Selby and Creaser for organic-rich shales was not used? This method specifically attacks authigenic Re and Os. In your case, you have also contribution of detrital Re and Os.

The initial $^{187}\text{Os}/^{188}\text{Os}$ value in our study (0.14 ± 0.03) is close to that of mantle/extraterrestrial value, which is only slightly higher than the majority of black shales between 2.7 and 2.0 Ga ($n = 5$) with initial osmium indistinguishable from the mantle value of 0.11 (Anbar et al., 2007; Hannah et al., 2004; Hannah et al., 2006; Yang et al., 2009; Kendall et al., 2013). Despite the large interval of sampling, we consider that our initial Os isotopic composition largely reflects that of seawater in the Late Archean – Early Proterozoic period. It may be related to the fact that most of the Re and Os in samples studied are scavenged in microscopic syndimentary pyrite, which preserve the composition of contemporaneous seawater. New Re-Os ages have been obtained using the Selby and Creaser method, which provided the same age as obtained in the first version of our ms (see above).

Lines 279-281: Do you present here these analyses? If not, why to mention?

This is the in situ S-isotope data which form the core of this study.

Line 287: do you mean ‘acid-volatile or elemental sulphur’?

acid-volatile of elemental sulphur

Lines 306-309: Ok so you have sulphides with transition metals, pentlandite, chalcopyrite, and galena. These are hydrothermal minerals. Why they are not discussed in the main text?

These represents minor amount of sulphides. Among the 250 thin sections investigated we were able to identify only few. These are minor and not representative of any hydrothermal circulations.

Some of D33S vs D36S diagrams have so much data that clearly data would fit on the ARA and MDF lines.

Indeed there is a lot of data, more than 1000 analyses. This is why this study is statistically representative.

Supplementary Materials

Lines 38-39: Martin et al., 1999 argued for only one glacial event.

Yes, the MBM although he also says that the diamictite in the Boundary Ridge is within the Boolgeeda BIF, thus implying that it cannot be the MBM.

Lines 40-44: What is the evidence for this diamictite at the Boundary Ridge locality have glacial instead of debris flow origin?

See Williford et al 2011. There is clear evidence of dropstone features there.

Lines 48-50: I am totally confused now. How does this statement correspond with lines 202-204 in the main text arguing for irreversible rise of atmospheric oxygen at 2.45 Ga or earlier? I presume that it means during deposition of the Boolgeeda IF? If so, why your data argue against the same earlier interpretation?

The Supplementary Information deals with the age of the MBM and Boolgeeda diamictites and the main text deals with age of the rise of oxygen, which are different things, obviously. To clear things

out we modified the sentence in the Supplementary Information:

Accordingly, our data argue against the interpretation that the disappearance of MIF-S anomalies and occurrence of strongly depleted $\delta^{34}\text{S}$ values identified at the top of the Boolgeeda IF at the Boundary Ridge locality represents the GOE.

into

Accordingly, our data argue against the interpretation that the disappearance of MIF-S anomalies and occurrence of strongly depleted $\delta^{34}\text{S}$ values identified at the top of the Boolgeeda IF at the Boundary Ridge locality represents the MIF to MDF transition as it was understood in Williford et al., (2011)⁸.

In the main text we modified are younger than ~2.31 Gyr into are ~2.31 Gyr old or slightly younger

Supplementary Figure 1a Poorly scanned figure with a wrong location for the Boundary Ridge.

The location of the Boundary Ridge is taken from Martin (1999) and Van Kranendonk et al (2015).

Figure 1b: The section is bounded by two thick \rightarrow dolerite sills. Would not you expect some hydrothermal fluid circulation and contribution to sulphides in sediments? This is the section from which Re-Os data came.

The figure below shows the contact between Kungarra siltstone and a dolerite sill from TCDP2 core. The contact is marked by a mm-scale layer (large red arrow), which did not disturb the sedimentary layering nor the two sedimentary pyrite grains (small red arrows). This clearly indicates fluid-absent conditions during emplacement of the dolerite sill. More generally, there is no evidence of fluid-induced metamorphic processes in the Turee Creek Group. Quartz and calcite veins are locally present in BIFs and siliciclastic successions, usually in association with late stage brittle faulting. The drill cores studied here were obtained in zones away from such faults.

Supplementary Figure 2: nothing distinctly glaciogenic on these figures.

See above discussion

Supplementary Figure 3: It would be useful to label with different symbols pyrite and mudstone samples.

This is irrelevant here

Lines 121- 134: This is an oversimplified argument. People argued for ages about placer pyrite deposits. Sulphide layers parallel to bedding could be hydrothermal as well as finely disseminated crystals. So there are overgrowths related to hydrothermal or metamorphic processes, this is relevant to discussion in the main text on lines 167-170.

Our new trace element data show that there is no hydrothermal-metamorphic overprint. This is also supported by basic observations of petrographical thin sections, which we did perform throughout the Turee Creek Group.

Lines 134-141: What is the source of this sulphur and what reductant was used in absence of abundant organic matter in this sediment? Presence of Fe-oxides and Fe-chlorite indicates late sulphidization. Why it cannot form by later processes? In fact, it seems to be the most logical explanation for me.

Local remobilisation of pre-existent small pyrite crystals is the most viable hypothesis for the formation of overgrowth zones and coarser pyrite grains (see discussion in Supplementary Information). This is supported by pyrite core-rim trace metal ratios and sulphur isotope compositions. Organic matter content is typical of Archean environment (~2000 ppm). Organic matter must have been locally much more important as indicated by strongly ³⁴S-depleted pyrites.

Figure 5: None of these pyrites look convincingly diagenetic to me.

Our textural description and trace element analysis clearly show that pyrite are early diagenetic or syngenetic.

Reviewers' comments:

Reviewer #2 (Remarks to the Author):

I am satisfied with the revised manuscript, and look forward to seeing it published in Nature Communications. In particular, the inclusion of additional information with respect to sulfide generations and petrogenesis is a welcome addition.

Reviewer #4 (Remarks to the Author):

Reviewer #3 raised a number of concerns about the authors' interpretation that the deposition of the Turee Creek Group was long-lived (250 Myrs). I think the authors have done a good job responding to these concerns. I find the argument that deposition continued until after 2.31 Ga to be compelling. But I think the argument that deposition occurred over 250 Myrs (2.45 to 2.2 Ga) to be a bit of an overstatement.

The authors provide an age of 2.31 Ga for rocks deposited prior to and during the early stages of the MBM. They provide an age of 2.2 Ga for a sill intruding the MBM. They then use the thickness of overlying sediments to argue that deposition occurred until 2.25 Ga. I don't think this is very compelling; there are a 100 Myrs missing here. I think all the authors can really say is that the MIF S memory effect was preserved in rocks younger than 2.31 Ga. The authors use sedimentation rates for the underlying Hamersley Group to estimate that deposition continued until as late as 2.25 Ga. But I don't think this argument is strong enough to use 2.25 Ga in the abstract, for example. I think that in addition to using rates from the Hamersley Group to estimate time they should use a typical Phanerozoic rate in order to come up with a conservative minimum age for the end of deposition. But most importantly, I think this is an unnecessary distraction from the S story. The significance of the preservation of the MIF memory effect past 2.3 Ga is enough to make this data set important. The authors don't need to push it back to 2.25 or 2.2 Ga to make it important.

Reviewer #3 also raised questions about the Re/Os methodology and the authors have addressed those concerns as well. The sampling approach (i.e. selecting samples separated by 100 m), also mentioned by #3 is not really addressed. The Re/Os system is outside of my expertise and I don't know how this sampling protocol could potentially undermine the reported age. But I think it would behoove the authors to address this concern. This date is very important to their argument that the MIF memory effect persisted until after 2.31 Ga.

Reviewer #3 also suggested that the flux of sulfate aerosols would not be sufficient to keep up with sulfide weathering under oxic conditions. The authors do not address this in their response or in the revised text. But I think this is a good point. It doesn't seem possible to me that this volcanic flux would be sufficient to have a meaningful effect on the global sulfur budget once sulfide weathering commenced. I think the presence or absence of the MIF memory effect following the end of MIF production is controlled by the relative contributions to seawater sulfate of Archean sulfides carrying MIF and those that don't. Not all Archean sulfides will carry a MIF signal. Igneous sulfides, for example, would not.

My main concern with this manuscript is that it lacks clarity and I think that makes it difficult to determine exactly what the main points are. For example, there is an important difference between the end of MIF production when O₂ passed the 0.02 % PAL threshold and the preservation of MIF, in the form of MIF memory effects, afterwards. But the authors don't make this distinction. Consider this sentence from the introduction:

"The redox chemistry of the atmosphere before about 2.45 Gyr ago was likely dominated by carbon

dioxide and methane with $pO_2 < 10^{-5}$ Present Atmospheric Level (PAL), as shown by the occurrence of MIF-S anomalies..."

Here the presence of MIF is tied directly to very low O_2 . Compare to this sentence from the abstract:

"This extends the sedimentary record of MIF-S to younger than 2.31 Gyr by at least 60 Myr, and clearly establishes the MIF-S signal as asynchronous between North America, Australia and South Africa."

Here the authors are talking about the MIF memory effect but they don't say this explicitly. This is important because not only is it confusing, it also makes it difficult to compare this study to that of Bekker et al., Guo et al., and Luo et al. These studies were looking for the absence of large MIF reflecting and end of MIF production and not for evidence of a MIF memory effect.

I guess everyone would agree that large MIF was no longer being produced at 2.3 Ga (at the completion of the GOE as commonly defined; e.g. Lyons et al., 2014). But MIF was still being preserved locally due to the memory effect. That is what this study demonstrates. Local and long-lived preservation of the MIF memory effect. The fact the Rooihogte/Timeball Hill shales, which are 2.3 Myrs old, don't show this memory effect is proof of its localized nature.

If the MIF memory effect is one main point, then the other one is the fact that this study identifies highly depleted $d_{34}S$ pyrite at 2.45 Ga. This is very interesting and significant. The lack of MIF does not uniquely define an oxic atmosphere (the Archean Array passes through or very near the origin in $d_{34}S$ - $D_{33}S$ space) but I think pyrite S approaching -40 per mil is strong evidence for high sulfate and thus oxidation of crustal sulfides.

The "Re-defining of the GOE" referred to in the title seems to be the contention that irreversible oxygenation occurred at 2.45 Ga rather than by ~2.3 Ga. But the only evidence given for irreversible oxygenation is highly depleted $d_{34}S$ in pyrite. What is it about depleted $d_{34}S$ values that speak to irreversibility? This might be true, but the authors don't really make this argument. I think they should.

I would say that the consensus on the GOE is that 1) it represents a transition from an anoxic Archean to an oxic Proterozoic and that 2) it occurred between 2.5 and 2.3 Ga. The present study argues that the GOE represents a transition from an anoxic Archean to an oxic Proterozoic and that it occurred at 2.45 Ga. So what is really new about this study?

As I understand it, the reason that others use a range of time (150 Myrs) to describe the GOE is that it occurred in fits and starts (because of the glacial events?) but that it was complete by 2.3 Ga. The authors seem to be arguing that it did not occur in fits and starts but all at once and at 2.45 Ga. This may be true, but I don't think the authors have successfully made this argument. I think they could make this argument, but they don't really go into it.

This is a very interesting and valuable data set. It captures an extended period during which an oxic atmosphere prevailed but MIF S was still being preserved locally in sedimentary rocks. It also captures some very depleted pyrite at 2.45 Ga. It is possible to argue that these depletions in $d_{34}S$ are a better indicator of irreversible oxygenation than is the absence of MIF at 2.3 per Bekker et al and others. The preservation of the MIF memory effect past 2.3 Ga supports this contention. Maybe this is what the authors are getting at. But if so, it's not clear.

I think the absolute age of Turee Creek deposition and the timing of glaciations are of secondary importance and should be subsidiary to the S story. As written, they are of equal importance. This undermines and confuses the S story in my opinion.

Reviewer #5 (Remarks to the Author):

The authors present Re-Os geochronology data to constrain sulfur isotope data related to an asynchronous GOE. I have strong reservations with regards to their interpretations of the sulfur data and feel that they haven't fully accounted for lithofacies controls. A starting point would be Ono et al. (2009) *Precam. Res.* v. 168, p.58-67.

Additionally, the Re-Os geochronology data is flawed and cannot be used to support their claims. Without these age constraints the paper is no longer novel or of interest to a wider readership. I was unable to reproduce the Re-Os geochronology data using Isoplot as they have not published uncertainties for the $^{187}\text{Re}/^{187}\text{Os}$ values or an error correlation function.

Response to Reviewers' comments:

Reviewer #2 (Remarks to the Author):

I am satisfied with the revised manuscript, and look forward to seeing it published in Nature Communications. In particular, the inclusion of additional information with respect to sulfide generations and petrogenesis is a welcome addition.

Reviewer #4 (Remarks to the Author):

Reviewer #3 raised a number of concerns about the authors' interpretation that the deposition of the Turee Creek Group was long-lived (250 Myrs). I think the authors have done a good job responding to these concerns. I find the argument that deposition continued until after 2.31 Ga to be compelling. But I think the argument that deposition occurred over 250 Myrs (2.45 to 2.2 Ga) to be a bit of an overstatement.

We thank the reviewer and agree that this is an important point that we cannot definitely say that Turee Creek sediments were deposited over 250 Myrs, and by extension that the MIF-S record also lasted that long. We also understand and agree that the interpretation needs to be focused on what we can say with total confidence: that Turee Creek sediments, and therefore the MIF-S record they contain, continued until after 2.31 Ga (but how long after is unknown at present). By scaling back our interpretation, our key points remain intact: that there is asynchronicity in the MIF-S record across paleo-continent, and that oxidative weathering and a MIF-S weathering "memory effect" is behind this.

Statements about deposition lasting until 2.25 Ga have been removed in the abstract and throughout the text.

The authors provide an age of 2.31 Ga for rocks deposited prior to and during the early stages of the MBM. They provide an age of 2.2 Ga for a sill intruding the MBM. They then use the thickness of overlying sediments to argue that deposition occurred until 2.25 Ga. I don't think this is very compelling; there are 100 Myrs missing here. I think all the authors can really say is that the MIF S memory effect was preserved in rocks younger than 2.31 Ga.

This is the key point: "all the authors can really say is that the MIF-S memory effect was preserved in rocks younger than 2.31 Ga". **We agree fully and the manuscript has been modified throughout to this end.**

The authors use sedimentation rates for the underlying Hamersley Group to estimate that deposition continued until as late as 2.25 Ga. But I don't think this argument is strong enough to use 2.25 Ga in the abstract, for example.

Agreed (see responses above). The 2.25 Ga age has been removed from the abstract and throughout the text.

I think that in addition to using rates from the Hamersley Group to estimate time they should use a typical Phanerozoic rate in order to come up with a conservative minimum age for the end of deposition. But most importantly, I think this is an unnecessary distraction from the S story. The significance of the preservation of the MIF memory effect past 2.3 Ga is enough to make this data set important. The authors don't need to push it back to 2.25 or 2.2 Ga to make it important.

We have removed all reliance on estimation of sedimentation rates because it is clearly problematic and a distraction. We have fully embraced the reviewer's position that the true focus of our paper should be (and now is) on the MIF-S signal being pushed to younger than 2.31 Ga in Western

Australia, which is clear evidence of MIF-S memory effects, and that we cannot use the terminal occurrence of small MIF-S signals to define the Great Oxidation Event.

Reviewer #3 also raised questions about the Re/Os methodology and the authors have addressed those concerns as well. The sampling approach (i.e. selecting samples separated by 100 m), also mentioned by #3 is not really addressed. The Re/Os system is outside of my expertise and I don't know how this sampling protocol could potentially undermine the reported age. But I think it would behoove the authors to address this concern. This date is very important to their argument that the MIF memory effect persisted until after 2.31 Ga.

We have now analyzed new samples for Re-Os geochronology, as well as revisited our sample selection approach to directly respond to these concerns (see response to reviewer 5 for details). That said, we fully agree and understand the reviewer's comment about the central importance of chronology in our findings. Accordingly, instead of hinging the interpretation almost entirely on the Re-Os age by itself, we now emphasize that there are multiple lines of evidence that the Meteorite Bore Member is 2.31 Gyr old or slightly younger:

- 1) the 2.31 Ga Re-Os age
- 2) the 2.34 Ga U-Pb zircon age (the paper of Caquineau et al.²⁹, in which this age is reported, is now in press in *Precamb. Res.*)
- 3) a maximum $2,454 \pm 23$ Myr U-Pb zircon age was obtained by Caquineau et al.²⁹ for the Boolgeeda diamictite documented here in core TCDP1. This age has been added in Figures 1, 2 and 3 and notified in the Supplementary Information.
- 4) through the global correlation of glacial units that our new data enables. Even if the Re-Os age is considered problematic (hopefully not anymore considering our new dataset), the 2.34 and 2.45 Gyr U-Pb zircon ages documented for the Meteorite Bore Member and the Boolgeeda diamictites cement the correlation with the globally recognized glacial deposits at ~ 2.31 Gyr and ~ 2.43 Gyr.

Reviewer #3 also suggested that the flux of sulfate aerosols would not be sufficient to keep up with sulfide weathering under oxic conditions. The authors do not address this in their response or in the revised text. But I think this is a good point. It doesn't seem possible to me that this volcanic flux would be sufficient to have a meaningful effect on the global sulfur budget once sulfide weathering commenced. I think the presence or absence of the MIF memory effect following the end of MIF production is controlled by the relative contributions to seawater sulfate of Archean sulfides carrying MIF and those that don't. Not all Archean sulfides will carry a MIF signal. Igneous sulfides, for example, would not.

Again, we agree fully with the reviewer and have modified our text accordingly. We are now clear in stating that under the oxidizing atmospheric conditions, weathering should be more important for sulfur fluxes to the ocean than volcanic gas input, and variability should be expected as a function of which sulfides are weathered (sedimentary vs. igneous).

My main concern with this manuscript is that it lacks clarity and I think that makes it difficult to determine exactly what the main points are. For example, there is an important difference between the end of MIF production when O₂ passed the 0.02 % PAL threshold and the preservation of MIF, in the form of MIF memory effects, afterwards. But the authors don't make this distinction. Consider this sentence from the introduction:

"The redox chemistry of the atmosphere before about 2.45 Gyr ago was likely dominated by carbon dioxide and methane with $pO_2 < 10^{-5}$ Present Atmospheric Level (PAL), as shown by the occurrence of MIF-S anomalies..."

Here the presence of MIF is tied directly to very low O₂. Compare to this sentence from the abstract:

"This extends the sedimentary record of MIF-S to younger than 2.31 Gyr by at least 60 Myr,

and clearly establishes the MIF-S signal as asynchronous between North America, Australia and South Africa."

Here the authors are talking about the MIF memory effect but they don't say this explicitly. This is important because not only is it confusing, it also makes it difficult to compare this study to that of Bekker et al., Guo et al., and Luo et al. These studies were looking for the absence of large MIF reflecting an end of MIF production and not for evidence of a MIF memory effect.

I guess everyone would agree that large MIF was no longer being produced at 2.3 Ga (at the completion of the GOE as commonly defined; e.g. Lyons et al., 2014). But MIF was still being preserved locally due to the memory effect. That is what this study demonstrates. Local and long-lived preservation of the MIF memory effect. The fact the Rooihooft/Timeball Hill shales, which are 2.3 Myrs old, don't show this memory effect is proof of its localized nature.

In retrospect and with the reviewer's insightful perspective, it is also apparent to us that clarity was lacking. Throughout the text, much effort has now been made to make the distinction clear between the process of atmospheric oxygenation terminating MIF-S production vs. oxidative weathering delivering ancient sulphur bearing MIF-S.

We have now corrected the text throughout to make the distinction clear and to make it clear that we are indeed talking about memory effects, and also made efforts to highlight the localized and variable nature expected for the memory effect.

If the MIF memory effect is one main point, then the other one is the fact that this study identifies highly depleted d34S pyrite at 2.45 Ga. This is very interesting and significant. The lack of MIF does not uniquely define an oxic atmosphere (the Archean Array passes through or very near the origin in d34S-D33S space) but I think pyrite S approaching -40 per mil is strong evidence for high sulfate and thus oxidation of crustal sulfides.

Also, this very negative d34S pyrite is also in an interval with high sulphur contents. Both point to enhanced sulphate input that in turn implies oxidative weathering. We have now clarified in several places the links between depleted d34S pyrite, sulphur content, and sulphate supply by oxidative weathering.

The "Re-defining of the GOE" referred to in the title seems to be the contention that irreversible oxygenation occurred at 2.45 Ga rather than by ~2.3 Ga. But the only evidence given for irreversible oxygenation is highly depleted d34S in pyrite. What is it about depleted 34S values that speak to irreversibility? This might be true, but the authors don't really make this argument. I think they should.

In the process of revising our manuscript to focus on what we can demonstrate with certainty, and to stick to our main points – that we have removed statements regarding the exact timing of atmospheric oxygenation and irreversibility. Our title has been changed accordingly.

I would say that the consensus on the GOE is that 1) it represents a transition from an anoxic Archean to an oxic Proterozoic and that 2) it occurred between 2.5 and 2.3 Ga. The present study argues that the GOE represents a transition from an anoxic Archean to an oxic Proterozoic and that it occurred at 2.45 Ga. So what is really new about this study?

As I understand it, the reason that others use a range of time (150 Myrs) to describe the GOE is that it occurred in fits and starts (because of the glacial events?) but that it was complete by 2.3 Ga. The authors seem to be arguing that it did not occur in fits and starts but all at once and at 2.45 Ga. This may be true, but I don't think the authors have successfully made this argument. I think they could make this argument, but they don't really go into it.

We now clarify that the point we are trying to make, which is truly new and one we can defend with confidence, is that the terminal disappearance of MIF-S ca. 2.3 Ga should not be used to attribute an “end” to the GOE, but rather the end of the (localized) memory effect. We have modified the text in multiple places to make this more clear.

This is a very interesting and valuable data set. It captures an extended period during which an oxic atmosphere prevailed but MIF S was still being preserved locally in sedimentary rocks. It also captures some very depleted pyrite at 2.45 Ga. It is possible to argue that these depletions in ^{34}S are a better indicator of irreversible oxygenation that is the absence of MIF at 2.3 per Bekker et al and others. The preservation of the MIF memory effect past 2.3 Ga supports this contention. Maybe this is what the authors are getting at. But if so, it's not clear.

We thank the reviewer for sharing our enthusiasm on this dataset. In keeping in line with the need to increase clarity and to stick to what we can defend with certainty, we have not emphasized so much how we define the actual onset of the GOE, but that we cannot define an end of the GOE ca. 2.33–2.31 because this has no meaning beyond signaling the end of the occurrence of MIF-S memory effects.

I think the absolute age of Turee Creek deposition and the timing of glaciations are of secondary importance and should be subsidiary to the S story. As written, they are of equal importance. This undermines and confuses the S story in my opinion.

The glacial correlations are important because they are used to time and correlate the disappearance of MIF-S signals worldwide. However we take the reviewer's point; the absolute age and extent of the Turee Creek Group deposition has been toned down in the revision, in particular the use of sedimentation rates to argue that Turee Creek sediments were deposited all the way to ~2.25 Gyr has been removed because it detracts from our main points.

Reviewer #5 (Remarks to the Author):

The authors present Re-Os geochronology data to constrain sulfur isotope data related to an asynchronous GOE. I have strong reservations with regards to their interpretations of the sulfur data and feel that they haven't fully accounted for lithofacies controls. A starting point would be Ono et al. (2009) Precam. Res. v. 168, p.58-67. Additionally, the Re-Os geochronology data is flawed and cannot be used to support their claims. Without these age constraints the paper is no longer novel or of interest to a wider readership. I was unable to reproduce the Re-Os geochronology data using Isoplot as they have not published uncertainties for the $^{187}\text{Re}/^{187}\text{Os}$ values or an error correlation function.

As discussed in more detail below, we feel that this general comment is somehow biased and likely misleading in part. One thing we agree with Reviewer #5, however, and other Reviewers before him, is that we failed to clearly justify the range of depths between samples used for Re-Os dating. The basic reason for considering samples within a relatively large depth range is that we are dealing with glacial sediments, which are very homogeneous sediments that formed in a semi-closed oceanic system, with no, or only little external inputs. Nonetheless, we agree that this aspect is of sufficient concern and require clearer justification. In order to address this issue we analysed 4 new diamictite samples (T2-252.55, T2-259.3, T2-264.6, T2-271.1) using the ‘ $\text{CrO}_3 - \text{H}_2\text{SO}_4$ ’ technique. These were collected within a depth range of ~20 meters from each other. Together with the previously analysed diamictite sample T2-272.46 (2 pyrite separate analyses and 1 bulk sediment analysis), we end up with a total of 7 analyses (5 bulk and 2 pyrite separates). Results define an isochron of 2312.5 ± 6.2 Ma (all analyses) or 2317 ± 16 Ma (bulk samples only), which is not significantly different from our previous age of 2309.0 ± 9.2 Ma using all samples or 2314 ± 19 Ma using the ‘ $\text{CrO}_3 - \text{H}_2\text{SO}_4$ ’ technique.

These new results clearly demonstrate that the approach used was the good one. It is worth re-emphasizing that the Re-Os age of 2.31 Ga for the Meteorite Bore Member glacial diamictite matches the minimum U-Pb zircon age of 2.34 Ga, which implies that the diamictites must be younger than

2.34 Ga. Thus, the 2.31 Gyr age has been robust since the first submission of the paper one year ago and has withstood further tests by new analyses.

Comments inserted in the text :

Line 274 : *The reason for mixing pyrites from four different samples of mudstone/siltstone of the Kungarra Formation underlying the Meteorite Bore Member is due to the paucity of pyrite.*

Mixing samples from this range of depths regardless of motivation is incorrect and invalidates the data. They are assuming that the initial Os composition hasn't changed over this depth, highly unlikely. Combining sedimentary samples and sulphides is also invalid, they should plot separately.

First, the sedimentary samples that have been analysed using *aqua regia* have been excluded from the dataset because this digestion method may potentially liberated Os from detrital material. Only the sedimentary samples analysed using $\text{CrO}_3 - \text{H}_2\text{SO}_4$ media are presented because this method preferentially liberate hydrogenous Os (Supplementary Figure 3b).

Second, we are now presenting data obtained on 5 diamictites samples collected within 20 meters from each others and analysed using the same $\text{CrO}_3 - \text{H}_2\text{SO}_4$ protocol. These are homogenous sediments deposited in a semi-closed environment. The only pyrite sample shown in Supplementary Figure 3a is sample T2 – 272.46 and its duplicate. The reason for this is that, it represents an early diagenetic pyrite showing the same hydrogenous source of Os and therefore should, most likely, preserve the same Os initial ratio. All other bulk and pyrite samples that were combined from different depths were excluded from the present dataset.

Third, the sedimentary and pyrite samples all together fulfil the main assumptions behind the isochron method:

- both the sediments and the diagenetic pyrites have been isotopically equilibrated at the same time;
- both the sediments and the diagenetic pyrites have the same initial ratio corresponding to that of contemporaneous seawater;
- no subsequent gain, loss, or redistribution of Re or Os occurred since deposition.

Accordingly, considering the small depth interval of sampling and high homogeneity of the diamictite lithology, we consider that the initial Os isotopic composition of seawater was quite stable during sedimentation of the analysed glacial diamictites. The initial $^{187}\text{Os}/^{188}\text{Os}$ value in our study (0.151 ± 0.0052) is slightly more radiogenic than that of mantle/extraterrestrial value, which corresponds to the majority of black shales between 2.7 and 2.0 Ga (Anbar et al., 2007; Hannah et al., 2004; Hannah et al., 2006; Yang et al., 2009; Kendall et al., 2013). Previously, the oldest known example of radiogenic Os delivery to seawater was demonstrated by sedimentary rocks of the Huronian Supergroup, deposited in the aftermath of a widespread Paleoproterozoic glaciation at ca. 2.3 Ga (Sekine, 2011). The slightly higher value than the chondritic initial Os reference value in our study may indicate that local paleo-seawater $^{187}\text{Os}/^{188}\text{Os}$ ratio was to some extent higher than the mantle value possibly due to an input from oxidative continental weathering, which supplied radiogenic Os from the upper crust.

With regards of combining sedimentary samples and sulphides, several high-profile publications show the results where syn-depositional sulphides are used together with mudstones/black shales on the same isochron and produce coherent results. For example, Hannah et al. (2006) showed that precise Re-Os age of 2004 ± 9 Ma can be achieved using both black shales (2006 ± 26 Ma) and intercalated sulphides (2004 ± 20 Ma) with almost identical initial $^{187}\text{Os}/^{188}\text{Os}$ ratio of 0.130 ± 0.042 and 0.133 ± 0.058 , respectively. Syn-sedimentary pyrite was also plotted on the same Re-Os isochron with organic-rich mudrocks by Cohen et al. (1999, EPSL). The reason behind this is that the main source of Os in organic-rich sediments consists of sulphides (pyrite) and organic material as a sorbent for Re and Os metals. The syn-sedimentary and early diagenetic pyrites have the same source of Re and Os as the organic-rich sediments, and therefore can be used together according to the isochron principles described there above.

Line 282 : *The analysis of these samples were replicated by starting digestion using CrO₃ – H₂SO₄ mixture as described by Selby and Creaser (2003)⁴⁹ to extract preferentially hydrogenous Re and Os.*

Need to explain why only 3 were digested this way - the values are distinct enough. All sedimentary samples need to be digested with the chromic acid solution.

As discussed above, we are now only considering samples digested using CrO₃ – H₂SO₄ media. In the first version of the paper, we didn't use this digestion method because the TOC contents of our samples are quite low. After the first round of review, we applied the CrO₃ – H₂SO₄ digestion method using quite large sample volume (up to 3 g).

Line 289 : *Rhenium was separated from a portion of the residual solution using anion exchange chromatography.*

See Cumming et al. (2013) Geology, for an updated methodology on Re columns

Cumming et al. (2013) have been using the method described by Selby and Creaser (2003) with the exception of Cr⁶⁺ to Cr³⁺ reduction technique (NaOH instead of SO₂ gas) prior to column chromatography for Re purification. Although this technique sounds easier in application, it does not invalidate the reduction technique described by Selby and Creaser (2003), because the final goal (reduction) is achieved by both methods.

Line 305: *The Re and Os concentrations in pyrite range from 1.9 to 3.6 ppb and from 109 to 262 ppt, 305 respectively (Supplementary Table 1).*

No uncertainties for the ¹⁸⁷Re/¹⁸⁸Os measurements are presented and the error correlation function (rho) is not presented. I can't reproduce this data.

Supplementary Table 1. Re-Os concentrations and Os isotopic compositions for shales and pyrite separates.

Sample	Lith	Re ppb	Total Os ppt	¹⁸⁷Re/¹⁸⁸Os	± 2σ	¹⁸⁷Os/¹⁸⁸Os	± 2σ
T2 - 252.55	Diamictite	3.70	350	69.02	0.41	2.855	0.018
T2 - 259.3	Diamictite	3.68	629	33.03	0.17	1.450	0.002
T2 - 264.3	Diamictite	4.32	391	73.84	0.41	3.048	0.012
T2 - 271.1	Diamictite	3.59	870	22.22	0.12	1.022	0.002
T2 - 272.46	Diamictite	4.05	418	54.20	0.31	2.288	0.005
T2-272.46	Py	2.89	109	865.93	3.30	34.090	0.010
T2 - 272.46	Py	3.58	206	147.14	0.52	5.940	0.004

Using these 7 points, we obtained:

Using only bulk sample analyses we obtained:

Line 308: This can be explained by the presence of variable quantities of sulphides, possibly due to 'nugget' effect.

Shouldn't occur if samples are properly processed e.g., crushing and powdering >20g sedimentary rock in a Zr-ceramic shatterbox. See Kendall et al., 2009, GSL Special Publication.

This issue is not critical anymore because the samples digested using *aqua regia* are not considered.

Line 316: The data indicate that some scatter is present in the system ($MSWD > 1$), probably due to variable initial $^{187}\text{Os}/^{188}\text{Os}$ ratios in the detrital material and/or possible influence of Archaean seawater, which may have affected the Os budget in the sediments, due to re-equilibration with hydrogenous Os from the ocean column.

Or the fact that you've taken samples from multiple depths and integrated them... This is a fundamental flaw in understanding the principles of a Re-Os isochron.

The basic reason for considering such a large depth range is that we are dealing with glacial sediments, which are very homogeneous sediments that formed in a semi-closed system ocean, with no, or only

little external inputs. We have performed new analyses of a single rock type (glacial diamictites) collected within a limited depth range (<20m) and demonstrate that the original dataset was correct.

Reviewers' comments:

Reviewer #1 (Remarks to the Author):

The Philippot et al. paper has improved from the first time I reviewed this paper about one year ago. Major criticisms in my first review related to the Re-Os work. With additional analyses and elimination of questionable data, the Re-Os age is more credible now, although some additional detail/justification to support the approach used would be ideal. Suggest publication after minor revisions.

Line 94: No information given here in the main text about the timing of diagenesis for the pyrites used for Re-Os dating. Early? Late? There is abundant information about pyrite morphology and timing in the supplementary, but you need to be clearer in the main text about the timing of diagenesis.

When referring to ages, use ka, Ma, Ga. When referring to time durations, use kyr, Myr, Gyr.

Lines 93-95 is misleading because the sentence implies that all analyses were carried out on diagenetic pyrites. Compare with lines 269-271, describing 5 diamictite and two pyrite analyses.

Provide a more detailed text description about the petrography of the MBM diamictites (used for Re-Os dating) after the sentence mentioning the local depositional environment in supplementary, to accompany the photographs shown in the supplementary. Was the matrix fine-grained and organic-rich? But don't call glacial sediments "homogeneous" as in the rebuttal to reviewer #5 – if the dated samples are really diamictites, then they are not well sorted sediments and have a wide range of grain sizes?

The authors make minimal effort in main text and supplementary to justify the use of diamictites for Re-Os dating. As this is an uncommon material for Re-Os dating, it would be wise to do this and also justify the large ~20 m sample interval. Most Re-Os sedimentary geochronology is done on organic-rich mudrocks using intervals < 1 m thick.

The authors ignored reviewer #5 regarding the use of the error correlation function rho. Most Re-Os papers report it. The 188Os appears in both isotope ratios on a Re-Os isochron diagram and the 188Os signal is small vs 187Re and 187Os signals on a NTIMS instrument – hence error correlation occurs. Report the rho value for each sample and use it in the isochron regression.

Reviewer #4 (Remarks to the Author):

I am satisfied with the changes made in this revised manuscript.

Reviewer #5 (Remarks to the Author):

The authors have made multiple changes to improve the robustness of the Re/Os age for these samples and altered the text. However, they are using an isochron technique for data regression yet have not provided an error correlation function value for their Re/Os data. As a result the age can't be fully verified or reliably reproduced from the data tables. In an isochron system, the uncertainties of the parent and daughter isotopic ratios typically are correlated and qualified by error correlations. By excluding error correlations for linear regressions the authors risk generating an invalid uncertainty for the slope, and thus an invalid age and uncertainty

Reviewer #1 (Remarks to the Author):

The Philippot et al. paper has improved from the first time I reviewed this paper about one year ago. Major criticisms in my first review related to the Re-Os work. With additional analyses and elimination of questionable data, the Re-Os age is more credible now, although some additional detail/justification to support the approach used would be ideal. Suggest publication after minor revisions.

Line 94: No information given here in the main text about the timing of diagenesis for the pyrites used for Re-Os dating. Early? Late? There is abundant information about pyrite morphology and timing in the supplementary, but you need to be clearer in the main text about the timing of diagenesis.

We modified the sentence

To provide a precise calibration point within the Turee Creek Group, Re-Os dating was performed on diagenetic pyrites (Supplementary Figs. 5, 6, 7, 8, 9 and Supplementary Table 2)

As

To provide a precise calibration point within the Turee Creek Group, Re-Os dating was performed on glacial diamictites (bulk mudstone matrix and early diagenetic pyrite separates; Supplementary Figs. 5, 6, 7, 8, 9 and Supplementary Table 2) sampled at the base of the MBM diamictite (the rationale for choosing the MBM diamictites for Re-Os dating is discussed in the Supplementary Information).

When referring to ages, use ka, Ma, Ga. When referring to time durations, use kyr, Myr, Gyr.

Done

Lines 93-95 is misleading because the sentence implies that all analyses were carried out on diagenetic pyrites. Compare with lines 269-271, describing 5 diamictite and two pyrite analyses.

See above comment line 94

Provide a more detailed text description about the petrography of the MBM diamictites (used for Re-Os dating) after the sentence mentioning the local depositional environment in supplementary, to accompany the photographs shown in the supplementary. Was the matrix fine-grained and organic-rich? But don't call glacial sediments "homogeneous" as in the rebuttal to reviewer #5 – if the dated samples are really diamictites, then they are not well sorted sediments and have a wide range of grain sizes?

The authors make minimal effort in main text and supplementary to justify the use of diamictites for Re-Os dating. As this is an uncommon material for Re-Os dating, it would be wise to do this and also

justify the large ~20 m sample interval. Most Re-Os sedimentary geochronology is done on organic-rich mudrocks using intervals < 1 m thick. `

We agree that the term homogeneous is subject to confusion in the present case and is therefore inappropriate. What we meant by homogeneous is that the 400 meters of diamictites consist of a very uniform fine-grained matrix (mudstone) with no evidence of lithological bedding. This matrix contains abundant randomly distributed dropstones of various sizes as shown in Supplementary Figure 4d.

We added the following text in the main text (see comment Line 94 above)

(the reasons for choosing the Meteorite Bore Member diamictites for Re-Os dating are discussed in the Supplementary Information).

And in the Supplementary Information Line 105-131

The main lithologies exposed at the surface are shown in Supplementary Fig. 4 together with their drill core equivalent collected at depths. 135 samples from the three drill cores were collected to reconstruct a chemostratigraphic record of sulphur isotopes throughout the Turee Creek Group. In addition, 5 samples were selected at the base of the Meteorite Bore Member diamictites to perform Re-Os chronology.

The Meteorite Bore Member represents a massive package of ~400 m thick glacial diamictite showing no apparent bedding. The sequence is composed of chlorite-bearing mudstone containing randomly distributed dropstones of various sizes (mm- to m-scale) and origin (carbonate, chert, gneiss, quartzite and local pockets of sandstone showing convolute structure). The samples used for Re-Os dating were collected throughout a ~20 meter scale interval between 252.55 and 272.46 metres depth. This section consists of a uniform mudstone matrix with local dropstones one to several centimetres in size (Supplementary Figure 4d, sample T2-253.3) and randomly distributed nodular aggregates of pyrites interpreted to be of early diagenetic origin (see below). For the five diamictites used for bulk rock dating (T2-252.55, T2-259.3, T2-264.3, T2-271.1 and T2-272.46, Supplementary Table 1), we were careful to only collect the mudstone part of the sample and avoid portions with dropstones. The reasons for choosing the Meteorite Bore Member diamictite for dating are as follows:

1) Bulk-rock chemical analyses showed that the entire section is relatively static in composition with sulphur and Total Organic Content (TOC) contents of about 1300 to 2300 ppm and 500 to 1000 ppm, respectively.

2) A detrital zircon U-Pb age of 2340 ± 22 Ma obtained by Caquineau et al (2018)⁸ was obtained in the same sedimentary interval, thus providing an independent test of evaluating the significance of the Re-Os age.

3) A detrital zircon U-Pb age of 2.454 ± 23 Ma was obtained by Caquineau et al (2018)⁸ on the diamictite horizon of the Boolgeeda Iron Formation at the base of the TCDP1 core (Figure 3). Since the Boolgeeda Iron Formation in TCDP1 and the overlying Kungarra Formation containing the MBM diamictites in TCDP2 are in sedimentary continuity offered a second independent mean for evaluating the significance of the Re-Os age as well as the rate of sedimentation of the Kungarra Formation.

The authors ignored reviewer #5 regarding the use of the error correlation function rho. Most Re-Os papers report it. The 188Os appears in both isotope ratios on a Re-Os isochron diagram and the 188Os signal is small vs 187Re and 187Os signals on a NTIMS instrument – hence error correlation occurs. Report the rho value for each sample and use it in the isochron regression.

See below answer to Reviewer #5 comment

Reviewer #4 (Remarks to the Author):

I am satisfied with the changes made in this revised manuscript.

Reviewer #5 (Remarks to the Author):

The authors have made multiple changes to improve the robustness of the Re/Os age for these samples and altered the text. However, they are using an isochron technique for data regression yet have not provided an error correlation function value for their Re/Os data. As a result the age can't be fully verified or reliably reproduced from the data tables. In an isochron system, the uncertainties of the parent and daughter isotopic ratios typically are correlated and qualified by error correlations. By excluding error correlations for linear regressions the authors risk generating an invalid uncertainty for the slope, and thus an invalid age and uncertainty

We added the following sentence in Methods (line 307-312) and corrected the new age obtained both in the text and the figures. Note that the new age (2312.7 ± 5.6 Ma) does not differ significantly from the previous age (2312.5 ± 6.2 Ma).

The Re-Os isochron regressions were performed using Isoplot⁴⁶, applying the ¹⁸⁷Re decay constant of $1.666 \times 10^{-11} \text{ year}^{-1}$ (Ref 47). The uncertainties for ¹⁸⁷Re/¹⁸⁸Os and ¹⁸⁷Os/¹⁸⁸Os were determined by error propagation, and the error correlation ρ ('rho') (Kendall et al. (2009)⁴⁸ and references therein). Error correlation between ¹⁸⁷Re/¹⁸⁸Os and ¹⁸⁷Os/¹⁸⁸Os is usually significant for shales and diamictites⁴⁸ (0.35 to 0.57 for our samples), and is therefore recommended for usage on Re-Os isochron regression diagrams.

The plot of isotope data for the five diamictites and the two pyrite separates on the Re-Os isochron diagram (Supplementary Figure 3a, Supplementary Table 1) defines a best-fit line with an age of 2312.7 ± 5.6 Ma (MSWD = 1.4) and an initial ¹⁸⁷Os/¹⁸⁸Os ratio of 0.151 ± 0.005 . The five diamictite samples displayed a similar age of 2316 ± 12 Ma (Supplementary Figure 3b), with a well-defined initial ratio of 0.149 ± 0.008 .

We added the rho value in Supplementary Figure 1.

Supplementary Table 1. Re-Os concentrations and Os isotopic compositions for diamictites and pyrite separates.

Sample	Lithology	Re ppb	Total Os ppt	¹⁸⁷ Re/ ¹⁸⁸ Os	$\pm 2\sigma$	¹⁸⁷ Os/ ¹⁸⁸ Os	$\pm 2\sigma$	Rho*
T2 – 252.55	Diamictite	3.70	350	69.02	0.41	2.855	0.018	0.35
T2 – 259.3	Diamictite	3.68	629	33.03	0.17	1.450	0.002	0.47
T2 – 264.3	Diamictite	4.32	391	73.84	0.41	3.048	0.012	0.53
T2 – 271.1	Diamictite	3.59	870	22.22	0.12	1.022	0.002	0.40
T2 - 272.46	Diamictite	4.05	418	54.20	0.31	2.288	0.005	0.46
T2-272.46	Py	2.89	109	865.93	3.30	34.090	0.010	0.54
T2 - 272.46	Py	3.58	206	147.14	0.52	5.940	0.004	0.57

Rho* stands for error correlation²³.

REVIEWERS' COMMENTS:

Reviewer #1 (Remarks to the Author):

I am satisfied with the changes made by the authors in response to my comments.